# A novel representation of biological nitrogen fixation and competitive dynamics between nitrogen-fixing and non-fixing plants in a land model (GFDL LM4.1-BNF)

Sian Kou-Giesbrecht[1]*, Sergey Malyshev[2], Isabel Martínez Cano[3], Stephen W. Pacala[3], Elena Shevliakova[2], Thomas A. Bytnerowicz[4], and Duncan N. L. Menge[1]

[1]Department of Ecology, Evolution and Environmental Biology, Columbia University, New York, 10027, USA
[2]Geophysical Fluid Dynamics Laboratory (GFDL), National Oceanic and Atmospheric Administration (NOAA), Princeton, 08540, USA
[3]Department of Ecology and Evolutionary Biology, Princeton University, Princeton, 08544, USA
[4]Department of Integrative Biology, The University of Texas, Austin, 78712, USA

*Correspondence to*: Sian Kou-Giesbrecht (sk4220@columbia.edu)

**Abstract.** Representing biological nitrogen fixation (BNF) is an important challenge for coupled carbon (C) and (N) land models. Initial representations of BNF in land models applied simplified phenomenological relationships. More recent representations of BNF are mechanistic and include the dynamic response of symbiotic BNF to N limitation of plant growth. However, they generally do not include the competitive dynamics between N-fixing and non-fixing plants, which is a key ecological mechanism that determines ecosystem-scale symbiotic BNF. Furthermore, asymbiotic BNF is generally not included in land models. Here, we present LM4.1-BNF, a novel representation of BNF (asymbiotic and symbiotic) and an updated representation of N cycling in the Geophysical Fluid Dynamics Laboratory Land Model 4.1 (LM4.1). LM4.1-BNF incorporates a mechanistic representation of asymbiotic BNF by soil microbes, a representation of the competitive dynamics between N-fixing and non-fixing plants, and distinct asymbiotic and symbiotic BNF temperature responses derived from corresponding observations. LM4.1-BNF makes reasonable estimations of major carbon (C) and N pools and fluxes and their temporal dynamics, in comparison to the previous version of LM4.1 with N cycling (LM3-SNAP) and to previous representations of BNF in land models generally (phenomenological representations and those without competitive dynamics between N-fixing and non-fixing plants and/or asymbiotic BNF) at a temperate forest site. LM4.1-BNF effectively reproduces asymbiotic BNF rate (13 kg N ha$^{-1}$ yr$^{-1}$) in comparison to observations (11 kg N ha$^{-1}$ yr$^{-1}$). LM4.1-BNF effectively reproduces the temporal dynamics of symbiotic BNF rate: LM4.1-BNF simulates a symbiotic BNF pulse in early succession that reaches 73 kg N ha$^{-1}$ yr$^{-1}$ at 15 years then declines to ~ 0 kg N ha$^{-1}$ yr$^{-1}$ at 300 years, similarly to observed symbiotic BNF which reaches 75 kg N ha$^{-1}$ yr$^{-1}$ at 17 years then declines to ~ 0 kg N ha$^{-1}$ yr$^{-1}$ in late successional forests. As such, LM4.1-BNF can be applied to project the dynamic response of vegetation to N limitation of plant growth and the degree to which this will constrain the terrestrial C sink under elevated atmospheric $CO_2$ concentration and other global change factors.

# 1 Introduction

The terrestrial carbon (C) sink is controlled by the availability of nitrogen (N) for plant growth (Elser et al., 2007; LeBauer
and Treseder, 2008; Wright et al., 2018). Land models are applied to project the terrestrial C sink (Arora et al., 2020) and are
progressively incorporating representations of N cycling and N limitation of plant growth (Goll et al., 2017; Lawrence et al.,
2019; Medvigy et al., 2009; Smith et al., 2014; Wang et al., 2010; Yang et al., 2009; Zaehle and Friend, 2010). However, the
degree to which N limitation of plant growth will constrain the terrestrial C sink under elevated atmospheric $CO_2$
concentration is unresolved (Terrer et al., 2019), as there is substantial variation between different land models (Wieder et
al., 2015b).

The representation of biological N fixation (BNF), the primary natural input of N to terrestrial ecosystems (Fowler et al.,
2013; Vitousek et al., 2013), is a key challenge to incorporating N cycling into land models because of its complexity
(Davies-Barnard et al., 2020; Meyerholt et al., 2020; Stocker et al., 2016; Thomas et al., 2015; Wieder et al., 2015a). BNF
occurs in multiple niches across terrestrial ecosystems: by symbioses between N-fixing bacteria living in root nodules of
plants (hereafter, symbiotic BNF) and by a host of other organisms such as soil microbes, bryophytes, and lichens (hereafter,
asymbiotic BNF for simplicity although some of these organisms are symbiotic associations; see Reed et al., 2011).
Symbiotic and asymbiotic BNF are regulated by a myriad of abiotic and biotic controls, that vary temporally, spatially, and
among different niches (Zheng et al., 2019). In particular, symbiotic BNF responds dynamically to N limitation of plant
growth: it is up-regulated under N limitation of plant growth and down-regulated under non-N limitation of plant growth
(Vitousek et al., 2013). BNF could, as such, be pivotal to overcoming N limitation of plant growth under elevated
atmospheric $CO_2$ concentration (Liang et al., 2016; Terrer et al., 2016, 2018).

Many coupled C-N land models use the empirical relationship of BNF with either net primary production (NPP; Goll et al.,
2017) or evapotranspiration (ET; B. Smith et al., 2014; Yang et al., 2009; Zaehle & Friend, 2010) to represent BNF.
However, these are simplified phenomenological relationships that are not based on the ecological mechanisms underlying
BNF (Cleveland et al., 1999). Furthermore, implementing and comparing a NPP-based and ET-based representation of BNF
within a land model (CLM5) resulted in projections of the terrestrial C sink that differed by 50 Pg C in 2100 under the
representative concentration pathway 8.5 (RCP8.5; Wieder et al., 2015a). Finally, a recent meta-analysis of BNF found no
evidence for the empirical relationship of BNF with either NPP or ET (Davies-Barnard and Friedlingstein, 2020).

Recent coupled C-N land models have simulated symbiotic BNF mechanistically rather than phenomenologically as
responding dynamically to N limitation of plant growth. The Geophysical Fluid Dynamics Laboratory (GFDL) Land Model
3 (LM3) can include the Symbiotic Nitrogen Acquisition by Plants (SNAP) model (Sulman et al., 2019), in which plant C
allocation to N-fixing bacteria is optimized to maximize plant growth. However, LM3-SNAP as well as other land models
that have implemented a mechanistic representation of symbiotic BNF such as CLM5 (Lawrence et al., 2019), CABLE
(Haverd et al., 2018; Peng et al., 2020; Wang et al., 2010), and E3SM (Zhu et al., 2019), represent a single general plant C
pool capable of BNF and cannot represent community dynamics. In observed ecosystems, symbiotic BNF responds

dynamically to N limitation of plant growth at both the population scale (via individual-scale regulation of symbiotic BNF rate; Menge, Wolf, & Funk, 2015) and at the community scale (via competitive dynamics between N-fixing and non-fixing plants; Boring and Swank, 1984; Chapin III et al., 1994; Menge and Hedin, 2009): Under strong N limitation, N-fixing plants up-regulate symbiotic BNF rate and have a competitive advantage over non-fixing plants, but, under weak N limitation, N-fixing plants down-regulate symbiotic BNF rate and are competitively excluded by non-fixing plants because of the high C cost of symbiotic BNF (Gutschick, 1981; Sheffer et al., 2015). As such, the competitive dynamics between N-fixing and non-fixing plants is a key ecological mechanism that could determine ecosystem-scale symbiotic BNF. Finally, the abundance of N-fixing trees is spatially variable (Menge et al., 2019; Staccone et al., 2020), but its representation is not possible in land models that represent a single general plant C pool capable of BNF although is necessary to accurately estimate regional symbiotic BNF.

Asymbiotic BNF is generally not included in coupled C-N land models. However, asymbiotic BNF is an important natural input of N to terrestrial ecosystems: In some terrestrial ecosystems, asymbiotic BNF is on par with symbiotic BNF and asymbiotic BNF has been suggested to account for a substantial proportion of global BNF (Reed et al., 2011). Phenomenological representations of BNF merge asymbiotic and symbiotic BNF, although they are regulated by different controls (Zheng et al., 2019). Mechanistic representations of BNF either merge asymbiotic and symbiotic BNF (e.g. LM3-SNAP; Sulman et al., 2019), represent asymbiotic BNF as a constant from averaged observations (e.g. CABLE; Wang and Houlton, 2009), or represent asymbiotic BNF phenomenologically as a function of ET (e.g. CLM5; Lawrence et al., 2019). Importantly, although asymbiotic and symbiotic BNF exhibit different temperature responses (Bytnerowicz et al., in review), the symbiotic BNF temperature response is, when included, derived primarily from asymbiotic BNF observations (Houlton et al., 2008), and the asymbiotic BNF temperature response is omitted.

Here, we present LM4.1-BNF, a novel representation of BNF and an updated representation of N cycling in the GFDL land model 4.1 (LM4.1; Shevliakova et al., in prep). LM4.1 includes height-structured competition for light and water between plant cohorts using the perfect plasticity approximation (Martinez Cano et al., 2020; Purves et al., 2008; Strigul et al., 2008; Weng et al., 2015). LM4.1-BNF builds on the framework of LM4.1, including competition for light, water, and N between plant cohorts that associate with N-fixing bacteria and non-fixer plant cohorts. LM4.1-BNF introduces several improvements to the representation of N cycling in LM3-SNAP by incorporating novel representations of the following ecological mechanisms:

1. *Symbiotic BNF and competitive dynamics between N-fixing and non-fixing plants*: Plant cohorts with a N-fixer vegetation type conduct symbiotic BNF and compete with plant cohorts with a non-fixer vegetation type.

2. *Asymbiotic BNF*: Soil microbes conduct asymbiotic BNF, as well as decomposition, nitrification, and denitrification.

3. *BNF temperature response*: Asymbiotic and symbiotic BNF have different temperature responses derived from asymbiotic BNF observations (Houlton et al., 2008) and symbiotic BNF observations (Bytnerowicz et al., in review) respectively.

4. *N limitation*: N limitation is determined by current stored non-structural N relative to the demand for non-structural N. N limitation increases active root uptake of inorganic N and decreases root N exudation following observations (Canarini et al., 2019; Nacry et al., 2013).

5. *Dynamic plant C allocation to growth and N uptake*: N limitation decreases the growth of leaves, sapwood, and seeds, proportionally increasing the growth of fine roots following observations (Poorter et al., 2012). N limitation stimulates C allocation to N uptake (including symbiotic BNF) relative to growth. C limitation, which is determined by current stored non-structural C relative to the demand for non-structural C, stimulates C allocation to growth relative to N uptake. Thereby, plant C allocation is optimized to maximize growth following observations (Rastetter and Shaver, 1992).

We focus our analysis on temperate forests which are generally N-limited (Elser et al., 2007; LeBauer and Treseder, 2008). We parameterise a N-fixer vegetation type based on *Robinia pseudoacacia* (black locust), which is the most abundant N-fixing tree species in the coterminous US, accounting for 64% of tree-associated BNF in the coterminous US (Staccone et al., 2020). We compare *Robinia* to a non-fixer vegetation type based on *Acer rubrum* (red maple), which is the most abundant non-fixing tree species in the North region of the coterminous US (Oswalt et al., 2019). We evaluate LM4.1-BNF at Coweeta Hydrologic Laboratory in North Carolina, US, which has observations on symbiotic BNF by *Robinia* (Boring and Swank, 1984).

We conduct three analyses to assess the performance of LM4.1-BNF in estimating major C and N pools and fluxes in comparison to previous representations of BNF in land models generally. In the first analysis, we compare mechanistic and phenomenological representations of BNF; We compare LM4.1-BNF (with BNF represented mechanistically as described above) to LM4.1-BNF with BNF represented as a function of NPP and to LM4.1-BNF with BNF represented as a function of ET. In the second analysis, we examine the role of competitive dynamics between N-fixing and non-fixing plants; We compare LM4.1-BNF simulations with both *Robinia* and *Acer* to LM4.1-BNF simulations with only *Acer* and LM4.1-BNF simulations with only *Acer* that can associate with N-fixing bacteria, which are representative of land models that represent a single general plant C pool capable of BNF and cannot represent community dynamics. In the third analysis, we examine the role of asymbiotic BNF; We compare LM4.1-BNF simulations with asymbiotic BNF to LM4.1-BNF simulations without asymbiotic BNF, which is representative of land models that do not include asymbiotic BNF.

## 2 Model description

### 2.1 Overview of a land tile and vegetation types

We provide an overview of LM4.1-BNF with a focus on the novel elements relative to LM4.1 (Shevliakova et al., in prep) and LM3-SNAP (Sulman et al., 2019). A complete description of LM4.1-BNF is available in Appendix A. Note that LM4.1 can be coupled with the GFDL atmosphere model to serve as a base for the GFDL climate and Earth system models (Zhao et al., 2018a, 2018b).

LM4.1-BNF consists of a grid, in which grid cells are approximately 100 km by 100 km. LM4.1-BNF represents the heterogeneity of the land surface as a mosaic of land tiles within a grid cell. Each land tile represents a fraction of the grid cell area and does not have an associated location within the grid cell. A land tile may represent natural vegetation at a given stage of recovery post-disturbance, urban area, pastureland, rangeland, or cropland. Land tiles are created dynamically due to a disturbance, such as human land use, fire, or natural mortality of vegetation.

A land tile contains multiple plant cohorts that compete for light and water following the Perfect Plasticity Approximation (Martinez Cano et al., 2020; Purves et al., 2008; Strigul et al., 2008; Weng et al., 2015) and compete for N (presented below). Plant cohorts consist of identical individual trees belonging to a vegetation type that occupy a given canopy layer and that have a spatial density (determined by recruitment and mortality). A vegetation type can be associated with exclusively arbuscular mycorrhizae (AM), exclusively ectomycorrhizae (EM), both AM and N-fixing bacteria, or both EM and N-fixing bacteria. A land tile can contain multiple plant cohorts of the same or of different vegetation types. As such, there is intraspecific competition (among plant cohorts of the same vegetation type within a tile) and interspecific competition (among plant cohorts of different vegetation types within a tile). Growth is based on allometric equations (Eq. (A40-42)) and is modulated by N availability. Recruitment and mortality follow Weng et al., 2015 and Martinez Cano et al., (2020) and are not directly influenced by N availability but are indirectly influenced by N availability via its effect on growth.

There are six plant tissue C and N pools: leaf, fine root, sapwood, heartwood, seed, and non-structural C or N. The C:N ratios of the leaf, fine root, sapwood, heartwood, and seed tissue pools are fixed (for a given vegetation type) (Table 1, D1, and D2). There are three soil organic C and N pools (labile plant-derived, labile microbe-derived, and recalcitrant) and two soil inorganic N pools (ammonium ($NH_4^+$) and nitrate ($NO_3^-$)) in each soil layer. There are 20 soil layers of varying thickness to a total depth of 10 m. Soil C and N are transferred between soil layers via leaching. The soil C:N ratio is not fixed. Fig. 1 displays a diagram of key C and N pools and fluxes.

We define a N-fixer vegetation type with a parameterisation based on *Robinia pseudoacacia* and a non-fixer vegetation type with a parameterisation based on *Acer rubrum*. Both *Acer* and *Robinia* associate with AM. We used the US Forest Inventory and Analysis (FIA) database (US Forest Service, 2020a), the US FIA Forest Health Monitoring database (US Forest Service, 2020b), and the Biomass and Allometry Database (BAAD; Falster et al., 2015) to parameterise the allometries of these vegetation types (Eq. (A40-42); Appendix B and C). The vegetation types also differed in other key traits (Table 1). In particular, the C:N ratio of leaves differed between vegetation types that associated with AM, EM, and N-fixing bacteria (Adams et al., 2016; Averill et al., 2019). See Table D1 for all vegetation type-specific parameters. Other model parameters are from Weng et al., (2015) or Sulman et al., (2019) or are derived from published observations (Appendix C). Some parameters were not well constrained by available observations and were tuned to fit to observed patterns of C and N cycling in temperate forests (Appendix C). See Table D2 for general parameters.

**Table 1: Key parameter differences between vegetation types. See Table D1 for remaining vegetation type-specific parameters.**

| Vegetation type | Leaf C:N ratio | Maximum rate of carboxylation ($V_{cmax}$) at 15°C | Wood C density | Leaf mass per area |
|---|---|---|---|---|
| *Acer rubrum* | 30 kg C kg N$^{-1}$ | 17 μmol m$^{-2}$ s$^{-1}$ | 340 kg C m$^{-3}$ | 0.0482 kg C m$^{-2}$ |
| *Robinia pseudoacacia* | 14 kg C kg N$^{-1}$ | 23 μmol m$^{-2}$ s$^{-1}$ | 280 kg C m$^{-3}$ | 0.0380 kg C m$^{-2}$ |

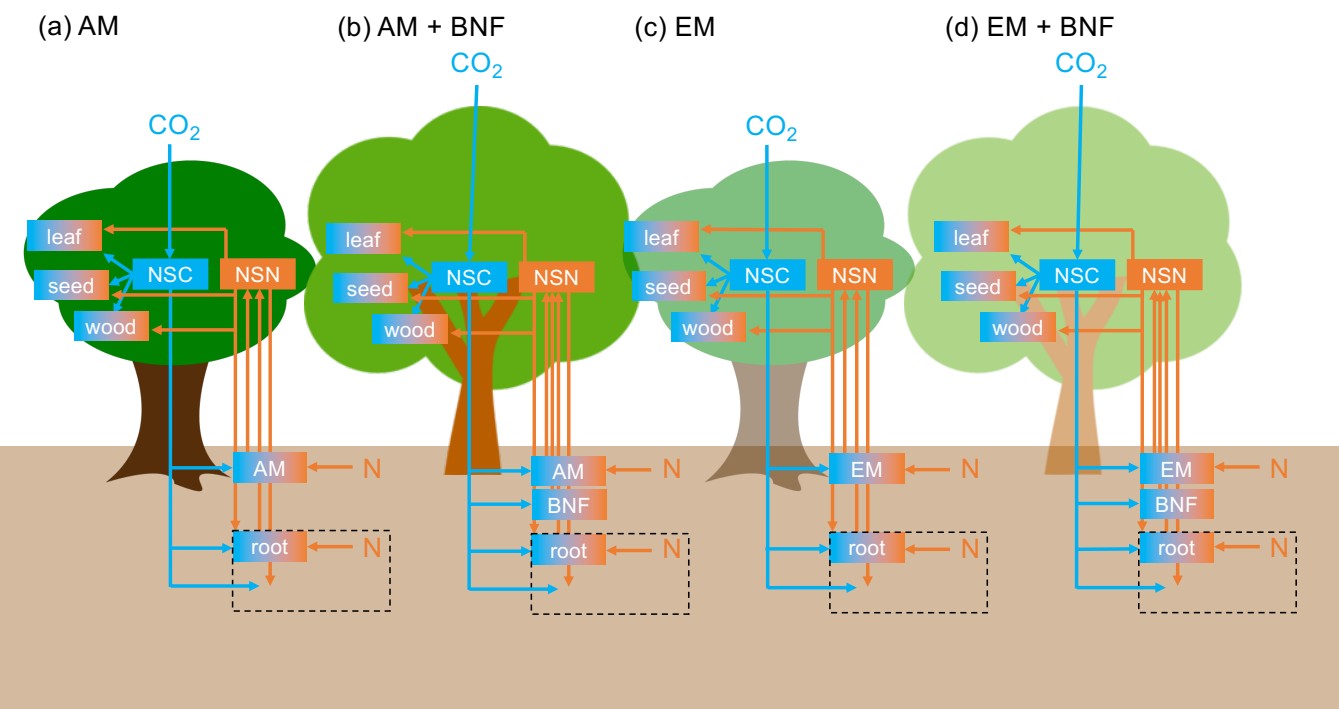

**Figure 1: Diagram of key C and N pools (boxes) and fluxes (arrows). C pools and fluxes are indicated in blue. N pools and fluxes are indicated in orange. NSC represents non-structural C and NSN represents non-structural N. Orange and blue boxes have a fixed C:N ratio. Plant turnover, symbiont turnover, and soil C and N pools and fluxes are not displayed for visual clarity. The black dashed box represents the rhizosphere. a) Vegetation type that associates with arbuscular mycorrhizae (AM). (b) Vegetation type that associates with AM and N-fixing bacteria. (c) Vegetation type that associates with ectomycorrhizae (EM). (d) Vegetation type that associates with EM and N-fixing bacteria.**

**2.2 Symbiotic BNF and N uptake by roots, AM, and EM**

All vegetation types take up inorganic N via passive and active root uptake. Passive root uptake of inorganic N follows LM3-SNAP (Eq. (A1)). Active root uptake of inorganic N follows LM3-SNAP but is modified to increase with N stress following observations (Nacry et al., 2013) (described in further detail below; Eq. (16)). AM take up inorganic N following LM3-SNAP (Eq. (A4)). EM decompose and take up organic C and N following LM3-SNAP but is modified to additionally take up inorganic N following observations (Phillips et al., 2013) (Eq. (A8)).

The symbiotic BNF rate by N-fixing bacteria ($N_{Nfix}$; [kg N indiv$^{-1}$ yr$^{-1}$]) is

$$N_{Nfix} = r_{Nfix}B_{Nfix}f_s(T) \tag{1}$$

where $r_{Nfix}$ is a rate constant, $B_{Nfix}$ is the biomass C of the nodule (includes both plant and N-fixing bacteria tissue) [kg C indiv$^{-1}$], and $f_s(T)$ is the soil temperature dependence function. For *Robinia,*

$$f_s(T) = \max\left[0.0, \left(\frac{45.67-(T-273.15)}{45.67-31.89}\right)\left(\frac{(T-273.15)-1.43}{31.89-1.43}\right)^{\frac{31.89-1.43}{45.67-31.89}}\right] \tag{2}$$

where $T$ is the average soil temperature across soil layers [K]. This function reaches its maximum at 31.9 ˚C (Fig. 2). It is a modified beta distribution function (Yan and Hunt, 1999) and is derived from Bytnerowicz et al., (in review).

Respiration associated with symbiotic BNF is $cost_{Nfix}N_{Nfix}$, where $cost_{Nfix}$ is the C cost of symbiotic BNF per unit N (Eq. (A52)).

Note that the description of EM is included although both *Acer* and *Robinia* associate with AM.

### 2.3 Asymbiotic BNF

Soil microbes are represented as a single C pool that conducts decomposition, nitrification, denitrification, and asymbiotic BNF. The rates of C and N decomposition, rates of C and N decomposition during denitrification, rates of change of biomass C and N, and maintenance respiration rate of soil microbes follow LM3-SNAP (Eq. (A11-A19)). The N surplus or deficit of soil microbes and C and N growth rates of soil microbes are modified to include asymbiotic BNF (Eq. (A22-24)).

The asymbiotic BNF rate of soil microbes in soil layer k ($N_{Nfix\,asymb}(k)$; [kg N m$^{-2}$ yr$^{-1}$]) is

$$N_{Nfix\,asymb}(k) = r_{Nfix\,asymb}C_M(k)f\big(T(k)\big) \tag{3}$$

where $r_{Nfix\,asymb}$ is a rate constant, $C_M(k)$ is the biomass C of soil microbes in soil layer k [kg C m$^{-2}$], and $f_a\big(T(k)\big)$ is the soil temperature dependence function.

$$f_a\big(T(k)\big) = e^{-2.6+0.21(T(k)-273.15)\left(1-\frac{0.5(T(k)-273.15)}{24.4}\right)} \tag{4}$$

This function reaches its maximum at 24.4 ˚C (Fig. 2). It is a modified normal distribution function and is derived from the observations compiled by Houlton et al., 2008 with the study of symbiotic BNF removed (Schomberg and Weaver, 1992) and is normalized to a maximum of 1.

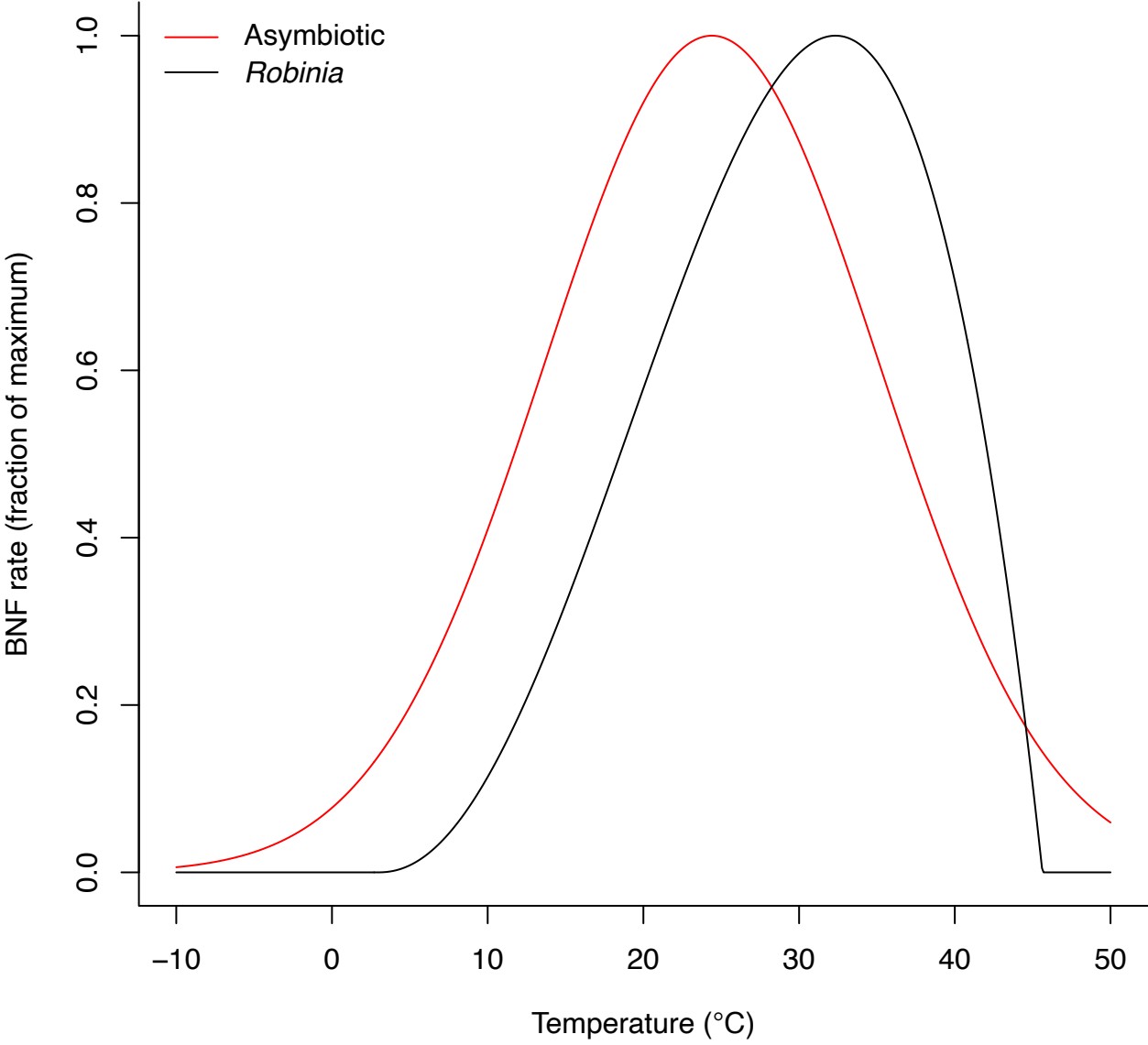

**Figure 2: Soil temperature dependence function of asymbiotic BNF and symbiotic BNF by *Robinia*. Asymbiotic BNF reaches its maximum at 24.4 °C and is derived from (Houlton et al., 2008). Symbiotic BNF by *Robinia* reaches its maximum at 31.9°C and is derived from (Bytnerowicz et al., in review)**

**2.4 N limitation, plant C allocation to growth, and plant C allocation to rhizosphere priming**

The non-structural C pool ($NSC$; [kg C indiv$^{-1}$]) gains C from photosynthesis. $NSC$ loses C to respiration and C allocation to growth, symbionts, and root C exudation. The rate of change of $NSC$ ($\frac{dNSC}{dt}$; [kg C indiv$^{-1}$ yr$^{-1}$]) is

$$\frac{dNSC}{dt} = P - R - \left(G_{C,l} + G_{C,r} + G_{C,sw} + G_{C,seed}\right) - C_{alloc} - L_{C,exudate} \qquad (5)$$

where $P$ is the photosynthesis rate [kg C indiv$^{-1}$ yr$^{-1}$], $R$ is the respiration rate (maintenance and growth) [kg C indiv$^{-1}$ yr$^{-1}$], $G_{C,l}$ is the growth rate of the leaf C pool ($C_l$; [kg C indiv$^{-1}$]) [kg C indiv$^{-1}$ yr$^{-1}$], $G_{C,r}$ is the growth rate of the fine root C pool ($C_r$; [kg C indiv$^{-1}$]) [kg C indiv$^{-1}$ yr$^{-1}$], $G_{C,sw}$ is the growth rate of the sapwood C pool ($C_{sw}$; [kg C indiv$^{-1}$]) [kg C indiv$^{-1}$ yr$^{-1}$], $G_{C,seed}$ is growth rate of the seed C pool ($C_{seed}$; [kg C indiv$^{-1}$]) [kg C indiv$^{-1}$ yr$^{-1}$], $C_{alloc}$ is the rate of C allocation to symbionts (Eq. (A43-46)), and $L_{C,exudate}$ is the rate of root C exudation (Eq. (A38)). Note that sapwood is converted to heartwood following Martinez Cano et al., (2020).

The non-structural N pool ($NSN$; [kg N indiv$^{-1}$]) gains N from N uptake via roots and symbionts. $NSN$ loses N to N allocation to growth, symbionts, and root N exudation. The rate of change of $NSN$ ($\frac{dNSN}{dt}$; [kg N indiv$^{-1}$ yr$^{-1}$]) is

$$\frac{dNSN}{dt} = U - \left(\frac{G_{C,l}}{C:N_l} + \frac{G_{C,r}}{C:N_r} + \frac{G_{C,sw}}{C:N_{sw}} + \frac{G_{C,seed}}{C:N_{seed}}\right) - N_{alloc} - L_{N,exudate} \qquad (6)$$

where $U$ is the N uptake rate via roots and symbionts (Eq. (A55)) [kg N indiv$^{-1}$ yr$^{-1}$], $C:N_l$ is the fixed C:N ratio of leaves, $C:N_r$ is the fixed C:N ratio of fine roots, $C:N_{sw}$ is the fixed C:N ratio of sapwood, $C:N_{seed}$ is the fixed C:N ratio of seeds, $N_{alloc}$ is the rate of N allocation to symbionts (Eq. (A47)), and $L_{N,exudate}$ is the rate of root N exudation (Eq. (A39)).

Non-N-limited growth is calculated according to Weng et al., (2015) and Martinez Cano et al., (2020). The total allocation of $NSC$ to growth is determined by the target $NSC$ ($NSC_{target}$; [kg C indiv$^{-1}$]) and minimizes the deviation between $NSC$ and $NSC_{target}$. $NSC_{target}$ is a multiple of the target $C_l$ ($C_{l,target}$; [kg C indiv$^{-1}$]), which reflects the ability of a plant to refoliate after defoliation (Hoch et al., 2003; Richardson et al., 2013), and is calculated as

$$NSC_{target} = q \, C_{l,target} \qquad (7)$$

where $q$ is a proportionality constant. The allocation of $NSC$ to the growth of each tissue depends on the total allocation of $NSC$ to growth and the target C pool of each tissue, and minimizes the deviation between the C pool of each tissue and the target C pool of each tissue. The target C pool of each tissue is dynamic and is determined by allometry (Eq. (A40-42)), canopy position, and phenology.

In LM4.1-BNF, $G_{C,l}$, $G_{C,r}$, $G_{C,sw}$, and $G_{C,seed}$ are adjusted to include N limitation and are calculated as

$$G_{C,l} = (1 - N_{stress})\Delta_l \qquad (8)$$

$$G_{C,r} = \Delta_r \qquad (9)$$

$$G_{C,sw} = (1 - N_{stress})\Delta_{sw} \qquad (10)$$

$$G_{C,seed} = (1 - N_{stress})\Delta_{seed} \qquad (11)$$

where $N_{stress}$ is N stress [unitless] and $\Delta_l, \Delta_r, \Delta_{sw}$, and $\Delta_{seed}$ are the non-N-limited growth rates of $C_l, C_r, C_{sw}$, and $C_{seed}$ respectively [kg C indiv$^{-1}$ yr$^{-1}$] following Weng et al., (2015) and Martinez Cano et al., (2020). Because plants increase C allocation to fine roots relative to other tissues when N-limited (Poorter et al., 2012), $G_{C,r}$ is not adjusted to include N limitation.

In LM4.1-BNF, $N_{stress}$ is the relative difference between $NSN$ and $NSN_{target}$ and is calculated as

$$N_{stress} = \max\left[0, \frac{NSN_{target} - NSN}{NSN_{target}}\right] \tag{12}$$

where $NSN_{target}$ is the target $NSN$ [kg N indiv$^{-1}$]. $N_{stress}$ is smoothed with a low-pass filter over 30 days to reflect the persisting influence of N stress (Mooney et al., 1991). $NSN_{target}$ is calculated as

$$NSN_{target} = \frac{NSC_{target}}{C:N_l} \tag{13}$$

This is similar to LM3-SNAP, which compared the target leaf and root N pools to $NSN$, but is modified to reflect the treatment of $NSC_{target}$ in LM4.1 by including the target sapwood and seed N pools.

Plant turnover decreases $C_l, C_r$, and $C_{sw}$ and from $N_l, N_r$, and $N_{sw}$ at a constant tissue-specific rate. A fraction of the turnover of $C_l$ and $N_l$ is retranslocated into $NSC$ and $NSN$ respectively.

Under N limitation, plants increase root C exudation to stimulate N mineralization in the rhizosphere (rhizosphere priming; Cheng et al., 2014; Finzi et al., 2015). $L_{C,exudate}$ increases with $N_{stress}$ and is calculated as

$$L_{C,exudate} = r_{leakage,C} \; NSC \; N_{stress} \tag{14}$$

where $r_{leakage,C}$ is a rate constant.

Under N limitation, plants decrease root N exudation (Canarini et al., 2019). $L_{N,exudate}$ decreases with $N_{stress}$ and is calculated as

$$L_{N,exudate} = r_{leakage,N} \; NSN \; (1 - N_{stress}) \tag{15}$$

where $r_{leakage,N}$ is a rate constant.

Under N limitation, plants increase active root uptake of inorganic N (Nacry et al., 2013). The rate of active root uptake of inorganic N in soil layer k ($N_{active}(k)$; [kg N indiv$^{-1}$ yr$^{-1}$]) increases with $N_{stress}$ and is calculated as

$$N_{active}(k) = f\big(NO_3(k), NH_4(k), f_{rhiz}(k)\big)N_{stress} \tag{16}$$

where $f\big(NO_3(k), NH_4(k), f_{rhiz}(k)\big)$ is a function of the $NH_4^+$ pool in soil layer k ($NH_4(k)$; [kg N m$^{-2}$]), the $NO_3^-$ pool in soil layer k ($NO_3(k)$; [kg N m$^{-2}$]), and the rhizosphere volume fraction of soil layer k ($f_{rhiz}(k)$; [m$^3$ m$^{-3}$]), and is given in Eq. (A3).

## 2.5 Plant C allocation to symbionts (AM, EM and N-fixing bacteria)

The rate of C allocation to AM ($C_{alloc,AM}$; [kg C indiv$^{-1}$ yr$^{-1}$]) is

$$C_{alloc,AM} = f_{alloc,AM} \; NSC \tag{17}$$

where $f_{alloc,AM}$ is the fraction of $NSC$ allocated to AM per unit time. $C_{alloc,AM}$ is not related to $N_{stress}$ because, although AM increase N uptake, AM is maintained by the plant primarily for phosphorus uptake (Smith and Smith, 2011).

The rate of C allocation to EM ($C_{alloc,EM}$; [kg C indiv$^{-1}$ yr$^{-1}$]) is

$$C_{alloc,EM} = f_{alloc,EM} \, NSC \, N_{stress} \tag{18}$$

where $f_{alloc,EM}$ is the maximum fraction of $NSC$ allocated to EM per unit time. $C_{alloc,EM}$ is a function of $N_{stress}$ because biomass C of EM increases with N limitation (Phillips et al., 2013).

Plants that associate with N-fixing bacteria can regulate symbiotic BNF to different extents, termed their BNF strategy (Menge et al., 2015). For plants with a perfectly facultative BNF strategy, symbiotic BNF increases with N limitation. For plants with an incomplete BNF strategy, symbiotic BNF increases with N limitation but is maintained at a minimum. For plants with an obligate BNF strategy, symbiotic BNF is constant. For plants with either a facultative or an incomplete BNF strategy, the rate of C allocation by the plant to N-fixing bacteria ($C_{alloc,Nfix}$; [kg C indiv$^{-1}$ yr$^{-1}$]) is

$$C_{alloc,Nfix} = \max[f_{alloc,Nfix} \, NSC \, N_{stress}, f_{alloc,Nfix,min} \, NSC] \tag{19}$$

where $f_{alloc,Nfix}$ is the fraction of $NSC$ allocated to N-fixing bacteria per unit time and $f_{alloc,Nfix,min}$ is the minimum fraction of $NSC$ allocated to N-fixing bacteria per unit time. For plants with a perfectly facultative BNF strategy $f_{alloc,Nfix,min} = 0$ and for plants with an incomplete down-regulator BNF strategy $f_{alloc,Nfix,min} > 0$. *Robinia* has an incomplete down-regulator BNF strategy.

For plants with an obligate BNF strategy, $C_{alloc,Nfix}$ is

$$C_{alloc,Nfix} = f_{alloc,N \, fix} \, NSC \tag{20}$$

Additionally, plants allocate a small quantity of N to symbionts such that symbiont growth can be initiated. The rate of N allocation by the plant to symbionts ($N_{alloc,j}$; j = AM, EM, N-fixing bacteria; [kg N indiv$^{-1}$ yr$^{-1}$]) is

$$N_{alloc,j} = \frac{C_{alloc,j}}{C:N_{alloc}} \tag{21}$$

where $C:N_{alloc}$ is the C:N ratio of C and N allocated to symbionts by the plant.

Plant C allocation to symbionts increases biomass C of symbionts, which increases plant N uptake via symbionts (Appendix A.1 and A.6).

## 2.6 Dynamic plant C allocation to growth and N uptake

The order of plant C allocation to growth, symbionts, and rhizosphere priming is determined by C limitation relative to N limitation (Cheng et al., 2014; Finzi et al., 2015; Poorter et al., 2012; Treseder, 2004; Zheng et al., 2019). If a plant is more C-limited than N-limited, $NSC < NSN \cdot C:N_{leaf}$, the plant allocates C to growth, then to N-fixing bacteria (if associated) and EM (if associated), then to rhizosphere priming, and finally to AM. If a plant is more N-limited than C-limited, $NSC > NSN \cdot C:N_{leaf}$, the plant allocates C to N-fixing bacteria (if associated) and EM (if associated), then to rhizosphere priming, then to growth, and finally to AM.

## 2.7 Soil N₂O and NO emissions

Soil N₂O and NO emissions occur during nitrification (aerobic oxidation of $NH_4^+$ with oxygen as an electron acceptor, which produces N₂O and NO as by-products) and denitrification (anaerobic oxidation of organic C with $NO_3^-$ as an electron acceptor, which produces N₂O as a by-product).

Following LM3V-N (Huang and Gerber, 2015), soil N₂O emission rate in soil layer k ($N_2O(k)$; [kg N m⁻² yr⁻¹]) is

$$N_2O(k) = 0.004\, nit(k) + \frac{1}{1+\max\left(3.52, 22\exp\left(-0.8\frac{NO_3(k)}{HR(k)}\right)\right)\max\left(0.1, 0.015\frac{\theta(k)}{\theta_{sat}}-0.32\right)} denit(k) \qquad (22)$$

where $nit(k)$ is nitrification rate in soil layer k [kg N m⁻² yr⁻¹] (Eq. (A56)), $HR(k)$ is heterotrophic respiration in soil layer k [kg C m⁻² yr⁻¹], $\theta(k)$ is volumetric soil water content of soil layer k [m³ m⁻³], $\theta_{sat}$ is saturation volumetric soil water content, and $denit(k)$ is denitrification rate in soil layer k [kg N m⁻² yr⁻¹] (Eq. (A57)).

Following LM3V-N (Huang and Gerber, 2015), soil NO emission rate in soil layer k ($NO(k)$; [kg N m⁻² yr⁻¹]) is

$$NO(k) = 0.004\left(15.2 + \frac{35.5\tan^{-1}\left(0.68\pi\left(2.09\left(1-\frac{\theta(k)}{\theta_{sat}}\right)^{4/3}-1.68\right)\right)}{\pi}\right)nit(k) \qquad (23)$$

## 2.8 Phenomenological representations of BNF (LM4.1-BNF_NPP and LM4.1-BNF_ET)

To compare mechanistic and phenomenological representations of BNF in our first analysis, we developed LM4.1-BNF_NPP and LM4.1-BNF_ET which have phenomenological representations of BNF. In LM4.1-BNF_NPP, BNF is represented as a function of NPP in LM4.1-BNF. BNF rate ($BNF_{NPP}$; [kg N m⁻²]) is

$$BNF_{NPP} = a_{NPP}(1 - e^{b_{NPP}NPP}) \qquad (24)$$

where $NPP$ is net primary production [kg C m⁻² yr⁻¹], and $a_{NPP}$ and $b_{NPP}$ are constants from Meyerholt et al., 2016.

In LM4.1-BNF_ET, BNF is represented as a function of ET in LM4.1-BNF. BNF rate ($BNF_{ET}$; [kg N m⁻²]) is

$$BNF_{ET} = a_{ET}ET + b_{ET} \qquad (25)$$

where $ET$ is evapotranspiration [mm yr⁻¹] and $a_{ET}$ and $b_{ET}$ are constants from Meyerholt et al., 2016.

$BNF_{NPP}$ and $BNF_{ET}$ enter $NH_4(k)$ (distributed across all soil layers proportional to thickness). In LM4.1-BNF_NPP and LM4.1-BNF_ET, growth and turnover of symbionts and plant C allocation to symbionts do not occur, and asymbiotic BNF does not occur. All other components of C and N cycling in LM4.1-BNF_NPP and LM4.1-BNF_ET are the same as LM4.1-BNF.

## 3 Numerical experiments and evaluation description

### 3.1 Numerical experiments description

We ran numerical experiments for the grid cell containing Coweeta Hydrological Laboratory (CHL) in North Carolina, US (35.05˚N, 83.45˚W), which is part of the Long-Term Ecological Research Network and has observations on symbiotic BNF by *Robinia* (Boring and Swank, 1984).

We ran the LM4.1-BNF spin up for 1000 years at pre-industrial atmospheric $CO_2$ concentration (284.26 ppm) to allow the soil C and N pools to reach an approximate steady state. Then, we initialised LM4.1-BNF, LM4.1-BNF$_{NPP}$, and LM4.1-BNF$_{ET}$ numerical experiments with seedlings (removed vegetation C and N pools from the LM4.1-BNF spin up) and the LM4.1-BNF spin up soil C and N pools. We ran numerical experiments for another 300 years at current atmospheric $CO_2$ concentration (324.53 ppm). See Table D3 for a summary of atmospheric $CO_2$ concentration (Dlugokencky and Tans, 2020; Meinshausen et al., 2017), meteorological forcings (Sheffield et al., 2006), and N deposition rates (Dentener, 2006) used in the spin up and numerical experiments. We initialised the LM4.1-BNF spin up with *Acer* seedlings, and we initialised LM4.1-BNF numerical experiments with both *Acer* and *Robinia* seedlings.

To compare mechanistic and phenomenological representations of BNF in our first analysis (Table 2), we initialised LM4.1-BNF$_{NPP}$ and LM4.1-BNF$_{ET}$ numerical experiments with only *Acer* seedlings. To examine the role of competitive dynamics between N-fixing and non-fixing plants in our second analysis (Table 2), we initialised LM4.1-BNF numerical experiments with both *Acer* and *Robinia* seedlings, only *Acer* seedlings or only *Acer* seedlings that can associate with N-fixing bacteria (N-fixer *Acer*). To examine the role of asymbiotic BNF in our third analysis (Table 2), we initialised LM4.1-BNF numerical experiments with both *Acer* and *Robinia* seedlings, only *Acer* seedlings, or only N-fixer *Acer* seedlings. LM4.1-BNF, LM4.1-BNF$_{NPP}$, and LM4.1-BNF$_{ET}$ simulations are initialised such that all plant cohorts have the same height (0.5 m; Table D4) and dbh is determined from height by allometry (Eq. (A41); Table D4). See Table D4 for a summary of initial density, height, and dbh of seedlings in numerical experiments.

We ran the LM3-SNAP spin up for 1000 years at pre-industrial atmospheric $CO_2$ concentration to allow the soil C and N pools to reach an approximate steady state. Then, we initialised LM3-SNAP numerical experiments with seedlings and the LM3-SNAP spin up soil C and N pools. We ran numerical experiments for another 300 years at current atmospheric $CO_2$ concentration. See Table D3 for a summary of atmospheric $CO_2$ concentration, meteorological forcings, and N deposition rates used in the spin up and numerical experiments. We initialised the LM3-SNAP spin up and the LM3-SNAP numerical experiments with a temperate deciduous vegetation type.

**Table 2: Description of numerical experiments. In the first analysis, we compare mechanistic and phenomenological representations of BNF. In the second analysis, we examine the role of competitive dynamics between N-fixing and non-fixing plants. In the third analysis, we examine the role of asymbiotic BNF. Note that the same six numerical experiments were examined in both the second and third analyses.**

| Analysis | BNF representation | Species | Asymbiotic BNF |
|---|---|---|---|

| 1. Mechanistic and phenomenological | LM4.1-BNF | *Robinia* and *Acer* | represented |
|---|---|---|---|
| representations of BNF | LM4.1-BNF$_{NPP}$ | *Robinia* and *Acer* | - |
| | LM4.1-BNF$_{ET}$ | *Robinia* and *Acer* | - |
| | LM4.1-BNF | *Robinia* and *Acer* | represented |
| | LM4.1-BNF | *Robinia* and *Acer* | - |
| 2. Competitive dynamics between | LM4.1-BNF | *Acer* | represented |
| N-fixing and non-fixing plants | LM4.1-BNF | *Acer* | - |
| 3. Asymbiotic BNF | LM4.1-BNF | N-fixer *Acer* | represented |
| | LM4.1-BNF | N-fixer *Acer* | - |

## 3.2 Evaluation description

To evaluate the performance of LM4.1-BNF in representing symbiotic BNF, we compared symbiotic BNF rate from the numerical experiments of LM4.1-BNF and LM3-SNAP to site observations from CHL (Fig. 3a; Boring and Swank, 1984). To evaluate the performance of LM4.1-BNF in representing asymbiotic BNF, we compared asymbiotic BNF rate from the numerical experiments of LM4.1-BNF to site observations from CHL (Fig. 3b; Todd et al., 1978). To evaluate the performance of LM4.1-BNF in representing the competitive dynamics between N-fixing and non-fixing plants, we compared the basal area fraction of each vegetation type (*Acer* and *Robinia*) over time from the numerical experiments of LM4.1-BNF to US FIA database tree data (Fig. 4; US Forest Service, 2020a).

To evaluate the performance of LM4.1-BNF in representing ecosystem C and N pools and fluxes, we (1) compared total plant biomass C from the numerical experiments of LM4.1-BNF and LM3-SNAP to US FIA database tree data (Fig. 5; US Forest Service, 2020a), (2) compared soil C and N pools and fluxes from the numerical experiments of LM4.1-BNF and LM3-SNAP to US FIA database soil data (Fig. 6; US Forest Service, 2020a), site observations from CHL (Fig. 6; Knoepp, 2009a, 2009b, 2018; Swank and Waide, 1988), and a meta-analysis of temperate forests (Fig. 6; Stehfest and Bouwman, 2006), (3) compared ecosystem C fluxes from the numerical experiments of LM4.1-BNF and LM3-SNAP to eddy covariance observations from CHL at the hourly timescale (Fig. 7; Oishi, 2020), and (4) compared ecosystem C fluxes from the numerical experiments of LM4.1-BNF and LM3-SNAP to a meta-analysis of temperate forests (Fig. 8; Anderson-Teixeira et al., 2018). We also compared dbh growth rates and the dbh distribution from the numerical experiments of LM4.1-BNF to US FIA database tree data (Fig. D1-2; US Forest Service, 2020a). See Appendix B for data availability and processing. See Table D5 for a summary of the validated variables and data sources.

## 4 Evaluation: comparison of LM4.1-BNF to observations and to LM3-SNAP

## 4.1 Evaluation results

Here we describe the evaluation of LM4.1-BNF, in which we compare LM4.1-BNF simulations to observations and to LM3-SNAP simulations (which represents a single general plant C pool capable of BNF and cannot represent community

dynamics (i.e., competitive dynamics between N-fixing and non-fixing plants) and does not represent asymbiotic BNF). LM4.1-BNF captures observed symbiotic BNF, asymbiotic BNF, successional dynamics, and the major pools and fluxes of

C and N.

LM4.1-BNF effectively reproduces asymbiotic BNF rate (mean 13 kg N ha$^{-1}$ yr$^{-1}$ over the final 100 years), which is not represented in LM3-SNAP (Fig. 3a), in comparison to observations from CHL (11 kg N ha$^{-1}$ yr$^{-1}$). LM4.1-BNF effectively reproduces the temporal dynamics of symbiotic BNF rate: LM4.1-BNF simulates a symbiotic BNF rate pulse in early succession that reaches 73 kg N ha$^{-1}$ yr$^{-1}$ at 15 years, then declines to ~ 0 kg N ha$^{-1}$ yr$^{-1}$ at 300 years (Fig. 3b). LM3-SNAP

simulates high symbiotic BNF rate in late succession (mean 8 kg N ha$^{-1}$ yr$^{-1}$ over the final 100 years; Fig. 3b). Observations from CHL suggest that symbiotic BNF which reaches 75 kg N ha$^{-1}$ yr$^{-1}$ at 17 years then declines to ~ 0 kg N ha$^{-1}$ yr$^{-1}$ in late successional forests. Note that asymbiotic BNF is directly controlled by soil microbe biomass C (Eq. (3)) and peripherally controlled by soil microbe biomass N, total soil C, and total soil N (Fig. D3) and symbiotic BNF is directly controlled by nodule biomass C (Eq. (1)) and peripherally controlled by non-structural C and N stress (Fig. D4).

LM4.1-BNF effectively reproduces the successional dynamics of *Robinia* and *Acer* (Fig. 4): *Robinia* is competitively excluded by *Acer* at approximately the same time scale as observations (~ 5 % basal area fraction at 150 years).

At the ecosystem scale, LM4.1-BNF effectively reproduces the temporal dynamics of total plant biomass C in comparison to observations (mean 173 vs. 160 Mg C ha$^{-1}$ over the final 100 years; Fig. 5), whereas LM3-SNAP substantially underestimates total plant biomass C in comparison to observations (mean 44 vs. 160 Mg C ha$^{-1}$ over the final 100 years;

Fig. 5). LM4.1-BNF makes reasonable estimates for soil C and N pools and fluxes, which are comparable to those of LM3-SNAP. LM4.1-BNF underestimates total soil C and N (mean 13.4 vs. 33.0 Mg C ha$^{-1}$ and 0.2 vs. 1.8 Mg N ha$^{-1}$ respectively; Fig. 6ab) and overestimates soil NH$_4^+$ and NO$_3^-$ (mean 4.7 vs. 1.0 kg N ha$^{-1}$ and 1.6 vs. 0.1 kg N ha$^{-1}$ respectively; Fig. 6c) in comparison to observations. LM4.1-BNF underestimates N mineralization rate and net nitrification rate in comparison to observations (mean 14.9 vs. 39.5 kg N ha$^{-1}$ yr$^{-1}$ and 7.2 vs. 12.8 kg N ha$^{-1}$ yr$^{-1}$ respectively; Fig. 6d). LM4.1-BNF

overestimates N$_2$O emission rate (mean 5.1 vs. 0.9 kg N ha$^{-1}$ yr$^{-1}$) and underestimates NO emission rate (mean 0.2 vs. 0.8 kg N ha$^{-1}$ yr$^{-1}$; Fig. 6e) in comparison to observations. Note that LM3-SNAP cannot simulate N$_2$O and NO emission rates. LM4.1-BNF underestimates dissolved organic N (DON) leaching rate (mean ~ 0 vs. 0.6 kg N ha$^{-1}$ yr$^{-1}$) and NH$_4^+$ leaching rate (mean ~0 vs. 0.05 kg N ha$^{-1}$ yr$^{-1}$) in comparison to observations, and reasonably estimates NO$_3^-$ leaching rate (mean 0.1 vs. 0.1 kg N ha$^{-1}$ yr$^{-1}$; Fig. 6e) in comparison to observations. LM3-SNAP substantially overestimates NO$_3^-$ leaching rate in

comparison to observations (mean 8.3 vs. 0.1 kg N ha$^{-1}$ yr$^{-1}$; Fig. 6e).

Finally, LM4.1-BNF makes reasonable estimates for ecosystem C fluxes, particularly in comparison to LM3-SNAP. LM4.1-BNF effectively reproduces net ecosystem production (NEP) in both the growing and non-growing seasons at the hourly timescale, especially in comparison to LM3-SNAP which overestimates NEP in the non-growing season (Fig. 7). LM4.1-BNF effectively reproduces gross primary production (GPP) and NPP, especially in comparison to LM3-SNAP which

substantially overestimates both GPP and NPP (Fig. 8). LM4.1-BNF overestimates GPP and NPP in comparison to observations (mean 15.4 vs. 13.1 Mg C ha$^{-1}$ yr$^{-1}$ and 8.1 vs. 7.5 Mg C ha$^{-1}$ yr$^{-1}$ respectively). LM4.1-BNF overestimates HR

(mean 7.5 vs. 4.7 Mg C ha⁻¹ yr⁻¹) and consequentially underestimates net ecosystem production (NEP) (mean 0.6 vs. 4.8 Mg C ha⁻¹ yr⁻¹) in comparison to observations.

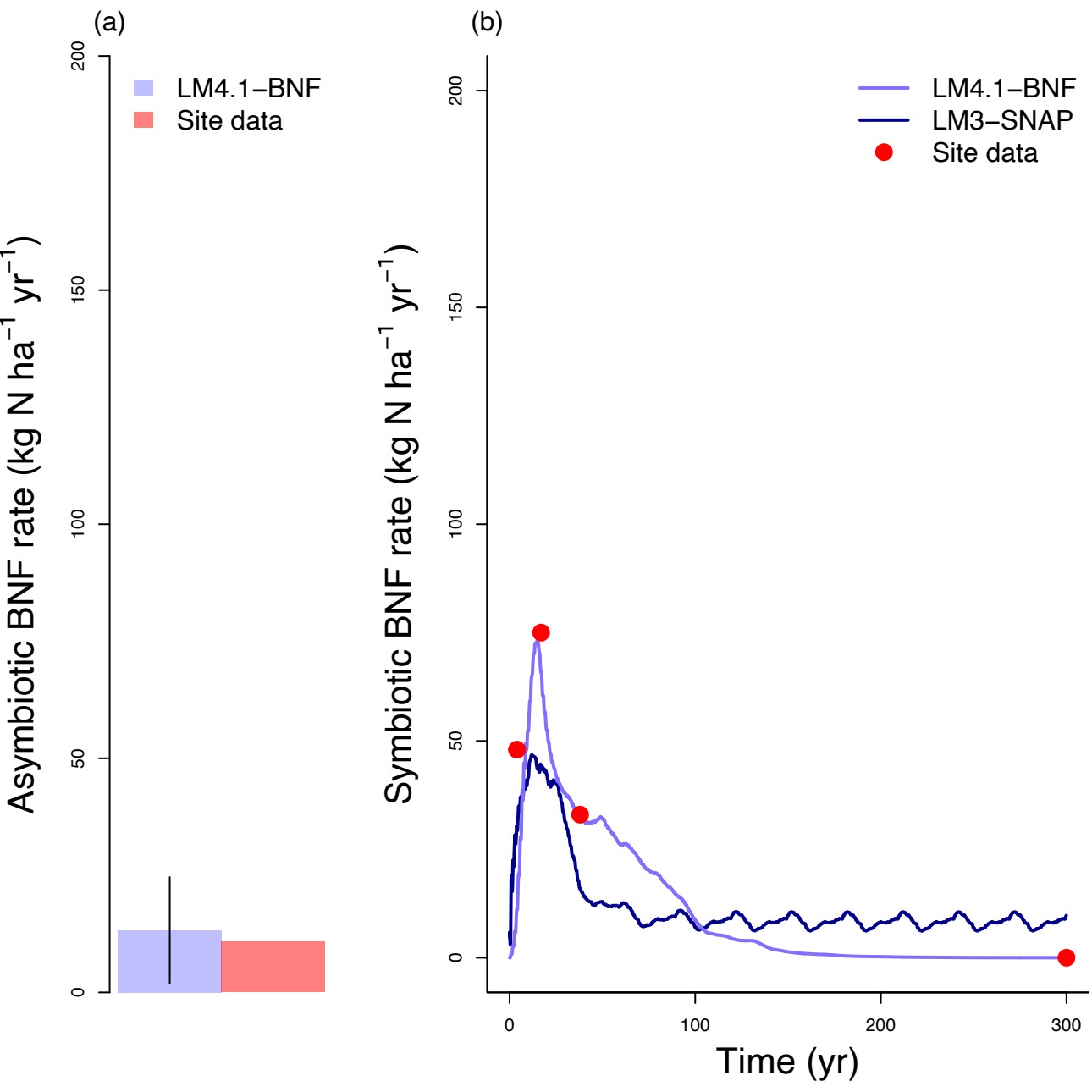

 **Figure 3: Simulated BNF rate by LM4.1-BNF and LM3-SNAP. (a) Simulated asymbiotic BNF rate compared to CHL site data. Simulated data are averaged over the last 100 yr of the 300 yr simulation to reflect the site data which is from mature forests. Error bars indicate two standard deviations. (b) Simulated symbiotic BNF rate over time compared to CHL site data for a 4 yr, 17 yr, 38 yr and mature forest (plotted at 300 yr).**

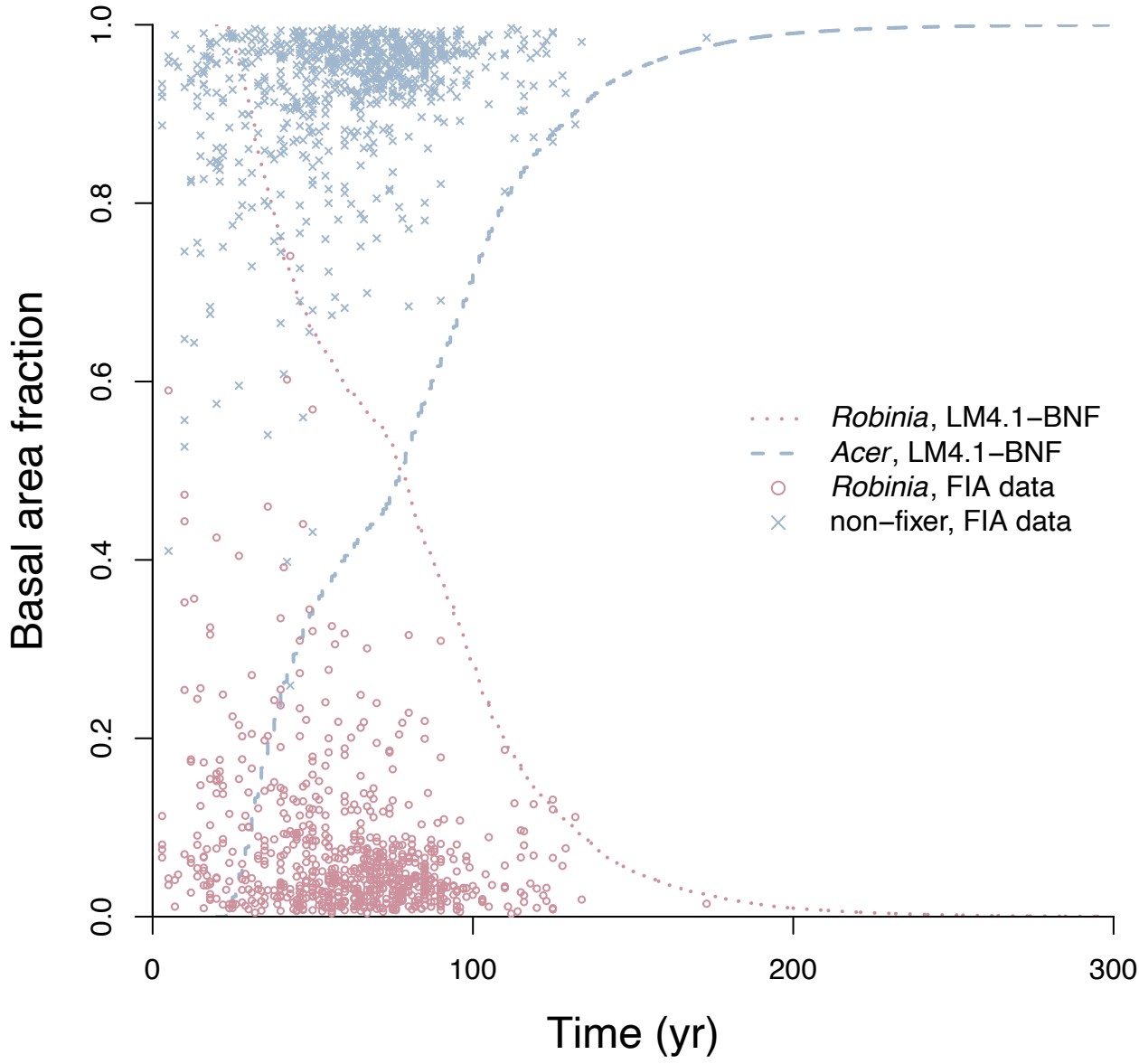

**Figure 4: Simulated relative basal area of *Acer* and *Robinia* over time compared to FIA data (in North Carolina). Simulated data are trees with dbh > 12.7 cm to reflect the dbh range of FIA data. FIA data of all non-fixing trees are aggregated to represent *Acer*. Each point represents an FIA plot. See Fig. D9 for absolute basal area.**

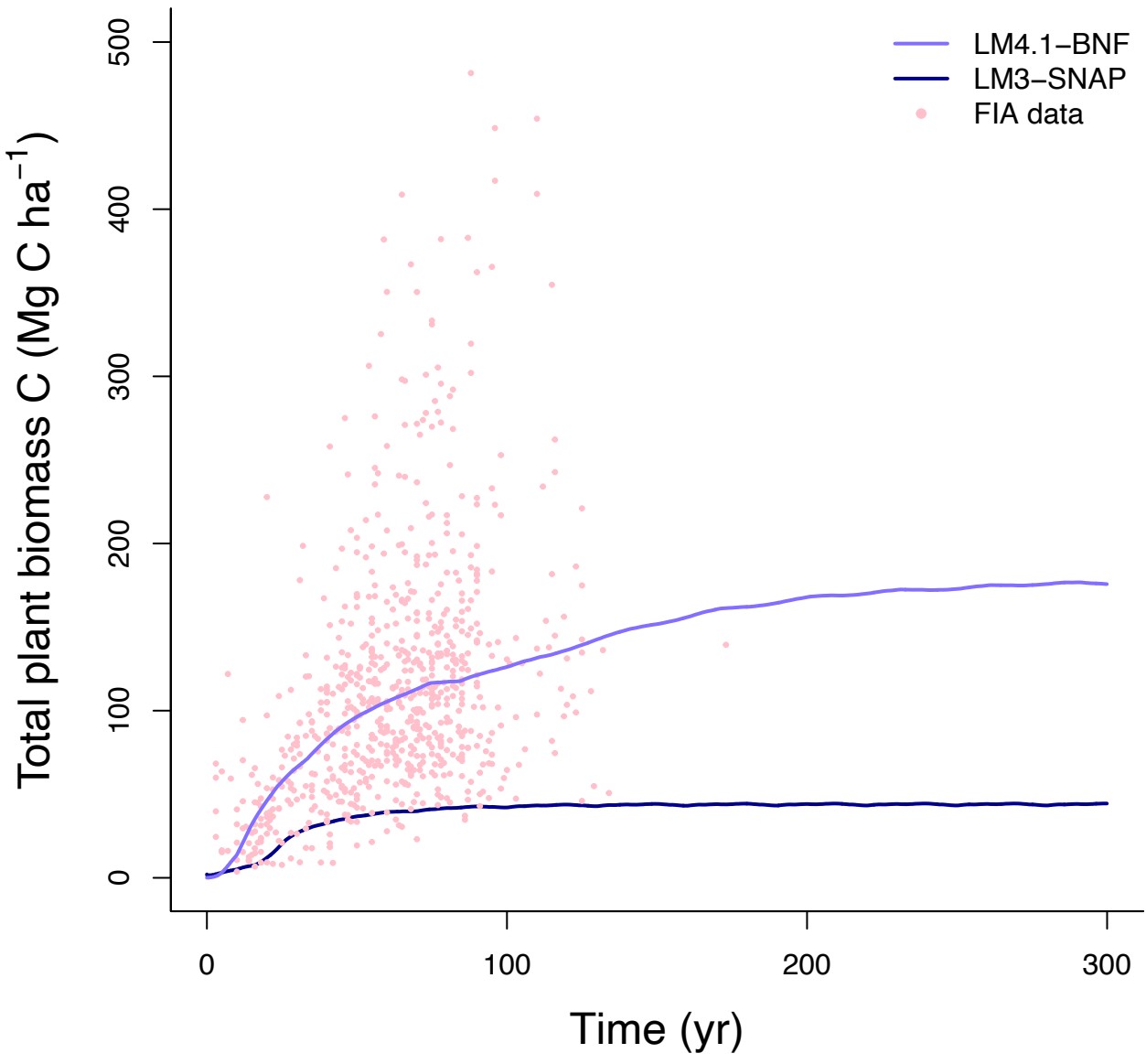

**Figure 5: Simulated total plant biomass C over time by LM4.1-BNF and LM3-SNAP compared to FIA data (in North Carolina).**
**Each point represents an FIA plot.**

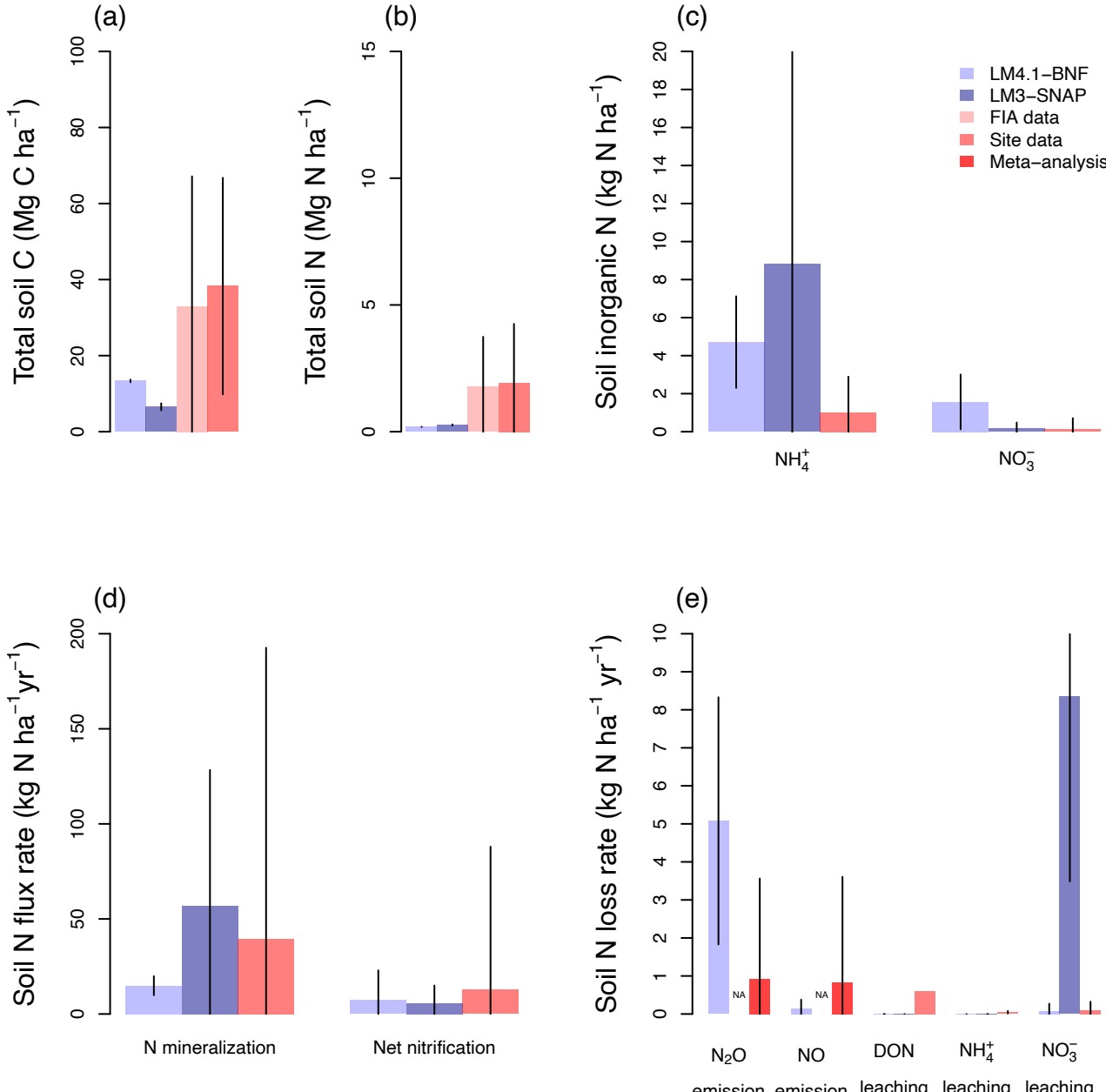

**Figure 6: Simulated soil C and N pools, soil N fluxes, and soil N loss rates by LM4.1-BNF and LM3-SNAP. (a) Simulated total soil C (depth 0-10 cm) compared to FIA data (in North Carolina) and CHL site data. (b) Simulated total soil N (depth 0-10 cm) compared to FIA data (in North Carolina) and CHL site data. (c) Simulated soil $NH_4^+$ and $NO_3^-$ (depth 0-10 cm) compared to CHL site data. (d) Simulated N mineralization rate and net nitrification rate (depth 0-10 cm) compared to CHL site data. (e) Simulated $N_2O$ and NO emission rates compared to a meta-analysis estimate for temperate forests, and simulated dissolved organic N (DON), $NH_4^+$, and $NO_3^-$ leaching rate compared to CHL site data. Simulated data are averaged over the last 100 yr of the 300 yr**

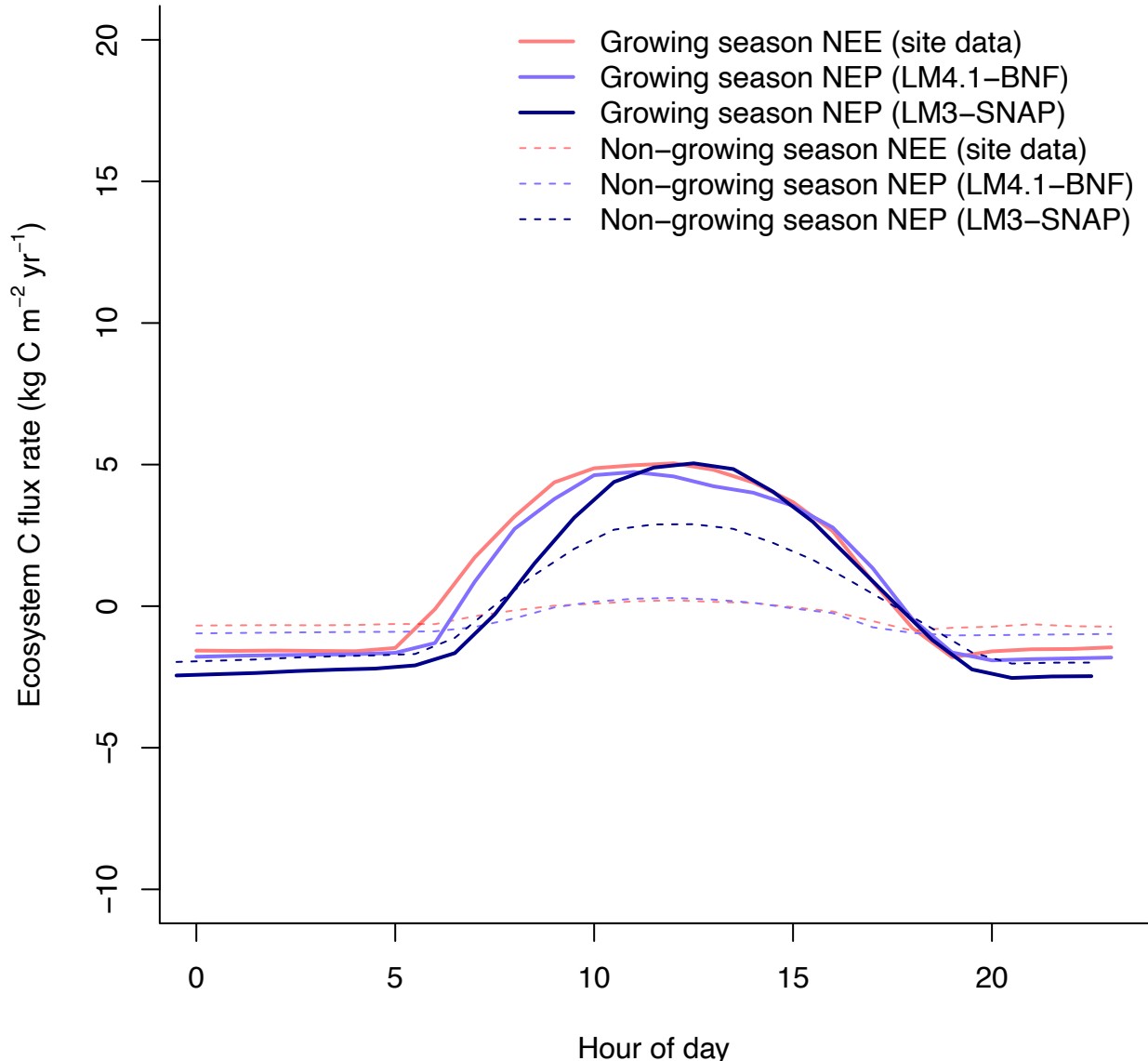


**Figure 7: Simulated net ecosystem production (NEP) by LM4.1-BNF and LM3-SNAP compared to net ecosystem exchange (NEE) CHL site data at the hourly timescale. Simulated data are averaged over the last 100 yr of the 300 yr simulation to reflect the data which is from mature forests.**

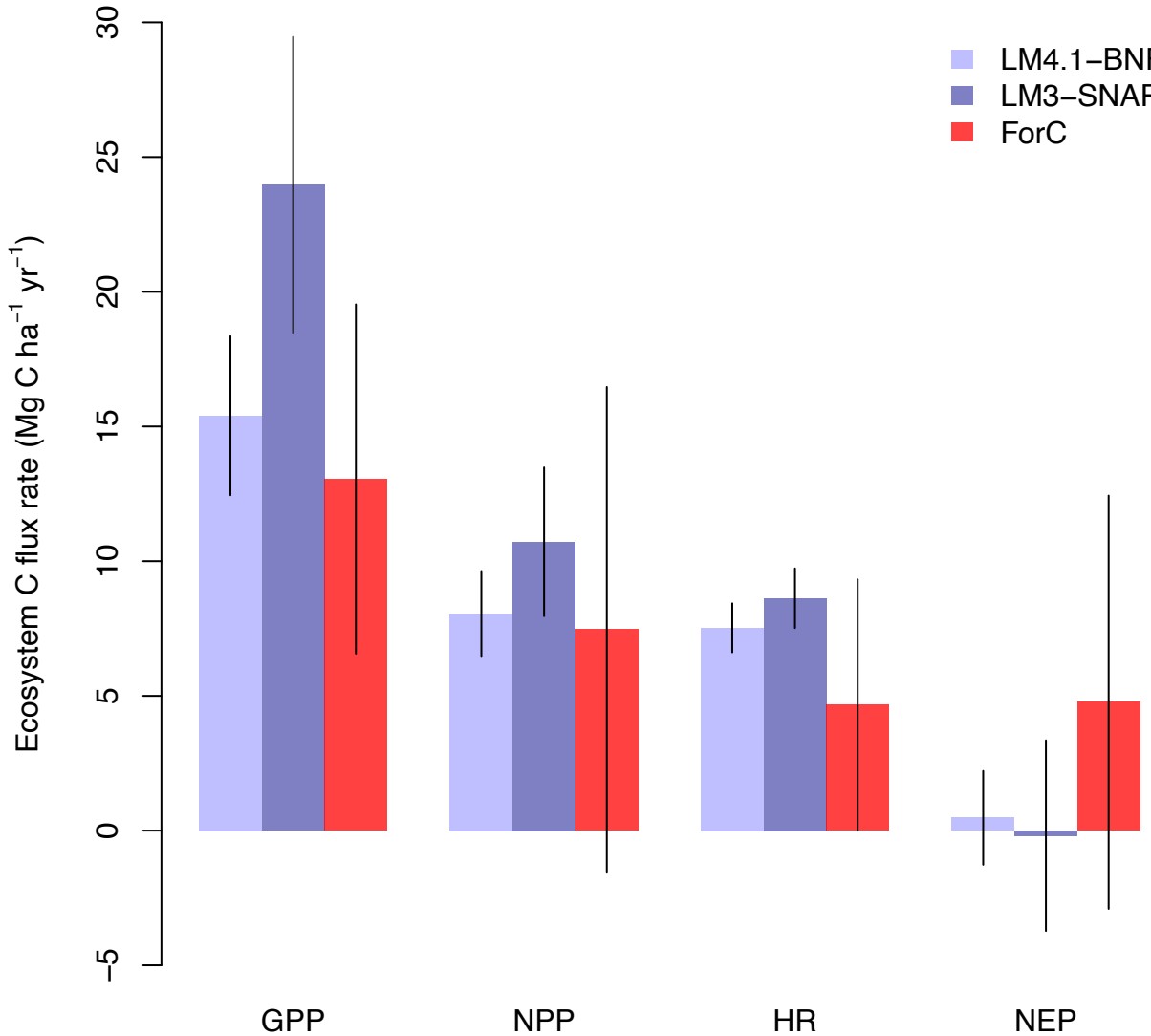

Figure 8: Simulated gross primary production (GPP), net primary production (NPP), heterotrophic respiration (HR), and net ecosystem production (NEP) by LM4.1-BNF and LM3-SNAP compared to the ForC database. Simulated data are averaged over the last 100 yr of the 300 yr simulation to reflect the data which is from mature forests. Error bars indicate two standard deviations.

**4.2 Evaluation discussion**

LM4.1-BNF effectively reproduces asymbiotic BNF and symbiotic BNF, i.e., its peak in early succession followed by its decline to zero in late succession (Fig. 3). In LM3-SNAP, asymbiotic BNF is not represented and symbiotic BNF is sustained in late succession (Fig. 3). This occurs because there is no competitive exclusion of N-fixing plants by non-fixing plants in LM3-SNAP, which represents a single general plant C pool capable of BNF and cannot represent community dynamics (Fig. 4). In observed ecosystems, N-fixing plants are competitively excluded by non-fixing plants due to weak N

limitation of plant growth in late succession (Menge et al., 2010; Sheffer et al., 2015) and symbiotic BNF is effectively zero (Boring and Swank, 1984).

This overestimation of symbiotic BNF in late succession by LM3-SNAP causes several problems. First, LM3-SNAP simulates lower total plant biomass C than LM4.1-BNF (mean 44 vs. 173 Mg C ha$^{-1}$ over the final 100 years; Fig. 5) because of the sustained high C cost of symbiotic BNF. Second, LM3-SNAP overestimates N losses in comparison to LM4.1-BNF,

especially $NO_3^-$ leaching rate (mean 8.3 vs. 0.1 kg N ha$^{-1}$ yr$^{-1}$ over the final 100 years; Fig. 6). Accurately estimating N leaching rates is important given its downstream consequences such as eutrophication and acidification (Fowler et al., 2013; Tian and Niu, 2015).

**5 LM4.1.-BNF performance relative to previous BNF representations**

Here we describe the three analyses we conducted to identify LM4.1-BNF improvements to estimating major C and N pools

and fluxes. First, we compare LM4.1-BNF to LM4.1-BNF with BNF represented as a function of NPP (LM4.1-BNF$_{NPP}$) and LM4.1-BNF with BNF represented as a function of ET (LM4.1-BNF$_{ET}$) to compare mechanistic and phenomenological representations of BNF. Second, we compare LM4.1-BNF simulations with both *Robinia* and *Acer* to LM4.1-BNF simulations with only *Acer* and LM4.1-BNF simulations with only *Acer* that can associate with N-fixing bacteria (hereafter, N-fixer *Acer*) to examine the role of competitive dynamics between N-fixing and non-fixing plants. Third, we compare

LM4.1-BNF simulations with asymbiotic BNF to LM4.1-BNF simulations without asymbiotic BNF to examine the role of asymbiotic BNF.

**5.1 Mechanistic and phenomenological representations of BNF**

In our first analysis (Table 2), we compare mechanistic and phenomenological representations of BNF, and their implications for C and N cycling. LM4.1-BNF, LM4.1-BNF$_{NPP}$, and LM4.1-BNF$_{ET}$ simulations estimate different total plant

biomass C (Fig. 9a). LM4.1-BNF predicts the largest total plant biomass C (mean 170 Mg C ha$^{-1}$ over the final 100 years), followed by LM4.1-BNF$_{ET}$ (mean 70 Mg C ha$^{-1}$ over the final 100 years) and LM4.1-BNF$_{NPP}$ (mean 0 Mg C ha$^{-1}$ over the final 100 years). This is because, in LM4.1-BNF, BNF responds dynamically to strong N limitation of plant growth in early succession and BNF (mean 36 kg N ha$^{-1}$ yr$^{-1}$ over the initial 100 years) supports total plant biomass C accumulation. Conversely, in LM4.1-BNF$_{NPP}$ and LM4.1-BNF$_{ET}$, BNF does not respond dynamically to strong N limitation of plant growth

in early succession and BNF is not sufficient (mean 22 and ~ 0 kg N ha$^{-1}$ yr$^{-1}$ over the initial 100 years for LM4.1-BNF$_{ET}$ and LM4.1-BNF$_{NPP}$ respectively) to support total plant biomass C accumulation (Fig. 9b). As such, LM4.1-BNF effectively reproduces the temporal dynamics of symbiotic BNF rate, whereas LM4.1-BNF$_{ET}$ and LM4.1-BNF$_{NPP}$ predicted relatively constant symbiotic BNF rates. In observed ecosystems, strong N limitation of plant growth occurs in early succession. N-fixing trees are generally important pioneer species and can relieve strong N limitation of plant growth in early succession

(Chapin III et al., 1994; Cierjacks et al., 2013; Menge et al., 2010). Consequently, symbiotic BNF is highest in early succession (Batterman et al., 2013; Boring and Swank, 1984; Menge and Hedin, 2009; Sullivan et al., 2014). Simulated soil C and N pools, soil N fluxes, soil N loss rates, and ecosystem C fluxes are relatively similar between simulations and are displayed in Fig. D5-6.

A similar result was found by Meyerholt et al., (2020), who compared five alternative representations of BNF within the O-

CN model, including a BNF representation based on NPP, a BNF representation based on ET, and a BNF representation responding dynamically to N limitation of plant growth. As with our results, they found that the BNF representation responding dynamically to N limitation of plant growth predicted the largest total plant biomass C. However, their study did not compare these results to simulations that include competitive dynamics between N-fixing and non-fixing plants because O-CN represents a single general plant C pool capable of BNF and cannot represent community dynamics.

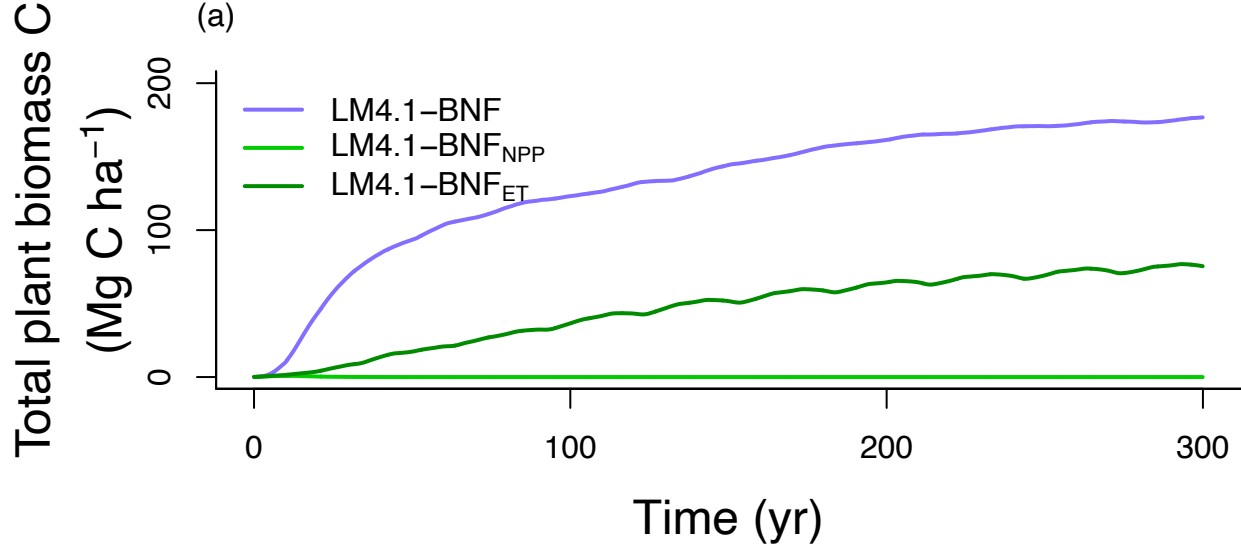

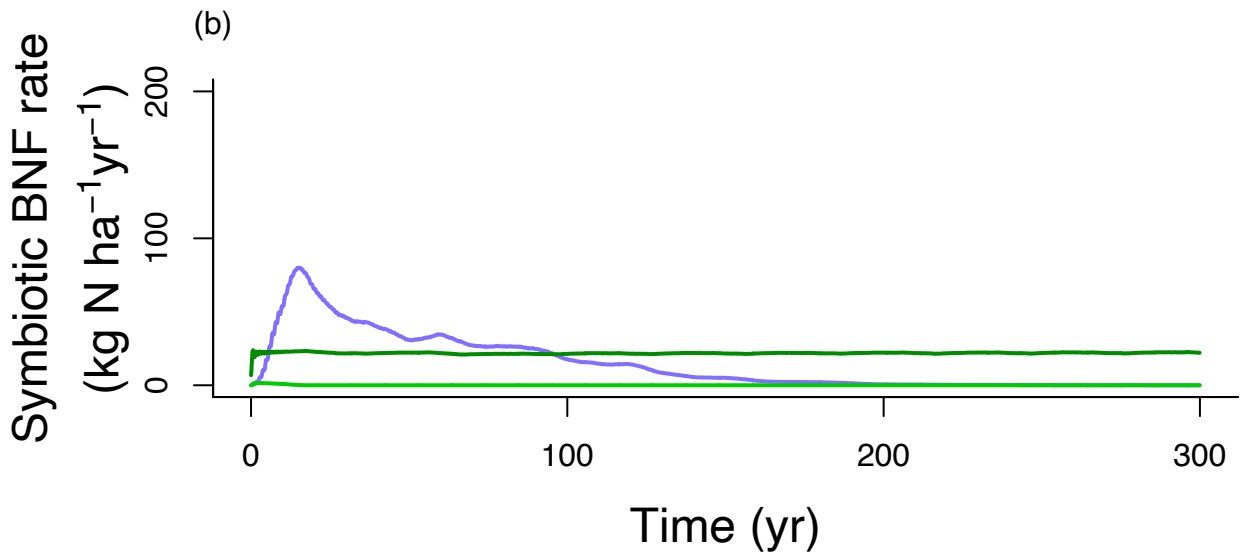


**Figure 9: Simulated (a) total plant biomass C and (b) symbiotic BNF rate over time from LM4.1-BNF, LM4.1-BNF$_{NPP}$, and LM4.1-BNF$_{ET}$.**

## 5.2 Competitive dynamics between N-fixing and non-fixing plants

In our second analysis (Table 2), we examine the role of competitive dynamics between N-fixing and non-fixing plants and its implication for C and N cycling. LM4.1-BNF simulations initialised with only *Acer* seedlings accumulate total plant biomass C slower than LM4.1-BNF simulations initialised with both *Robinia* and *Acer* seedlings (mean 48.6 vs. 80.9 Mg C ha$^{-1}$ over the initial 100 years respectively; Fig. 10a). In LM4.1-BNF simulations initialised with only *Acer* seedlings, stronger N limitation of plant growth in early succession due to the absence of a N-fixer vegetation type slows total plant biomass C accumulation. Nevertheless, total plant biomass C accumulates due to asymbiotic BNF and high N deposition at CHL (13.9 kg N ha$^{-1}$ yr$^{-1}$), reaching a similar level to LM4.1-BNF simulations initialised with both *Robinia* and *Acer* seedlings after 100 years.

LM4.1-BNF simulations initialised with only N-fixer *Acer* seedlings accumulate total plant biomass C similarly to LM4.1-BNF simulations initialised with both *Robinia* and *Acer* seedlings (mean 86.2 vs. 80.9 Mg C ha$^{-1}$ over the initial 100 years respectively; Fig. 10a). However, in LM4.1-BNF simulations initialised with only N-fixer *Acer* seedlings, a higher symbiotic BNF rate persists throughout succession in comparison to LM4.1-BNF simulations initialised with both *Robinia* and *Acer* seedlings (mean 23.6 vs. 0.2 kg N ha$^{-1}$ yr$^{-1}$ over the final 100 years respectively; Fig. 10b). This occurs because there is no competitive exclusion of N-fixing plants by non-fixing plants due to weak N limitation of plant growth in late succession, which occurs in LM4.1-BNF simulations initialised with both *Robinia* and *Acer* seedlings. Simulated soil C and N pools, soil N fluxes, soil N loss rates, and ecosystem C fluxes are relatively similar between simulations and are displayed in Fig. D7-8.

Levy-Varon et al., (2019) conducted a similar study, in which a N-fixer vegetation type was included in the ED2 model. Similarly, they found that simulations without a N-fixer vegetation type accumulate total plant biomass C slower than simulations with a N-fixer vegetation type. However, ED2 differs from LM4.1-BNF in a multitude of processes. In particular, ED2 does not include representations of asymbiotic BNF, mycorrhizae, or rhizosphere priming. Furthermore, the representation of BNF in ED2 assumes instantaneous down-regulation of symbiotic BNF rate in comparison to the time lag in down-regulation of symbiotic BNF rate in LM4.1-BNF (due to the time between plant C allocation to symbiotic BNF, the growth of N-fixing bacteria, and symbiotic BNF) following observations (Bytnerowicz et al., in review).

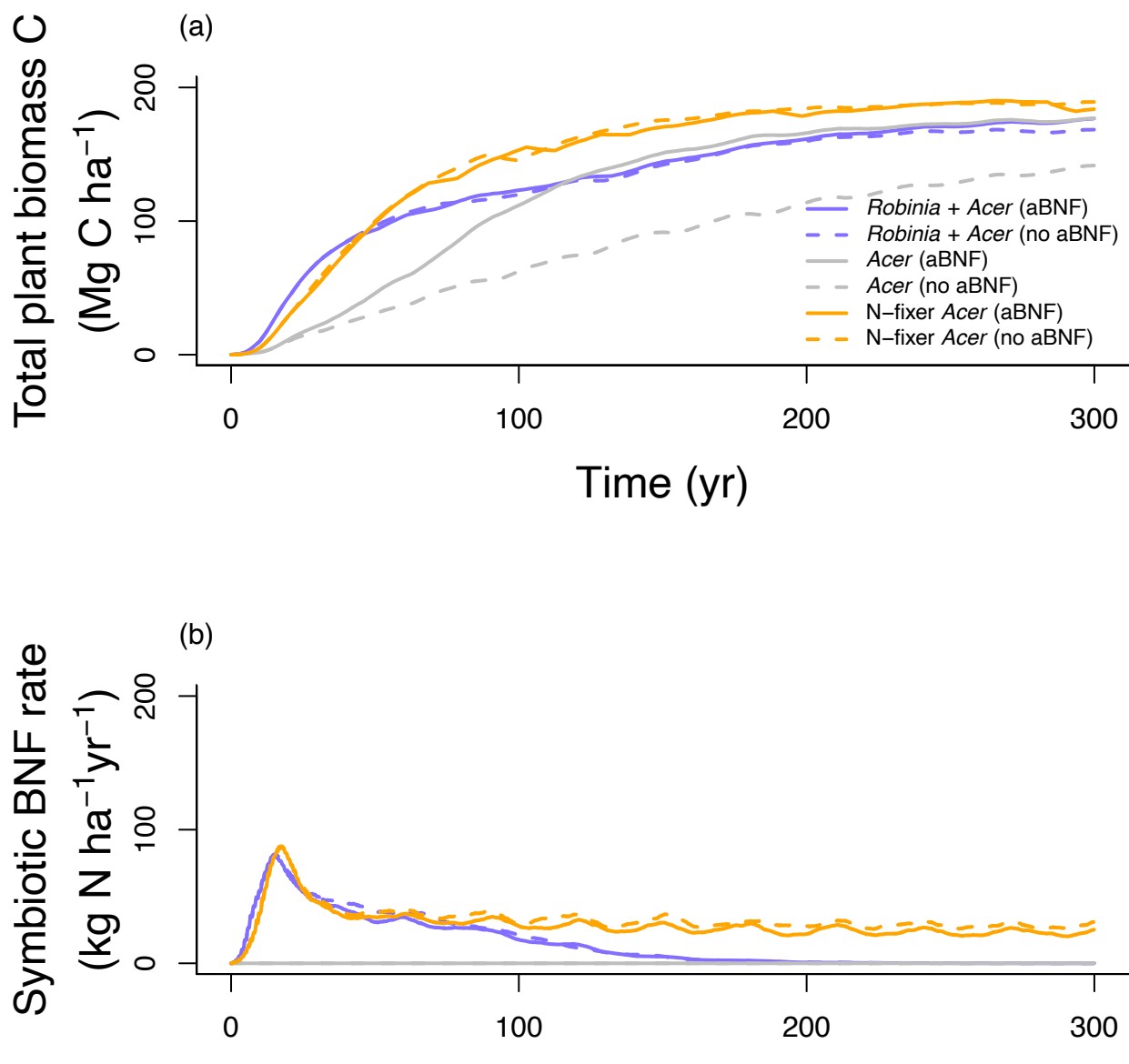

**Figure 10: Simulated (a) total plant biomass C and (b) symbiotic BNF rate over time from LM4.1-BNF initialised with both *Robinia* and *Acer*, only *Acer*, and only N-fixer *Acer*, with and without asymbiotic BNF. aBNF indicates asymbiotic BNF.**

## 5.3 Asymbiotic BNF

In our third analysis (Table 2), we examine the role of asymbiotic BNF and its implications for C and N cycling. LM4.1-BNF simulations initialised with *Acer* without asymbiotic BNF accumulate total plant biomass C slower than LM4.1-BNF simulations initialised with *Acer* with asymbiotic BNF (30.4 vs. 48.6 Mg C ha$^{-1}$ over the initial 100 years; Fig. 10a). In LM4.1-BNF simulations initialised with *Acer* without asymbiotic BNF, stronger N limitation of plant growth in early succession due to the absence of both asymbiotic BNF and a N-fixer vegetation type (i.e., symbiotic BNF) substantially slows total plant biomass C accumulation. Nevertheless, total plant biomass C accumulates due to high N deposition at CHL (13.9 kg N ha$^{-1}$ yr$^{-1}$), reaching a similar level to LM4.1-BNF simulations initialised with both *Robinia* and *Acer* seedlings and asymbiotic BNF after 300 years. Simulated soil C and N pools, soil N fluxes, soil N loss rates, and ecosystem C fluxes are relatively similar between simulations and are displayed in Fig. D7-8.

## 6 Discussion

### 6.1 Limitations

LM4.1-BNF captures the major pools and fluxes of C and N and their temporal dynamics. Importantly, LM4.1-BNF is novel in that it captures both the competitive dynamics between N-fixing and non-fixing plants as well as asymbiotic BNF. However, LM4.1-BNF has limitations.

LM4.1-BNF does not explicitly include asymbiotic BNF by bryophytes, lichens, and other organisms beyond soil microbes. This is regulated differently from asymbiotic BNF by soil microbes, specifically by light (Reed et al., 2011). In particular, in boreal forests and arctic tundra, asymbiotic BNF by bryophytes is a significant N flux (DeLuca et al., 2002). Additionally, herbaceous symbiotic BNF in the forest understory could be significant but few studies have quantified its magnitude and controls (Cleveland et al., 1999).

The asymbiotic BNF temperature response is heavily biased towards high latitudes; the studies we used in its derivation had a mean latitude of 60˚ (Chan, 1991; Chapin et al., 1991; Coxson and Kershaw, 1983; Liengen and Olsen, 1997; Roper, 1985). More studies on the asymbiotic BNF temperature response at lower latitudes are necessary.

The symbiotic BNF temperature response could acclimate to changing temperature (Bytnerowicz et al., in review). The C cost of symbiotic BNF, which we assumed to be constant per unit N, could depend on temperature or other factors. These issues could influence the simulated response of symbiotic BNF and consequently total plant biomass C to increasing temperatures due to climate change. Thus, further empirical work on the effect of temperature on symbiotic BNF is necessary.

Finally, more observations of N cycling in general are necessary to validate N cycling representations in land models (Stocker et al., 2016; Thomas et al., 2015; Vicca et al., 2018). Global observations on N limitation of plant growth, soil N, N

gas emission rates, N leaching rates, and, in particular, asymbiotic and symbiotic BNF rates are limited. Constraining these N pools and fluxes is critical to rigorously validating novel N cycling representations in land models.

## 6.2 Extensions

*Robinia pseudoacacia* is the most abundant N-fixing tree species in the coterminous US (Staccone et al., 2020) and is also a common N-fixing tree across temperate forests; it is also found in Africa, Asia, Australia, Europe, and South America
(Cierjacks et al., 2013). As such, *Robinia pseudoacacia* is representative of temperate N-fixing tree species.

The LM4.1-BNF representation of BNF, while implemented and validated in a temperate forest, can be expanded to other terrestrial ecosystems, such as tropical and boreal forests. This will require parameterisation of representative N-fixing and non-fixing tree species but will not require re-structuring the model equations. Furthermore, the LM4.1-BNF representation of BNF could be incorporated into other land models.

Although N-fixing trees are generally important pioneer species and can relieve strong N limitation of plant growth in early succession (Chapin III et al., 1994), N-fixing trees can also be strong competitors. As such, in addition to having a facilitative effect on neighbouring plant growth (Hulvey et al., 2013), they can also have no effect on neighbouring plant growth (Lai et al., 2018; Xu et al., 2020) or an inhibitory effect on neighbouring plant growth (Chapin III et al., 2016; Taylor et al., 2017). This depends on abiotic and biotic factors (Staccone et al., 2021) and could be explored further with LM4.1-
BNF.

## 7 Conclusions

Here we present LM4.1-BNF: an updated representation of BNF and other aspects of N cycling in LM4.1, which is the land component of the GFDL Earth System Model (Zhao et al., 2018a, 2018b). LM4.1-BNF is the first land model to include a representation of the competitive dynamics between N-fixing and non-fixing plants, a mechanistic representation of
asymbiotic BNF, and distinct asymbiotic and symbiotic BNF temperature responses derived from corresponding observations. Comparisons of simulations with observations show that LM4.1-BNF captures observed forest growth, successional dynamics, and major pools and fluxes of C and N and their temporal dynamics at population, community, and ecosystem scales. Furthermore, LM4.1-BNF represents these more accurately than previous representations of BNF in land models. By incorporating both the competitive dynamics between N-fixing and non-fixing plants, which is a key ecological
mechanism that determines ecosystem-scale symbiotic BNF, as well as asymbiotic BNF, LM4.1-BNF yields accurate ecosystem-scale estimates of BNF and its temporal dynamics. Furthermore, the novel representation of soil NO and $N_2O$ emissions in LM4.1-BNF enables the estimation of the magnitude of the terrestrial NO and $N_2O$ source, which can be driven by BNF (Kou-Giesbrecht and Menge, accepted).

The representation of BNF in LM4.1-BNF is general and could be incorporated into other land models. Extending LM4.1-
BNF to other biomes and incorporating LM4.1-BNF within the GFDL Earth System Model would allow a more accurate

assessment of the response of BNF and the terrestrial C sink to elevated atmospheric $CO_2$ concentration, which intensifies N limitation of plant growth (Terrer et al., 2019; Zheng et al., 2020), and elevated N deposition, which relieves N limitation of plant growth (Reay et al., 2008; Schulte-Uebbing and de Vries, 2018; Zheng et al., 2020). In particular, such an endeavour could address whether BNF and N deposition will provide sufficient N to sustain $CO_2$ sequestration by terrestrial ecosystems under elevated atmospheric $CO_2$ concentration.

## Appendix A Model Description

### A.1 N uptake by roots and symbionts

All vegetation types take up inorganic N via passive and active root uptake. Processes in Section A.1 occur on the fast timescale (30 min).

### A.1.1 Passive root uptake of inorganic N

The rate of passive root uptake of inorganic N in soil layer k ($N_{passive}(k)$; [kg N indiv$^{-1}$ yr$^{-1}$]) is

$$N_{passive}(k) = U_{H_2O}(k)\left(\frac{NO_3(k)+NH_4(k)}{H_2O(k)}\right) \tag{A1}$$

where $U_{H_2O}(k)$ is the water uptake flux in soil layer k [kg $H_2O$ m$^{-2}$ yr$^{-1}$], $NO_3(k)$ is the NO$_3^-$ pool in soil layer k [kg N m$^{-2}$], $NH_4(k)$ is the NH$_4^+$ pool in soil layer k [kg N m$^{-2}$], and $H_2O(k)$ is the soil water content of soil layer k [kg $H_2O$ m$^{-2}$]. Note that the solubility of NH$_4^+$ and NO$_3^-$ differ, reflecting a higher cation exchange capacity than anion exchange capacity.

### A.1.2 Active root uptake of inorganic N

The rate of active root uptake of inorganic N in soil layer k ($N_{active}(k)$; [kg N indiv$^{-1}$ yr$^{-1}$]) is

$$N_{active}(k) = \left(r_{NO_3}\frac{NO_3(k)f_{rhiz}(k)/\Delta z(k)}{NO_3(k)f_{rhiz}(k)/\Delta z(k)+k_{M,NO_3}}\frac{\Delta z(k)}{n_{indiv}} + r_{NH_4}\frac{NH_4(k)f_{rhiz}(k)/\Delta z}{NH_4(k)f_{rhiz}(k)/\Delta z(k)+k_{M,NH_4}}\frac{\Delta z(k)}{n_{indiv}}\right)N_{stress} \tag{A2}$$

where $r_{NO_3}$ and $r_{NH_4}$ are rate constants, $f_{rhiz}(k)$ is the rhizosphere volume fraction of soil layer k [m$^3$ m$^{-3}$], $\Delta z(k)$ is the thickness of soil layer k, $k_{M,NO_3}$ and $k_{M,NH_4}$ are half-saturation constants, and $n_{indiv}$ is the spatial density [indiv m$^{-2}$]. $f_{rhiz}(k)$ is calculated as

$$f_{rhiz}(k) = \pi((r_{rhiz} + r_{root})^2 - r_{root}^2)C_r(k)\,SRL\,\frac{n_{indiv}}{\Delta z(k)} \tag{A3}$$

where $r_{rhiz}$ is the radius of the rhizosphere around fine roots, $r_{root}$ is the radius of fine roots, $C_r(k)$ is the biomass C of fine roots in soil layer k [kg C indiv$^{-1}$], and $SRL$ is the specific root length. N stress of the plant ($N_{stress}$; [unitless]) is given in Eq. (A34). This follows LM3-SNAP but is modified to increase with N stress.

### A.1.3 Inorganic N uptake by arbuscular mycorrhizae

The rates of $NO_3^-$ and $NH_4^+$ uptake by AM in soil layer k ($N_{AM,NO_3}(k)$ and $N_{AM,NH_4}(k)$ respectively; [kg N indiv$^{-1}$ yr$^{-1}$]) are

$$N_{AM}(k) = r_{NO_3,AM} \frac{\frac{NO_3(k)}{\Delta z(k)}}{\frac{NO_3(k)}{\Delta z(k)}+k_{AM,NO_3}} \frac{\frac{B_{AM}(k)n_{indiv}}{\Delta z(k)}}{\frac{B_{AM}(k)n_{indiv}}{\Delta z(k)}+k_{AM}} \frac{\Delta z(k)}{n_{indiv}} + r_{NH_4,AM} \frac{\frac{NH_4(k)}{\Delta z(k)}}{\frac{NH_4(k)}{\Delta z(k)}+k_{AM,NH_4}} \frac{\frac{B_{AM}(k)n_{indiv}}{\Delta z(k)}}{\frac{B_{AM}(k)n_{indiv}}{\Delta z(k)}+k_{AM}} \frac{\Delta z(k)}{n_{indiv}} \tag{A4}$$

where $r_{NO_3,AM}$ and $r_{NH_4,AM}$ are rate constants, $k_{AM,NO_3}$ and $k_{AM,NH_4}$ are half-saturation constants, $B_{AM}(k)$ is the biomass C of AM in soil layer k [kg C indiv$^{-1}$], and $k_{AM}$ is a half-saturation constant. This follows LM3-SNAP.

### A.1.4 Organic and inorganic N uptake by ectomycorrhizae

The rates of C and N decomposition by EM in soil layer k of organic matter type i, where i = labile plant-derived, labile microbe-derived, or recalcitrant, ($D_{C,i,EM}(k)$ and $D_{N,i,EM}(k)$ respectively; [kg C indiv$^{-1}$ yr$^{-1}$] and [kg N indiv$^{-1}$ yr$^{-1}$]) are

$$D_{C,i,EM}(k) = \frac{V_{EM,max,ref,i}}{\exp\left(-E_{a,i}/RT_{ref}\right)} \exp\left(-E_{a,i}/RT(k)\right) \frac{\left(\frac{\theta(k)}{\theta_{sat}}\right)^3 \left(1-\frac{\theta(k)}{\theta_{sat}}\right)^{2.5}}{f_{\theta,max}} \frac{C_{U,i}(k)}{n_{indiv}} \frac{B_{EM}(k)/C_{U,i}(k)}{B_{EM}(k)/C_{U,i}(k)+k_{M.EM}} \tag{A5}$$

$$D_{N,i,EM}(k) = \frac{V_{EM,max,ref,i}}{\exp\left(-E_{a,i}/RT_{ref}\right)} \exp\left(-E_{a,i}/RT(k)\right) \frac{\left(\frac{\theta(k)}{\theta_{sat}}\right)^3 \left(1-\frac{\theta(k)}{\theta_{sat}}\right)^{2.5}}{f_{\theta,max}} \frac{N_{U,i}(k)}{n_{indiv}} \frac{B_{EM}(k)/C_{U,i}(k)}{B_{EM}(k)/C_{U,i}(k)+k_{M.EM}} \tag{A6}$$

where $V_{EM,max,ref,i}$ is the maximum decomposition rate of organic matter type i, $E_{a,i}$ is the activation energy of the decomposition of organic matter type i, $R$ is the ideal gas constant, $T_{ref}$ is reference temperature, $T(k)$ is soil temperature of soil layer k [K], $\theta(k)$ is volumetric soil water content of soil layer k [m$^3$ m$^{-3}$], $\theta_{sat}$ is saturation volumetric soil water content [m$^3$ m$^{-3}$], $f_{\theta,max}$ is a factor normalizing the dependence on $\theta(k)$ to a maximum value of 1 [unitless], $B_{EM}(k)$ is the biomass C of EM in soil layer k [kg C indiv$^{-1}$], $C_{U,i}(k)$ is the soil C pool of type i in soil layer k [kg C m$^{-2}$], $N_{U,i}(k)$ is the soil N pool type i in soil layer k [kg C m$^{-2}$], and $k_{M.EM}$ is a half-saturation constant. This follows LM3-SNAP.

The rates of C and N uptake by EM in soil layer k ($C_{EM}(k)$ and $N_{EM}(k)$ respectively; [kg C indiv$^{-1}$ yr$^{-1}$] and [kg N indiv$^{-1}$ yr$^{-1}$]) are

$$C_{EM}(k) = \sum_i \varepsilon_{C,i,EM} D_{C,i,EM}(k) \tag{A7}$$

$$N_{EM}(k) = \sum_i \varepsilon_{N,i,EM} D_{N,i,EM}(k) + r_{NO_3,EM} \frac{\frac{NO_3(k)}{\Delta z(k)}}{\frac{NO_3(k)}{\Delta z(k)}+k_{EM,NO_3}} \frac{\frac{B_{EM}(k)n_{indiv}}{\Delta z(k)}}{\frac{B_{EM}(k)n_{indiv}}{\Delta z(k)}+k_{EM}} \frac{\Delta z(k)}{n_{indiv}} + $$

$$r_{NH_4,EM} \frac{\frac{NH_4(k)}{\Delta z(k)}}{\frac{NH_4(k)}{\Delta z(k)}+k_{EM,NH_4}} \frac{\frac{B_{EM}(k)n_{indiv}}{\Delta z(k)}}{\frac{B_{EM}(k)n_{indiv}}{\Delta z(k)}+k_{EM}} \frac{\Delta z(k)}{n_{indiv}} \tag{A8}$$

where $\varepsilon_{C,i,EM}$ is the C uptake efficiency of soil C type i by EM, $\varepsilon_{N,i,EM}$ is the N uptake efficiency of soil N type i by EM, $r_{NO_3,EM}$ and $r_{NH_4,EM}$ are rate constants, $k_{EM,NO_3}$ and $k_{EM,NH_4}$ are half-saturation constants, and $k_{EM}$ is a half-saturation constant. $\sum_i (1 - \varepsilon_{C,i,EM}) D_{C,i,EM}(k)$ is released as $CO_2$. $\sum_i (1 - \varepsilon_{N,i,EM}) D_{N,i,EM}(k)$ enters $NH_4(k)$. This follows LM3-SNAP but is modified to additionally take up inorganic N.

### A.1.5 Symbiotic BNF by N-fixing bacteria

The symbiotic BNF rate by N-fixing bacteria ($N_{Nfix}$; [kg N indiv$^{-1}$ yr$^{-1}$]) is

$$N_{Nfix} = r_{Nfix} B_{Nfix} f(T) \tag{A9}$$

where $r_{Nfix}$ is a rate constant, $B_{Nfix}$ is the biomass C of the nodule (includes both plant and N-fixing bacteria tissue) [kg C indiv$^{-1}$], and $f_s(T)$ is the soil temperature dependence function. For *Robinia*,

$$f_s(T) = \max\left[0.0, \left(\frac{45.67-(T-273.15)}{45.67-31.89}\right)\left(\frac{(T-273.15)-1.43}{31.89-1.43}\right)^{\frac{31.89-1.43}{45.67-31.89}}\right] \tag{A10}$$

where $T$ is the average soil temperature across soil layers [K]. This reaches its maximum at 31.9 °C (Fig. 2). This is derived from Bytnerowicz et al., (in review).

### A.2 Asymbiotic BNF

Soil microbes are represented as a single C pool that conducts decomposition, nitrification, denitrification, and asymbiotic BNF. The rates of C and N decomposition by soil microbes in soil layer k of organic matter type i, where i = labile plant-derived, labile microbe-derived or recalcitrant, ($D_{C,i}(k)$ and $D_{N,i}(k)$ respectively; [kg C m$^{-2}$ yr$^{-1}$] and [kg N m$^{-2}$ yr$^{-1}$]) are

$$D_{C,i}(k) = \frac{V_{max,ref,i}}{\exp\left(-E_{a,i}/RT_{ref}\right)} \exp\left(-E_{a,i}/RT(k)\right) \frac{\left(\frac{\theta(k)}{\theta_{sat}}\right)^3 \left(1-\frac{\theta(k)}{\theta_{sat}}\right)^{2.5}}{f_{\theta,max}} C_{U,i}(k) \frac{C_M(k)/C_{U,i}(k)}{C_M(k)/C_{U,i}(k)+k_M} \tag{A11}$$

$$D_{N,i}(k) = \frac{V_{max,ref,i}}{\exp\left(-E_{a,i}/RT_{ref}\right)} \exp\left(-E_{a,i}/RT(k)\right) \frac{\left(\frac{\theta(k)}{\theta_{sat}}\right)^3 \left(1-\frac{\theta(k)}{\theta_{sat}}\right)^{2.5}}{f_{\theta,max}} N_{U,i}(k) \frac{C_M(k)/C_{U,i}(k)}{C_M(k)/C_{U,i}(k)+k_M} \tag{A12}$$

where $V_{max,ref,i}$ is the maximum decomposition rate of organic matter type i, $C_M(k)$ is the biomass C of soil microbes in soil layer k [kg C m$^{-2}$], and $k_M$ is the half-saturation constant. This follows LM3-SNAP.

The potential rates of C and N decomposition during denitrification by soil microbes in soil layer k of organic matter type i ($D_{C,i,denit,pot}(k)$ and $D_{N,i,denit,pot}(k)$ respectively; [kg C m$^{-2}$ yr$^{-1}$] and [kg N m$^{-2}$ yr$^{-1}$]) are

$$D_{C,i,denit,pot}(k) = \frac{V_{denit,max,ref,i}}{\exp\left(-E_{a,i}/RT_{ref}\right)} \exp\left(-E_{a,i}/RT(k)\right) \frac{\left(\frac{\theta(k)}{\theta_{sat}}\right)^{5.5}}{f_{\theta,max}} C_{U,i}(k) \frac{C_M(k)/C_{U,i}(k)}{C_M(k)/C_{U,i}(k)+k_{M,denit}} \tag{A13}$$

$$D_{N,i,denit,pot}(k) = \frac{V_{denit,max,ref,i}}{\exp\left(-E_{a,i}/RT_{ref}\right)} \exp\left(-E_{a,i}/RT(k)\right) \frac{\left(\frac{\theta(k)}{\theta_{sat}}\right)^{5.5}}{f_{\theta,max}} N_{U,i}(k) \frac{C_M(k)/C_{U,i}(k)}{C_M(k)/C_{U,i}(k)+k_{M,denit}} \tag{A14}$$

where $V_{denit,max,ref,i}$ is the maximum decomposition rate of organic matter type i during denitrification and $k_{M,denit}$ is the half-saturation constant. This follows LM3-SNAP.

The rates of C and N decomposition during denitrification by soil microbes in soil layer k of organic matter type i ($D_{C,i,denit}(k)$ and $D_{N,i,denit}(k)$ respectively; [kg C m$^{-2}$ yr$^{-1}$] and [kg N m$^{-2}$ yr$^{-1}$]) are

$$D_{C,i,denit}(k) = \frac{D_{C,i,denit,pot}(k)NO_3(k)}{NO_3(k)+k_{denit}f_{denit}\sum_i D_{C,i,denit,pot}(k)} \tag{A15}$$

$$D_{N,i,denit}(k) = \frac{D_{N,i,denit,pot}(k)NO_3(k)}{NO_3(k)+k_{denit}f_{denit}\sum_i D_{C,i,denit,pot}(k)} \qquad \text{(A16)}$$

where $k_{denit}$ is the half-saturation constant, and $f_{denit}$ is the stoichiometric ratio of $NO_3^-$ demand for C decomposition. $D_{C,i,denit}(k)$ and $D_{N,i,denit}(k)$ include $NO_3^-$ limitation of denitrification. This follows LM3-SNAP.

The rate of change of biomass C of soil microbes in soil layer k ($\frac{dC_M(k)}{dt}$; [kg C m$^{-2}$ yr$^{-1}$]) is

$$\frac{dC_M(k)}{dt} = G_{M,C}(k) - \frac{C_M(k)}{\tau_M} \qquad \text{(A17)}$$

where $G_{M,C}(k)$ is the C growth rate of soil microbes in soil layer k (Eq. (A23)) and $\tau_M$ is the combined maintenance respiration and turnover time of soil microbes. This follows LM3-SNAP.

The rate of change of biomass N of soil microbes in soil layer k ($\frac{dN_M(k)}{dt}$; [kg N m$^{-2}$ yr$^{-1}$]) is

$$\frac{dN_M(k)}{dt} = G_{M,N}(k) - \frac{\frac{C_M(k)}{\tau_M}\varepsilon_{t,M}}{C:N_M} \qquad \text{(A18)}$$

where $G_{M,N}(k)$ is the N growth rate of soil microbes in soil layer k (Eq. (A24)), $\varepsilon_{t,M}$ is the fraction of maintenance respiration in combined maintenance respiration and turnover, and $C:N_M$ is the C:N ratio of soil microbes. This follows LM3-SNAP.

The maintenance respiration rate of soil microbes in soil layer k ($R_{maint}(k)$; [kg C m$^{-2}$ yr$^{-1}$]) is

$$R_{maint}(k) = \frac{C_M(k)}{\tau_M}\left(1 - \varepsilon_{t,M}\right) \qquad \text{(A19)}$$

This is released as $CO_2$. This follows LM3-SNAP.

The asymbiotic BNF rate of soil microbes in soil layer k ($N_{Nfix\ asymb}(k)$; [kg N m$^{-2}$ yr$^{-1}$]) is

$$N_{Nfix\ asymb}(k) = r_{Nfix\ asymb}C_M(k)f\left(T(k)\right) \qquad \text{(A20)}$$

where $r_{Nfix\ asymb}$ is a rate constant and $f_a\left(T(k)\right)$ is the soil temperature dependence function.

$$f_a\left(T(k)\right) = e^{-2.6+0.21(T(k)-273.15)\left(1-\frac{0.5(T(k)-273.15)}{24.4}\right)} \qquad \text{(A21)}$$

which reaches its maximum at 24.4 °C (Fig. 2). This is derived from the observations compiled by Houlton et al., 2008 with the study of symbiotic BNF removed (Schomberg and Weaver, 1992) and is normalized to a maximum of 1.

The N surplus or deficit of soil microbes in soil layer k ($\phi_N(k)$; [kg N m$^{-2}$ yr$^{-1}$]) is

$$\phi_N(k) = \sum_i \varepsilon_{N,i}D_{N,i}(k) + \sum_i \varepsilon_{N,i}D_{N,i,denit}(k) + N_{Nfix,asymb}(k) + \frac{\sum_i \varepsilon_{C,i}D_{C,i}(k)+\sum_i \varepsilon_{C,i}D_{C,i,denit}(k)-R_{maint}(k)}{C:N_M} \qquad \text{(A22)}$$

where $\varepsilon_{N,i}$ is the N uptake efficiency of soil N type i by soil microbes and $\varepsilon_{C,i}$ is the C uptake efficiency of soil C type i by
soil microbes. $\phi_N(k) > 0$ indicates net N mineralization (N surplus) and $\phi_N(k) < 0$ indicates net N immobilization (N deficit) by soil microbes in soil layer k. $\sum_i\left(1 - \varepsilon_{C,i}\right)\left(D_{C,i}(k) + D_{C,i,denit}(k)\right)$ is released as $CO_2$. $\sum_i(1 - \varepsilon_N)(D_{N,i}(k) + D_{N,i,denit}(k))$ enters $NH_4(k)$. This follows LM3-SNAP but was modified to include asymbiotic BNF.

$G_{M,C}(k)$ and $G_{M,N}(k)$ depend on whether growth of soil microbes is C-limited ($\phi_N(k) \geq -Imm_{max}(k)$) or N-limited ($\phi_N(k) < -Imm_{max}(k)$) and are calculated as

$G_{M,C}(k) =$

$$\begin{cases} \sum_i \varepsilon_{C,i} D_{C,i}(k) + \sum_i \varepsilon_{C,i} D_{C,i,denit}(k), & \phi_N(k) \geq -Imm_{max}(k) \\ C:N_M \left[ \sum_i \varepsilon_{N,i} D_{N,i}(k) + \sum_i \varepsilon_{N,i} D_{N,i,denit}(k) + N_{Nfix,asymb}(k) + Imm_{max}(k) \right] + R_{maint}(k), & \phi_N(k) < -Imm_{max}(k) \end{cases}$$

(A23)

$G_{M,N}(k) =$

$$\begin{cases} \frac{\sum_i \varepsilon_{C,i} D_{C,i}(k) + \sum_i \varepsilon_{C,i} D_{C,i,denit}(k) - R_{maint}(k)}{C:N_M}, & \phi_N(k) \geq -Imm_{max}(k) \\ \sum_i \varepsilon_{N,i} D_{N,i}(k) + \sum_i \varepsilon_{N,i} D_{N,i,denit}(k) + N_{Nfix,asymb}(k) + Imm_{max}(k), & \phi_N(k) < -Imm_{max}(k) \end{cases}$$

(A24)

This follows LM3-SNAP but was modified to include asymbiotic BNF.

The maximum N immobilization rate of soil microbes in soil layer k ($Imm_{max}(k)$; [kg N m$^{-2}$ yr$^{-1}$]) is

$$Imm_{max}(k) = \left( V_{max,ref,NH_4} \exp\left( {-E_{a,NH_4}} \big/ {RT(k)} \right) NH_4(k) + \right.$$

$$\left. V_{max,ref,NO_3} \exp\left( {-E_{a,NO_3}} \big/ {RT(k)} \right) NO_3(k) \right) \frac{\left( \frac{\theta(k)}{\theta_{sat}} \right)^3 \left( 1 - \frac{\theta(k)}{\theta_{sat}} \right)^{2.5}}{f_{\theta,max}}$$

(A25)

where $V_{max,ref,NH_4}$ is the maximum NH$_4^+$ immobilization rate, $E_{a,NH_4}$ is the activation energy of NH$_4^+$ immobilization, $V_{max,ref,NO_3}$ is the maximum NO$_3^-$ immobilization rate, and $E_{a,NO_3}$ is the activation energy of NO$_3^-$ immobilization. This follows LM3-SNAP.

When growth of soil microbes in soil layer k is N-limited ($\phi_N(k) < -Imm_{max}(k)$), there is overflow respiration of excess C. The overflow respiration of excess C in soil layer k ($R_{overflow}(k)$; [kg C m$^{-2}$ yr$^{-1}$]) is

$$R_{overflow}(k) = \begin{cases} 0, & \phi_N(k) \geq -Imm_{max}(k) \\ -\left( \phi_N(k) + Imm_{max}(k) \right) C:N_M, & \phi_N(k) < -Imm_{max}(k) \end{cases}$$

(A26)

This is released as CO$_2$.

If $\phi_N(k) > 0$, the net N mineralization flux in soil layer k is $\phi_N(k)$. If $-Imm_{max}(k) \leq \phi_N(k) \leq 0$, the net N immobilization flux in soil layer k is $-\phi_N(k)$. If $\phi_N(k) < -Imm_{max}(k)$, the net N immobilization flux in soil layer k is $-Imm_{max}(k)$. The fraction of the net N immobilization flux in soil layer k that is NH$_4^+$ immobilization is

$$\frac{\frac{V_{max,ref,NH_4}}{\exp\left( {-E_{a,NH_4}} \big/ {RT_{ref}} \right)} \exp\left( {-E_{a,NH_4}} \big/ {RT(k)} \right) NH_4(k)}{\frac{V_{max,ref,NH_4}}{\exp\left( {-E_{a,NH_4}} \big/ {RT_{ref}} \right)} \exp\left( {-E_{a,NH_4}} \big/ {RT(k)} \right) NH_4(k) + \frac{V_{max,ref,NO_3}}{\exp\left( {-E_{a,NO_3}} \big/ {RT_{ref}} \right)} \exp\left( {-E_{a,NO_3}} \big/ {RT(k)} \right) NO_3(k)}.$$ The fraction of the net N immobilization

flux in soil layer k that is NO$_3^-$ immobilization is

$$\frac{\frac{V_{max,ref,NO_3}}{\exp\left( {-E_{a,NO_3}} \big/ {RT_{ref}} \right)} \exp\left( {-E_{a,NO_3}} \big/ {RT(k)} \right) NO_3(k)}{\frac{V_{max,ref,NH_4}}{\exp\left( {-E_{a,NH_4}} \big/ {RT_{ref}} \right)} \exp\left( {-E_{a,NH_4}} \big/ {RT(k)} \right) NH_4(k) + \frac{V_{max,ref,NO_3}}{\exp\left( {-E_{a,NO_3}} \big/ {RT_{ref}} \right)} \exp\left( {-E_{a,NO_3}} \big/ {RT(k)} \right) NO_3(k)}.$$ This follows LM3-SNAP.

Processes in Section A.2 occur on the fast timescale (30 min).

**A.3 Plant growth and N limitation**

The non-structural C pool ($NSC$; [kg C indiv$^{-1}$]) gains C from photosynthesis. $NSC$ loses C to respiration and C allocation to growth, symbionts, and root C exudation. The rate of change of $NSC$ ($\frac{dNSC}{dt}$; [kg C indiv$^{-1}$ yr$^{-1}$]) is

$$\frac{dNSC}{dt} = P - R - \left(G_{C,l} + G_{C,r} + G_{C,sw} + G_{C,seed}\right) - C_{alloc} - L_{C,exudate} \tag{A27}$$

where $P$ is the photosynthesis rate [kg C indiv$^{-1}$ yr$^{-1}$], $R$ is the respiration rate (maintenance and growth) [kg C indiv$^{-1}$ yr$^{-1}$], $G_{C,l}$ is the growth rate of the leaf C pool ($C_l$; [kg C indiv$^{-1}$]) [kg C indiv$^{-1}$ yr$^{-1}$], $G_{C,r}$ is the growth rate of the fine root C pool

($C_r$; [kg C indiv$^{-1}$]) [kg C indiv$^{-1}$ yr$^{-1}$], $G_{C,sw}$ is the growth rate of the sapwood C pool ($C_{sw}$; [kg C indiv$^{-1}$]) [kg C indiv$^{-1}$ yr$^{-1}$], $G_{C,seed}$ is growth rate of the seed C pool ($C_{seed}$; [kg C indiv$^{-1}$]) [kg C indiv$^{-1}$ yr$^{-1}$], $C_{alloc}$ is the rate of C allocation to symbionts (Eq. (A43-46)), and $L_{C,exudate}$ is the rate of root C exudation (Eq. (A38)). Note that sapwood is converted to heartwood following Martinez Cano et al., (2020).

The non-structural N pool ($NSN$; [kg N indiv$^{-1}$]) gains N from N uptake via roots and symbionts. $NSN$ loses N to N

allocation to growth, symbionts, and root N exudation. The rate of change of $NSN$ ($\frac{dNSN}{dt}$; [kg N indiv$^{-1}$ yr$^{-1}$]) is

$$\frac{dNSN}{dt} = U - \left(\frac{G_{C,l}}{C:N_l} + \frac{G_{C,r}}{C:N_r} + \frac{G_{C,sw}}{C:N_{sw}} + \frac{G_{C,seed}}{C:N_{seed}}\right) - N_{alloc} - L_{N,exudate} \tag{A28}$$

where $U$ is the N uptake rate via roots and symbionts (Eq. (A55)) [kg N indiv$^{-1}$ yr$^{-1}$], $C:N_l$ is the fixed C:N ratio of leaves, $C:N_r$ is the fixed C:N ratio of fine roots, $C:N_{sw}$ is the fixed C:N ratio of sapwood, $C:N_{seed}$ is the fixed C:N ratio of seeds, $N_{alloc}$ is the rate of N allocation to symbionts (Eq. (A47)), and $L_{N,exudate}$ is the rate of root N exudation (Eq. (A39)).

Non-N-limited growth is calculated according to Weng et al., (2015). The total allocation of $NSC$ to growth is determined by the target $NSC$ ($NSC_{target}$; [kg C indiv$^{-1}$]) and minimizes the deviation between $NSC$ and $NSC_{target}$. $NSC_{target}$ is a multiple of the target $C_l$ ($C_{l,target}$; [kg C indiv$^{-1}$]), which reflects the ability of a plant to refoliate after defoliation (Hoch et al., 2003; Richardson et al., 2013), and is calculated as

$$NSC_{target} = q\ C_{l,target} \tag{A29}$$

where $q$ is a proportionality constant. The allocation of $NSC$ to the growth of each tissue depends on the total allocation of $NSC$ to growth and the target C pool of each tissue, and minimizes the deviation between the C pool of each tissue and the target C pool of each tissue. The target C pool of each tissue is dynamic and is determined by allometry (Eq. (A40-42)).

In LM4.1-BNF, $G_{C,l}$, $G_{C,r}$, $G_{C,sw}$, and $G_{C,seed}$ are adjusted to include N limitation and are calculated as

$$G_{C,l} = (1 - N_{stress})\Delta_l \tag{A30}$$

$$G_{C,r} = \Delta_r \tag{A31}$$

$$G_{C,sw} = (1 - N_{stress})\Delta_{sw} \tag{A32}$$

$$G_{C,seed} = (1 - N_{stress})\Delta_{seed} \tag{A33}$$

where $N_{stress}$ is N stress [unitless] and $\Delta_l, \Delta_r, \Delta_{sw}$, and $\Delta_{seed}$ are the non-N-limited growth rates of $C_l, C_r, C_{sw}$, and $C_{seed}$ respectively [kg C indiv$^{-1}$ yr$^{-1}$] following Weng et al., (2015). Because plants increase C allocation to fine roots relative to other tissues when N-limited (Poorter et al., 2012), $G_{C,r}$ is not adjusted to include N limitation.

In LM4.1-BNF, $N_{stress}$ is the relative difference between $NSN$ and $NSN_{target}$ and is calculated as

$$N_{stress} = \max\left[0, \frac{NSN_{target} - NSN}{NSN_{target}}\right] \tag{A34}$$

where $NSN_{target}$ is the target $NSN$ [kg N indiv$^{-1}$]. $N_{stress}$ is smoothed with a low-pass filter over 30 days to reflect the persisting influence of N stress (Mooney et al., 1991). $NSN_{target}$ is calculated as

$$NSN_{target} = \frac{NSC_{target}}{C:N_l} \tag{A35}$$

This is similar to LM3-SNAP, which compared the target leaf and root N pools to $NSN$, but is modified to reflect the treatment of $NSC_{target}$ in LM4.1 by including the target sapwood and seed N pools.

N demand for plant growth ($N_{demand}$; [kg N indiv$^{-1}$]) is

$$N_{demand} = \frac{G_{C,l}}{C:N_l} + \frac{G_{C,r}}{C:N_r} + \frac{G_{C,sw}}{C:N_{sw}} + \frac{G_{C,seed}}{C:N_{seed}} \tag{A36}$$

If $N_{demand} > 0.5\ NSN$, $G_{C,l}, G_{C,r}, G_{C,sw}$, and $G_{C,seed}$ are reduced to prevent $N_{demand}$ from depleting $NSN$

$$G_{C,t} = \begin{cases} G_{C,t}, & N_{demand} < 0.5\ NSN \\ G_{C,t}\ \frac{NSN}{N_{demand}}, & N_{demand} > 0.5\ NSN \end{cases} \tag{A37}$$

where t = leaf, root, sapwood, or seed and 0.5 was set to maintain a baseline $NSN$.

Plant turnover decreases $C_l$, $C_r$, and $C_{sw}$ and from $N_l$, $N_r$, and $N_{sw}$ at a constant tissue-specific rate and enters $C_{U,labile\ plant-derived}$ or $C_{U,recalcitrant}$ and $N_{U,labile\ plant-derived}$ or $N_{U,recalcitrant}$ respectively. A fraction of the turnover of $C_l$ and $N_l$ is retranslocated into $NSC$ and $NSN$ respectively.

Under N limitation, plants increase root C exudation to stimulate N mineralization in the rhizosphere (rhizosphere priming; Cheng et al., 2014; Finzi et al., 2015). $L_{C,exudate}$ increases with $N_{stress}$ and is calculated as

$$L_{C,exudate} = r_{leakage,C}\ NSC\ N_{stress} \tag{A38}$$

where $r_{leakage,C}$ is a rate constant. $L_{C,exudate}$ enters the rhizosphere $C_{U,labile\ plant-derived}$.

Under N limitation, plants decrease root N exudation (Canarini et al., 2019). $L_{N,exudate}$ decreases with $N_{stress}$ and is calculated as

$$L_{N,exudate} = r_{leakage,N}\ NSN\ (1 - N_{stress}) \tag{A39}$$

where $r_{leakage,N}$ is a rate constant. $L_{N,exudate}$ enters the rhizosphere $N_{U,labile\ plant-derived}$.

Plant growth occurs during the growing season. Plant maintenance respiration and plant turnover occur throughout the year. After the transition from the growing season to the non-growing season, turnover of $C_l$ and $N_l$ occur at an elevated rate until $C_l = 0$ and $N_l = 0$. Plant C allocation to symbionts occurs during the growing season. Root exudation occurs during the

growing season. Symbiont growth, maintenance respiration, and turnover occur throughout the year. N uptake by roots and symbionts occurs throughout the year.

Photosynthesis and respiration occur on the fast timescale (30min). Plant turnover occurs on the fast timescale (30 min). Plant growth, plant C allocation to symbionts, and root exudation occur on the daily timescale.

## A.4 Plant allometry

Plant allometry follows Martinez Cano et al., 2020. Crown area ($CA$; [m$^2$]) is a function of diameter at breast height (power function)

$$CA = \alpha_{CA} D^{\theta_{CA}} \tag{A40}$$

where $D$ is diameter at breast height [m], and $\alpha_{CA}$ and $\theta_{CA}$ are allometry parameters.

Height ($H$; [m]) is a function of diameter at breast height (generalized Michaelis-Menten equation)

$$H = \frac{\alpha_{HT} D^{\theta_{HT}}}{\gamma_{HT} + D^{\theta_{HT}}} \tag{A41}$$

where $\alpha_{HT}$, $\theta_{HT}$, and $\gamma_{HT}$ are allometry parameters.

Wood mass ($W$; [kg C]) is a function of diameter at breast height and height

$$W = \alpha_{BM} \rho_{wood} D^2 H \tag{A42}$$

where $\alpha_{BM}$ is an allometry parameter and $\rho_{wood}$ is wood density.

## A.5 Plant C allocation to symbionts (AM, EM and N-fixing bacteria)

The rate of C allocation to AM ($C_{alloc,AM}$; [kg C indiv$^{-1}$ yr$^{-1}$]) is

$$C_{alloc,AM} = f_{alloc,AM} NSC \tag{A43}$$

where $f_{alloc,AM}$ is the fraction of $NSC$ allocated to AM per unit time. $C_{alloc,AM}$ is not related to $N_{stress}$ because, although AM increase N uptake, AM is maintained by the plant primarily for phosphorus uptake (Smith and Smith, 2011).

The rate of C allocation to EM ($C_{alloc,EM}$; [kg C indiv$^{-1}$ yr$^{-1}$]) is

$$C_{alloc,EM} = f_{alloc,EM} NSC N_{stress} \tag{A44}$$

where $f_{alloc,EM}$ is the maximum fraction of $NSC$ allocated to EM per unit time. $C_{alloc,EM}$ is a function of $N_{stress}$ because biomass C of EM increases with N limitation (Phillips et al., 2013).

Plants that associate with N-fixing bacteria can regulate symbiotic BNF to different extents, termed their BNF strategy (Menge et al., 2015). For plants with a perfectly facultative BNF strategy, symbiotic BNF increases with N limitation. For plants with an incomplete BNF strategy, symbiotic BNF increases with N limitation but is maintained at a minimum. For plants with an obligate BNF strategy, symbiotic BNF is constant. For plants with either a facultative or an incomplete BNF strategy, the rate of C allocation by the plant to N-fixing bacteria ($C_{alloc,Nfix}$; [kg C indiv$^{-1}$ yr$^{-1}$]) is

$$C_{alloc,Nfix} = \max[f_{alloc,Nfix} NSC N_{stress}, f_{alloc,Nfix,min} NSC] \tag{A45}$$

where $f_{alloc,Nfix}$ is the fraction of $NSC$ allocated to N-fixing bacteria per unit time and $f_{alloc,Nfix,min}$ is the minimum fraction of $NSC$ allocated to N-fixing bacteria per unit time. For plants with a perfectly facultative BNF strategy $f_{alloc,Nfix,min} = 0$ and for plants with an incomplete down-regulator BNF strategy $f_{alloc,Nfix,min} > 0$. We model *Robinia* with an incomplete down-regulator BNF strategy.

For plants with an obligate BNF strategy, $C_{alloc,Nfix}$ is

$$C_{alloc,Nfix} = f_{alloc,N\,fix}\,NSC \tag{A46}$$

Additionally, plants allocate a small quantity of N to symbionts such that symbiont growth can be initiated. The rate of N allocation by the plant to symbionts ($N_{alloc,j}$; j = AM, EM, N-fixing bacteria; [kg N indiv$^{-1}$ yr$^{-1}$]) is

$$N_{alloc,j} = \frac{C_{alloc,j}}{C:N_{alloc}} \tag{A47}$$

where $C:N_{alloc}$ is the C:N ratio of C and N allocated to symbionts by the plant.

Processes in Section A.5 occur on the daily timescale.

## A.6 Growth and turnover of symbionts

Plant C allocation to symbionts is transferred to an intermediate C pool ($C_{int,j}$; j = AM, EM, or N-fixing bacteria; [kg C indiv$^{-1}$]). The rate of change of $C_{int,j}$ ($\frac{dC_{int,j}}{dt}$; [kg C indiv$^{-1}$ yr$^{-1}$]) is

$$\frac{dC_{int,j}}{dt} = C_{alloc,j} - \frac{G_j}{\varepsilon_{symb}} \tag{A48}$$

where $\varepsilon_{symb}$ is the proportion of C uptake by a symbiont from $C_{int,j}$ that is assimilated. The C growth rate of a symbiont with strategy j ($G_j$; j = AM or N-fixing bacteria; [kg C indiv$^{-1}$ yr$^{-1}$]) is

$$G_j = \varepsilon_{symb} r_{growth} C_{int,j} \tag{A49}$$

where $r_{growth}$ is the growth rate of a symbiont.

The growth rate of EM ($G_{EM}$; [kg C indiv$^{-1}$ yr$^{-1}$]) is

$$G_{EM} = \varepsilon_{symb}(r_{growth} C_{int,EM} + C_{EM}) \tag{A50}$$

$\frac{G_j}{\varepsilon_{symb}}\left(1 - \varepsilon_{symb}\right)$ (j = AM, EM, or N-fixing bacteria) is released as $CO_2$ (growth respiration).

The rate of change of biomass C of a symbiont with strategy j ($\frac{dB_j}{dt}$; j = AM or EM; [kg C indiv$^{-1}$ yr$^{-1}$]) is

$$\frac{dB_j}{dt} = G_j - \xi_j B_j - \frac{B_j}{\tau_j} \tag{A51}$$

where $\xi_j$ is rate of maintenance respiration of a symbiont with strategy j and $\tau_j$ is the turnover time of a symbiont with strategy j. $\xi_j B_j$ is released as $CO_2$ (maintenance respiration). $\frac{B_j}{\tau_j}$ enters $C_{U,labile\,microbe-derived}$ and $\frac{B_j}{C:N_j\tau_j}$ enters $N_{U,labile\,microbe-derived}$, where $C:N_j$ is the C:N ratio of a symbiont with strategy j.

The rate of change of biomass C of N-fixing bacteria ($\frac{dB_{Nfix}}{dt}$; [kg C indiv$^{-1}$ yr$^{-1}$]) is

$$825 \quad \frac{dB_{Nfix}}{dt} = G_{Nfix} - \xi_{Nfix}B_{Nfix} - cost_{Nfix}N_{Nfix} - \frac{B_{Nfix}}{\tau_{Nfix}} \tag{A52}$$

where $\xi_{Nfix}$ is rate of maintenance respiration of N-fixing bacteria, $cost_{Nfix}$ is the C cost of symbiotic BNF per unit N, and $\tau_{Nfix}$ is the turnover time of N-fixing bacteria. $\xi_{Nfix}B_{Nfix}$ is released as $CO_2$ (maintenance respiration) and $cost_{Nfix}N_{Nfix}$ is released as $CO_2$ (respiration associated with symbiotic BNF). $\frac{B_{Nfix}}{\tau_{Nfix}}$ enters $C_{U,labile\ microbe-derived}$ and $\frac{B_{Nfix}}{C:N_{Nfix}\tau_{Nfix}}$ enters $N_{U,labile\ microbe-derived}$, where $C:N_{Nfix}$ is the C:N ratio of N-fixing bacteria.

N acquired by symbionts is transferred to an intermediate N pool ($N_{int,j}$; [kg N indiv$^{-1}$]). The rate of change of $N_{int,j}$ ($\frac{dN_{int,j}}{dt}$; j = AM or EM; [kg N indiv$^{-1}$ yr$^{-1}$]) is

$$\frac{dN_{int,j}}{dt} = \sum_k N_j + \frac{C_{alloc,j}}{C:N_{alloc}} - \frac{1}{C:N_j}\left(G_j - \xi_j B_j\right) - r_{up,veg}N_{int,j} \tag{A53}$$

where $r_{up,veg}$ is the rate of plant N uptake from $N_{int,j}$.

The rate of change of $N_{int,Nfix}$ ($\frac{dN_{int,Nfix}}{dt}$; [kg N indiv$^{-1}$ yr$^{-1}$]) is

$$835 \quad \frac{dN_{int,Nfix}}{dt} = N_{Nfix} + \frac{C_{alloc,Nfix}}{C:N_{alloc}} - \frac{1}{C:N_{Nfix}}\left(G_{Nfix} - \xi_{Nfix}B_{Nfix} - cost_{Nfix}N_{Nfix}\right) - r_{up,veg}N_{int,Nfix} \tag{A54}$$

$U$ is calculated as

$$U = \sum_k N_{passive}(k) + \sum_k N_{active}(k) + \sum_j r_{up,veg}N_{int,j} \tag{A55}$$

Processes in Section A.6 occur on the fast timescale (30 min). The purpose of $C_{int,j}$ and $N_{int,j}$ are to translate between the fast timescale (30min) and the daily timescale. Plant C allocation to symbionts occurs on the daily timescale (alongside plant

growth), but plant N uptake occurs on the fast timescale.

## A.7 Dynamic plant C allocation to N uptake relative to plant growth

The order of plant C allocation to growth, symbionts, and rhizosphere priming is determined by C limitation relative to N limitation (Cheng et al., 2014; Finzi et al., 2015; Poorter et al., 2012; Treseder, 2004; Zheng et al., 2019). If a plant is more C-limited than N-limited, $NSC < NSN \cdot C:N_{leaf}$. The plant allocates C to growth, then to N-fixing bacteria (if associated)

and EM (if associated), then to rhizosphere priming, and finally to AM. If a plant more N-limited than C-limited, $NSC > NSN \cdot C:N_{leaf}$. The plant allocates C to N-fixing bacteria (if associated) and EM (if associated), then to rhizosphere priming, then to growth, and finally to AM.

## A.8 Soil N₂O and NO emissions

Soil $N_2O$ and NO emissions occur during nitrification (aerobic oxidation of $NH_4^+$ with oxygen as an electron acceptor, which

produces $N_2O$ and NO as by-products) and denitrification (anaerobic oxidation of organic C with $NO_3^-$ as an electron acceptor, which produces $N_2O$ as a by-product).

Nitrification rate in soil layer k ($nit(k)$; [kg N m$^{-2}$ yr$^{-1}$]) is

$$nit(k) = \frac{V_{nit,max,ref}}{\exp\left(-E_{a,nit}/RT_{ref}\right)} \exp\left(-E_{a,nit}/RT(k)\right) \frac{\left(\frac{\theta(k)}{\theta_{sat}}\right)^3 \left(1 - \frac{\theta(k)}{\theta_{sat}}\right)^{2.5}}{f_{\theta,max}} NH_4(k) \tag{A56}$$

where $V_{nit,max,ref}$ is the maximum nitrification rate and $E_{a,nit}$ is the activation energy of nitrification. This follows LM3-SNAP.

Denitrification rate in soil layer k ($denit(k)$; [kg N m$^{-2}$ yr$^{-1}$]) is

$$denit(k) = f_{denit} \sum_i D_{C,i,denit}(k) \tag{A57}$$

This follows LM3-SNAP.

Soil N$_2$O emission rate in soil layer k ($N_2O(k)$; [kg N m$^{-2}$ yr$^{-1}$]) is

$$N_2O(k) = 0.004\, nit(k) + \frac{1}{1+\max\left(3.52,22\exp\left(-0.8\frac{NO_3(k)}{HR(k)}\right)\right)\max\left(0.1,0.015\frac{\theta(k)}{\theta_{sat}}-0.32\right)} denit(k) \tag{A58}$$

where and $HR(k)$ is heterotrophic respiration in soil layer k (summation of maintenance respiration, overflow respiration, and decomposition respiration) [kg C m$^{-2}$ yr$^{-1}$]. This follows LM3V-N (Huang and Gerber, 2015).

Soil NO emission rate in soil layer k ($NO(k)$; [kg N m$^{-2}$ yr$^{-1}$]) is

$$NO(k) = 0.004 \left( 15.2 + \frac{35.5 \tan^{-1}\left(0.68\pi\left(2.09\left(1-\frac{\theta(k)}{\theta_{sat}}\right)^{4/3}-1.68\right)\right)}{\pi} \right) nit(k) \tag{A59}$$

This follows LM3V-N (Huang and Gerber, 2015).

Processes in Section A.8 occur on the fast timescale (30 min).

**A.9 Additional N sources**

N deposition enters $N_{U,labile\,plant-derived}$, $NH_4$ and $NO_3$. Processes in Section A.9 occur on the fast timescale (30 min).

**Appendix B Data availability and processing**

The US Forest Inventory and Analysis (FIA) database (US Forest Service, 2020a) was downloaded in 2019 from https://apps.fs.usda.gov/fia/datamart/CSV/datamart_csv.html. For tree data, plots with Plot Design Code 1 (national plot design) were selected, dead and cut trees were excluded, trees measured at the root collar were excluded, trees with visually estimated or modelled heights were excluded, and accessible forest land was selected (excludes agriculture and urban areas). Canopy trees had crown class code open grown, dominant, or codominant. Understory trees had crown class code intermediate or overtopped. For soil data, mineral soil layers were selected. For seedling data, plots with Plot Design Code 1 (national plot design) were selected, and accessible forest land was selected (excludes agriculture and urban areas).

US FIA Forest Health Monitoring database (US Forest Service, 2020b) was downloaded in 2019 from https://www.fia.fs.fed.us/tools-data/other_data/index.php. Dead trees and trees with damaged crowns were excluded.

Timberland and woodland land uses were selected.

The Biomass and Allometry Database (BAAD; Falster et al., 2015) was downloaded from https://esajournals.onlinelibrary.wiley.com/doi/abs/10.1890/14-1889.1. Field / wild plants were selected. Temperate forest plants were selected.

TRY database data (Kattge et al., 2020) were downloaded in 2019 from https://www.try-db.org/TryWeb/Prop0.php.

Total soil C, total soil N, soil $NH_4^+$, soil nitrate $NO_3^-$, N mineralization rate, net nitrification rate, and bulk density from CHL (Knoepp, 2009a, 2009b, 2018) were downloaded from https://coweeta.uga.edu/dbpublic/dataset_details.asp?accession=1124, https://coweeta.uga.edu/dbpublic/dataset_details.asp?accession=1125, and https://coweeta.uga.edu/dbpublic/dataset_details.asp?accession=1123.

Soil $N_2O$ and NO emissions are from Stehfest and Bouwman, (2006). Data from deciduous temperate forests were selected.

Net ecosystem production at the hourly timescale (Oishi, 2020) were downloaded from https://ameriflux.lbl.gov/doi/AmeriFlux/US-Cwt. Growing season was May through October.

Gross primary production, net primary production, heterotrophic respiration and net ecosystem production are from the ForC database (Anderson-Teixeira et al., 2018). Data from deciduous temperate broadleaf forests were selected.

**Appendix C Model parameterization**

Eq. (A40) was fit using nonlinear least-squares with the US FIA Forest Health Monitoring database (US Forest Service, 2020b) for each species to calculate $\alpha_{CA}$ and $\theta_{CA}$ (Fig. D10-11). Eq. (A41) was fit using nonlinear least-squares with the US FIA database tree data (US Forest Service, 2020a) for each species to calculate $\alpha_{HT}$, $\theta_{HT}$ and $\gamma_{HT}$ (Fig. D12-13). Eq. (A42) was fit using nonlinear least-squares with the US FIA database tree data (US Forest Service, 2020a) for each species to calculate $\alpha_{BM}$ (Fig. D14-15). Note that wood mass (aboveground and belowground) is calculated by the US FIA using

allometric equations of dbh.

The following equations from Weng et al., (2015) were utilized with the US FIA database tree data from consecutive censuses for each species (US Forest Service, 2020a) to calculate canopy tree background mortality rate ($\mu_C$) and understory tree background mortality rate ($\mu_U$)

$$n_{dead,C} = n_C(1 - e^{-\mu_C}) \tag{A60}$$

$$n_{dead,U} = n_U\left(1 - e^{-\mu_U\left(1 + \frac{4e^{30(0.025-D_U)}}{1+e^{30(0.025-D_U)}}\right)}\right) \tag{A61}$$

where $n_{dead,C}$ is the mean number of dead canopy individuals in a given year [indiv m$^{-2}$], $n_C$ is the mean number of canopy individuals in a given year [indiv m$^{-2}$], $n_{dead,U}$ is the mean number of dead understory individuals in a given year [indiv m$^{-2}$], $n_U$ is the mean number of understory individuals in a given year [indiv m$^{-2}$], and $D_U$ is the mean diameter at breast height of an understory individual in a given year [m].

The following equation from Martinez Cano et al., (2020) was fit using nonlinear least-squares with the BAAD (Falster et al., 2015) to calculate the proportionality constant between crown area and cross-sectional sapwood area ($\varphi_{CSA}$) (Fig. D16-17)

$$\varphi_{CSA} = \frac{SA_{sw}(\gamma_{HT}+D^{\theta_{HT}})}{D^{\theta_{CA}+\theta_{HT}}} \tag{A62}$$

where $SA_{sw}$ is the cross-sectional sapwood area at breast height [m$^2$] and $D$ is the diameter [m] from the BAAD (for deciduous angiosperm).

The following equation from Weng et al., (2015) was utilized with the BAAD (Falster et al., 2015) to calculate the proportionality constant between root area and leaf area ($\varphi_{RL}$)

$$\varphi_{RL} = \frac{C_r}{C_l} \bullet LMA \bullet SRA = 0.43 \bullet 0.0375 \left[\frac{kg\,C}{m^2}\right] \bullet 45 \left[\frac{m^2}{kg\,C}\right] = 0.79 \text{ , where 0.43 is from the BAAD (for deciduous}$$

angiosperm), 0.0375 kg C m$^{-2}$ is from Poorter et al., (2009) (for deciduous trees), and 45 m$^2$ kg C is from Jackson et al., (1997) (for temperate deciduous forests).

The following equation from Martinez Cano et al., (2020) was utilized with the BAAD (Falster et al., 2015) to calculate the fraction of sapwood in branches ($f_{br}$)

$f_{br} = \frac{C_{br}}{C_{sw}} = 0.255$, where 0.255 is from the BAAD. Note that we did not distinguish between deciduous angiosperm and evergreen gymnosperm due to insufficient data.

The following equation (from Eq. (A20)) was used to calculate $r_{Nfix\,asymb}$

$$r_{Nfix\,asymb} = \frac{N_{Nfix\,asymb}}{C_M} = \frac{12\left[\frac{kg\,N}{ha\,yr}\right]}{50\,000\left[\frac{kgC}{ha}\right]\times0.01} = 0.024 \left[\frac{kg\,N}{kg\,C\,yr}\right] \text{ where 12 kg N ha}^{-1}\text{ yr}^{-1} \text{ is from Reed et al., (2011), 50000 kg C}$$

ha$^{-1}$ is from Scharlemann et al., (2014), and 1% (0.01) is from Chapin et al., (2011).

The following equation (from Eq. (A38)) was used to calculate $r_{leakage,C}$

$$r_{leakage,C} = \frac{L_C}{NSC} = \frac{0.05 \bullet GPP}{q \bullet C_l} = \frac{0.05 \bullet 13050\left[\frac{kg\,C}{ha\,yr}\right]}{4 \bullet 3.2\left[\frac{kg\,C}{indiv}\right] \bullet 390\left[\frac{indiv}{ha}\right]} = 0.131 \left[\frac{1}{yr}\right] \text{ where 0.05 is from Jones et al., (2009), 13050 kg C ha}^{-1}\text{ yr}^{-1}$$

is from the ForC database (Anderson-Teixeira et al., 2018), 3.2 kg C indiv$^{-1}$ is from the BAAD (Falster et al., 2015), and

~390 indiv ha[-1] is from Crowther et al., (2015) (for temperate forests). Note that we did not distinguish between deciduous angiosperm and evergreen gymnosperm in the BAAD because other data are from all temperate forests.

$r_{growth}$ from LM3-SNAP was adapted to the units of LM4.1

$$r_{growth} = \frac{0.3\left[\frac{kg\,C}{yr\,m^2}\right]\left[\frac{10000m^2}{ha}\right]}{8\left[\frac{kg}{ha}\right]\left[\frac{kg\,C}{2kg}\right],106\left[\frac{kg}{ha}\right]\left[\frac{kg\,C}{2kg}\right],4\left[\frac{kg}{ha}\right]\left[\frac{kg\,C}{2kg}\right],250\left[\frac{kg}{ha}\right]\left[\frac{kg\,C}{2kg}\right]} = 65 \text{ yr}^{-1}$$ where 0.3 kg C yr[-1] m[-2] is from LM3-SNAP, 8, 106 and 4 kg ha[-1] are from Boring and Swank (1984), and 250 kg ha[-1] yr[-1] is from Binkley et al., (1992).

$r_{Nfix}$ was calculated with data from Bytnerowicz et al., (in review) which measured nodule biomass % C and symbiotic BNF
rate per unit nodule biomass using $^{15}N_2$ incorporation.

Parameters related to plant C allocation to symbionts ($f_{alloc,AM}$, $f_{alloc,EM}$, and $f_{alloc,Nfix}$) were estimated by examining a range of values (0.01, 0.02, 0.05, 0.10, 0.20, 0.50, and 1.00). Symbiont biomass C was compared to literature observations (that were not used for model parameterization) from Boring and Swank (1984) for nodule biomass C and Zhu and Miller
(2003) for mycorrhizal biomass C. Parameter values were chosen such that simulated symbiont biomass C was most similar to observed symbiont biomass C.

**Appendix D Figures and Tables**

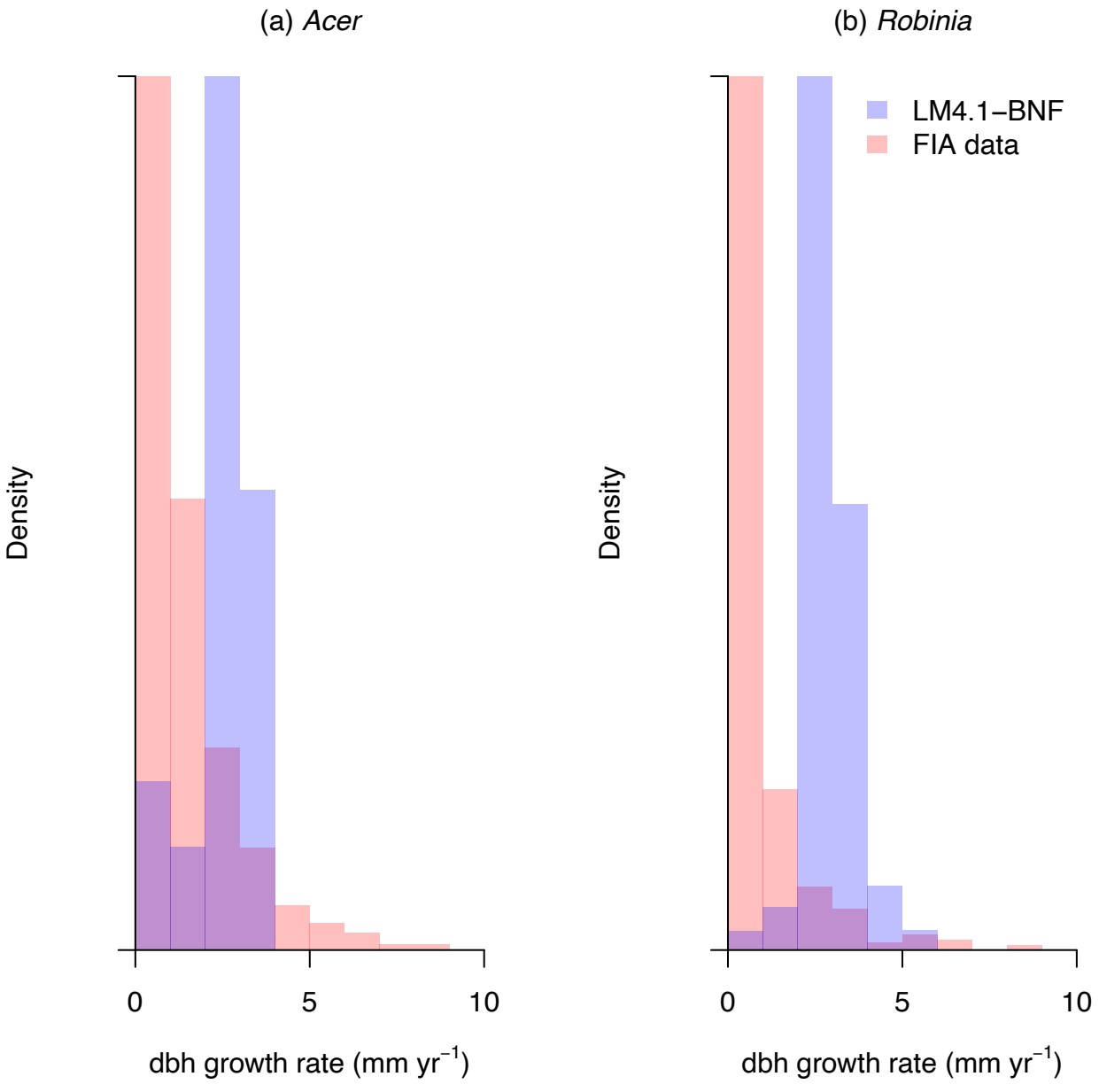

**Figure D1: Simulated dbh growth rate for (a)** *Acer* **/ non-fixer and (b)** *Robinia* **compared to FIA data (in North Carolina).**
**Simulated data are trees with dbh > 12.7 cm to reflect the dbh range of FIA data and are weighted by the stand age distribution of FIA data (Fig. D18). Simulated and FIA data are scaled to display an equal maximum density. LM3-SNAP cannot distinguish between plant cohorts with different dbhs or distinguish between** *Robinia* **and** *Acer***.**

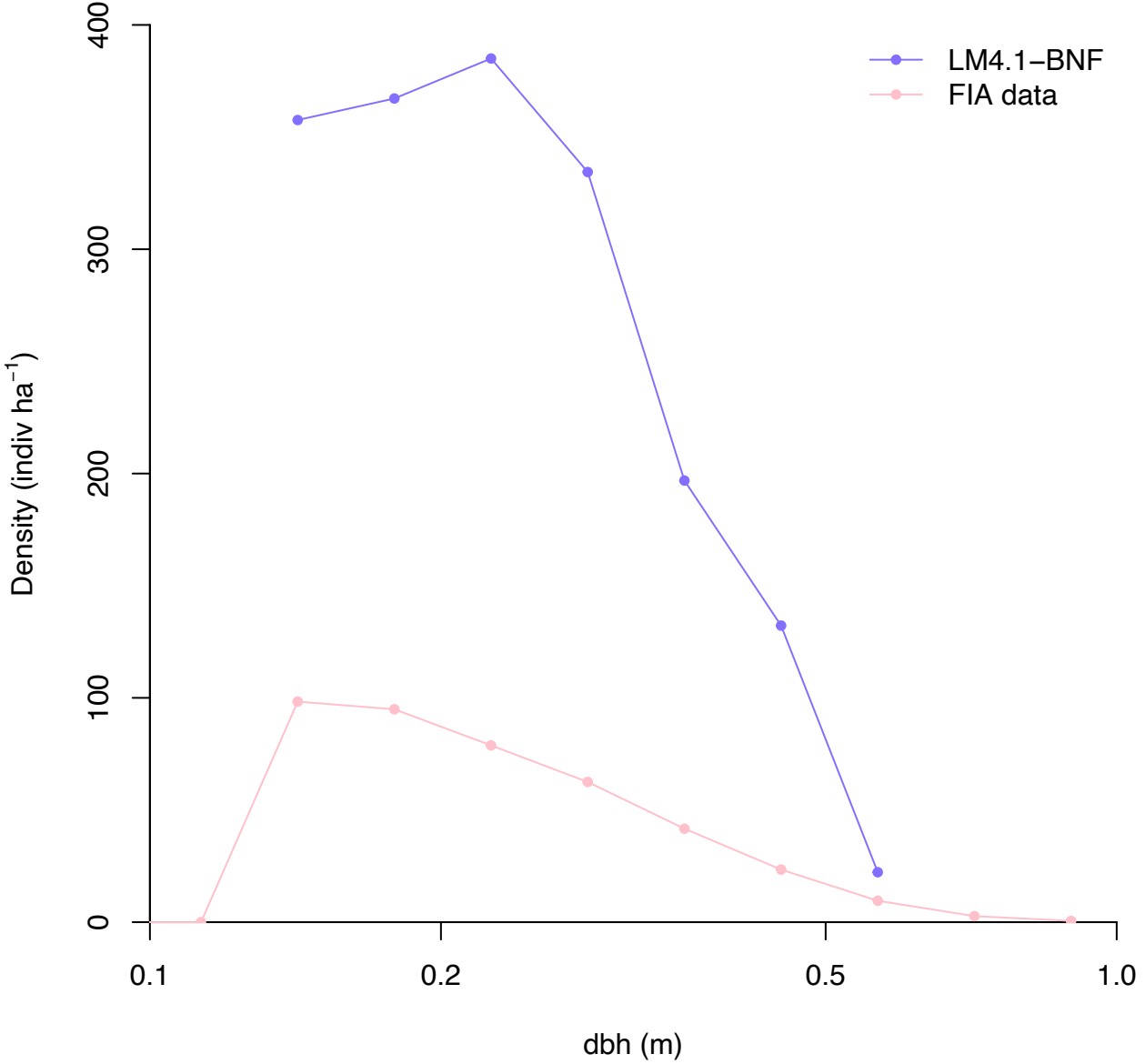

**Figure D2: Simulated dbh distribution compared to FIA data (in North Carolina). Simulated data are trees with dbh > 12.7 cm to**
**reflect the dbh range of FIA data and are weighted by the stand age distribution of FIA data (Fig. D18). LM3-SNAP cannot distinguish between plant cohorts with different dbhs or distinguish between *Robinia* and *Acer*.**

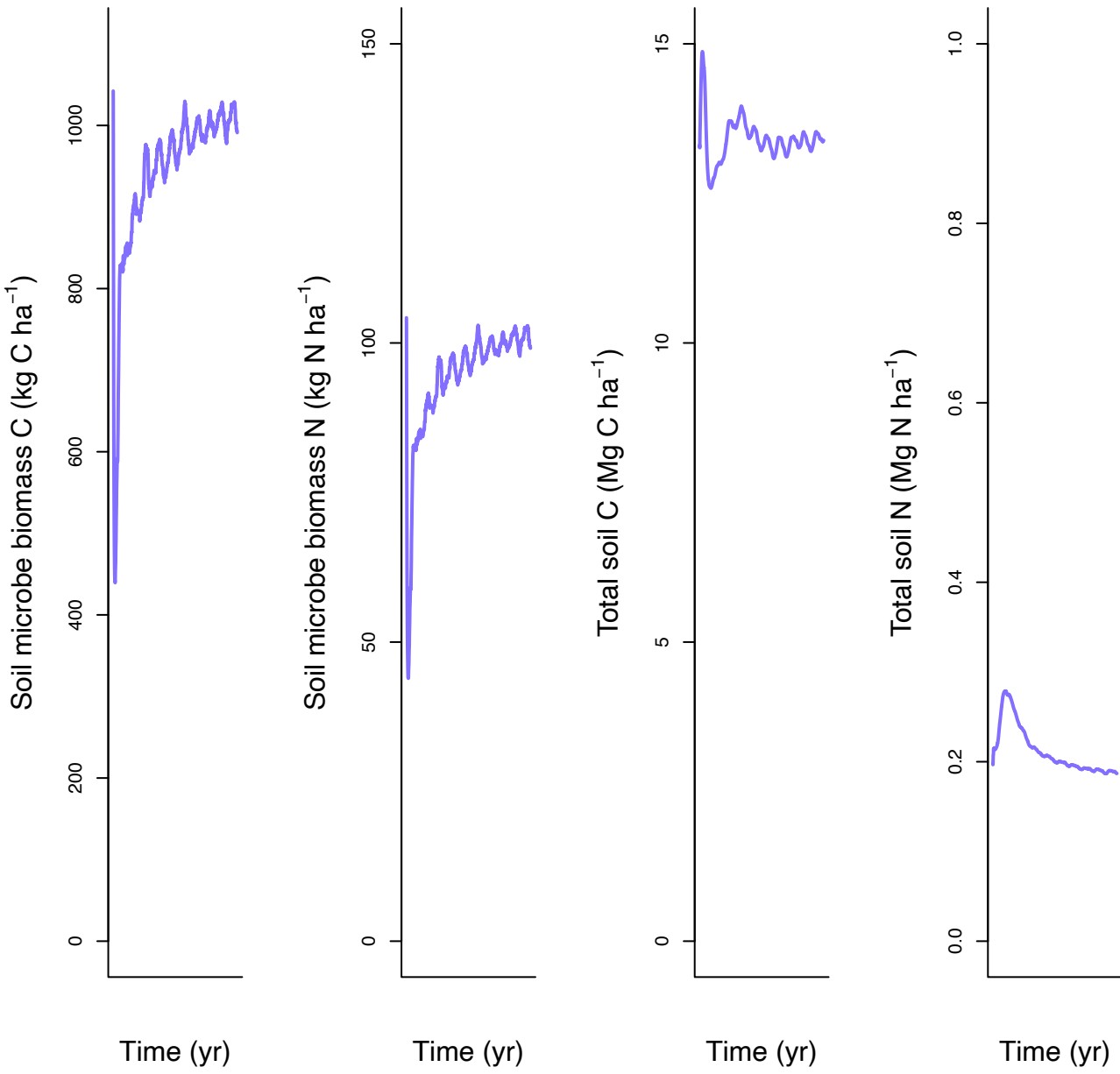

**Figure D3: Controls of asymbiotic BNF. a. Soil microbe biomass C, b. soil microbe biomass N, c. total soil C, and d. total soil N.**

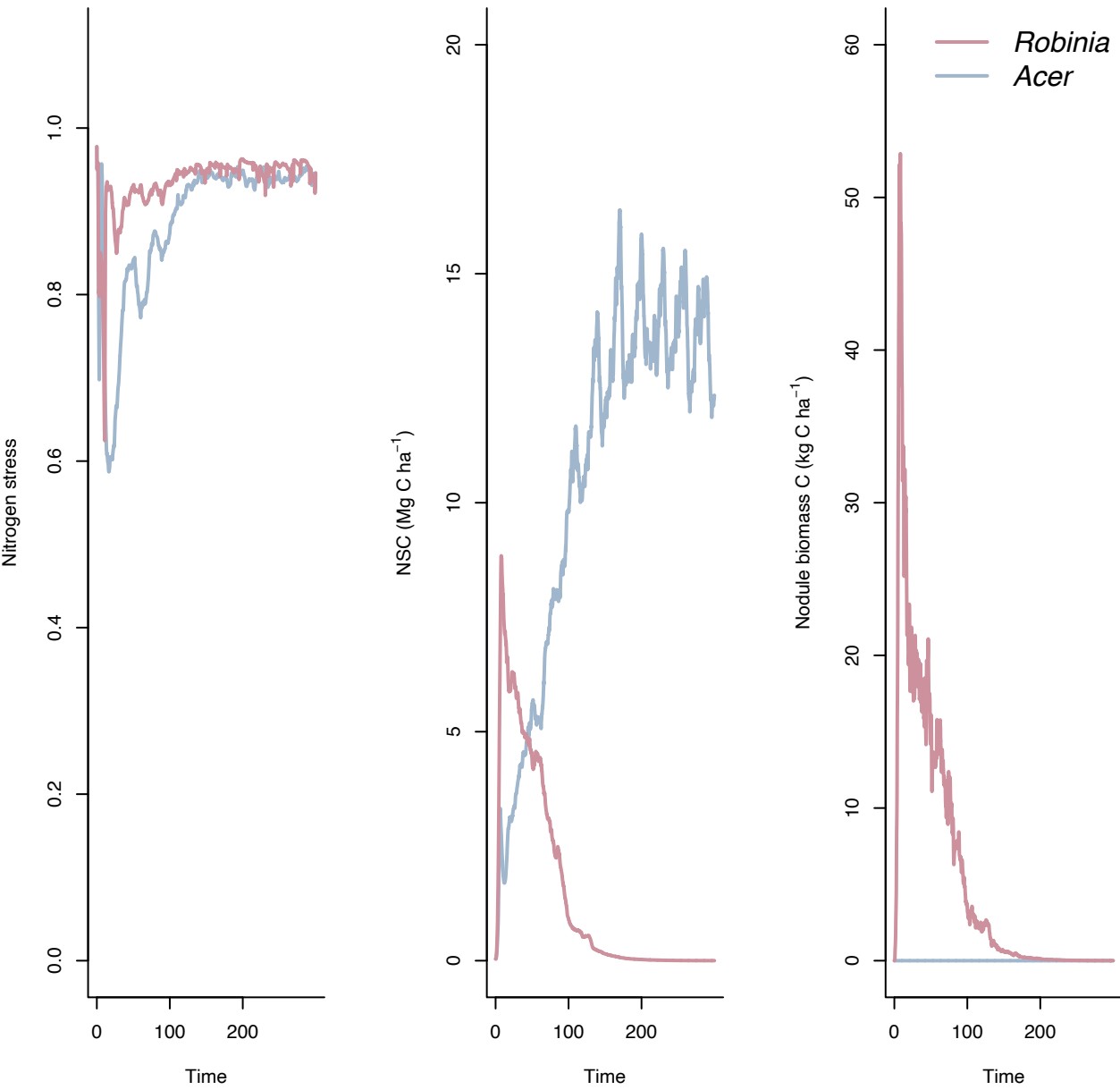

**Figure D4: Controls of symbiotic BNF. a. Nodule biomass C, b. non-structural C, and c. N stress.**

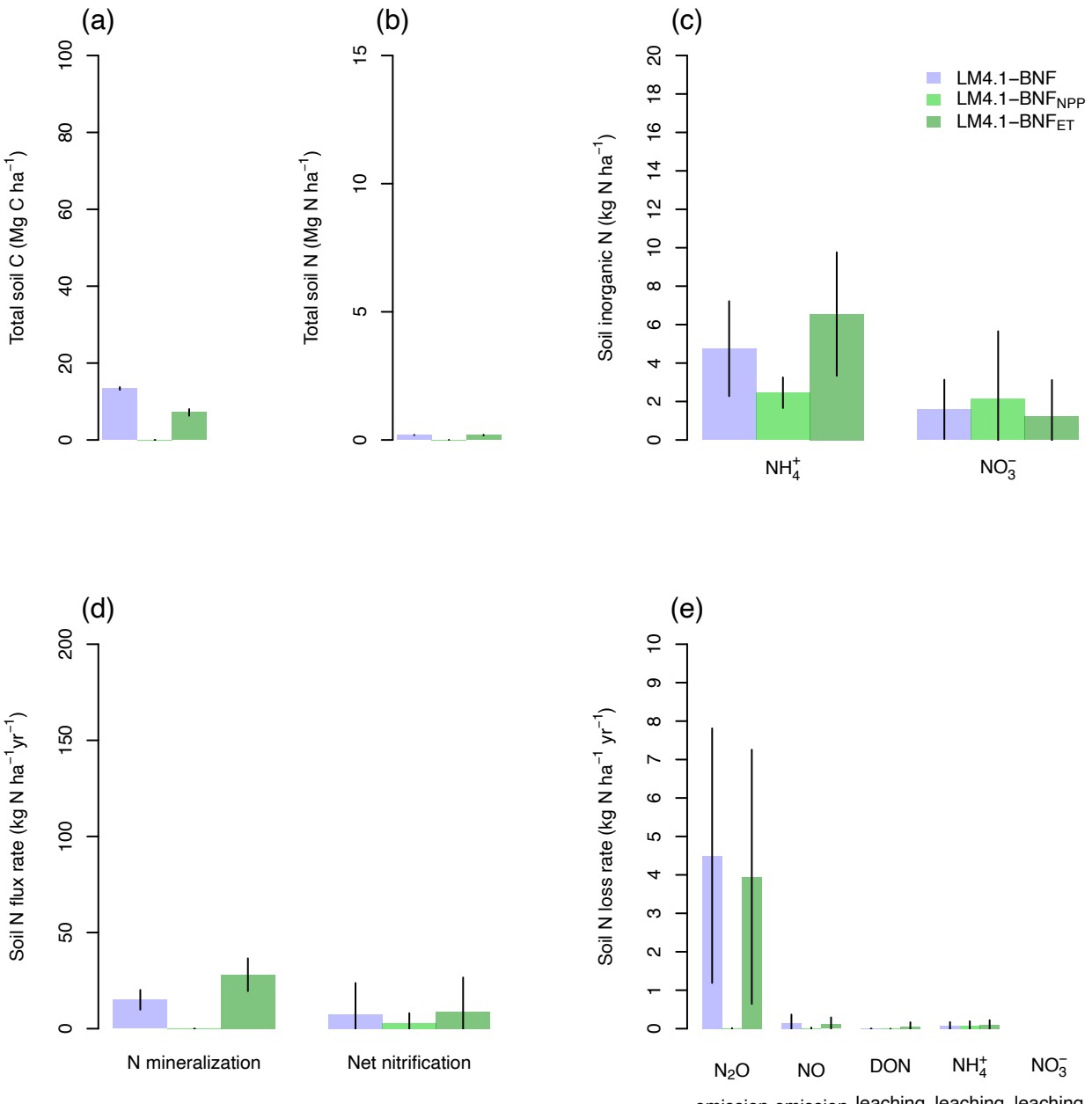

**Figure D5: Simulated soil C and N pools, soil N fluxes, and soil N loss rates from LM4.1-BNF, LM4.1-BNF$_{NPP}$, and LM4.1-BNF$_{ET}$.** (a) Simulated total soil C (depth 0-10 cm). (b) Simulated total soil N (depth 0-10 cm). (c) Simulated soil NH$_4^+$ and NO$_3^-$ (depth 0-10 cm). (d) N mineralization rate and net nitrification rate (depth 0-10 cm). (e) Simulated N$_2$O and NO emission rate and simulated dissolved organic N (DON), NH$_4^+$, and NO$_3^-$ leaching rate. Simulated data are averaged over the last 100 yr of the 300 yr simulation to reflect the site data which is from mature forests. Error bars indicate two standard deviations.


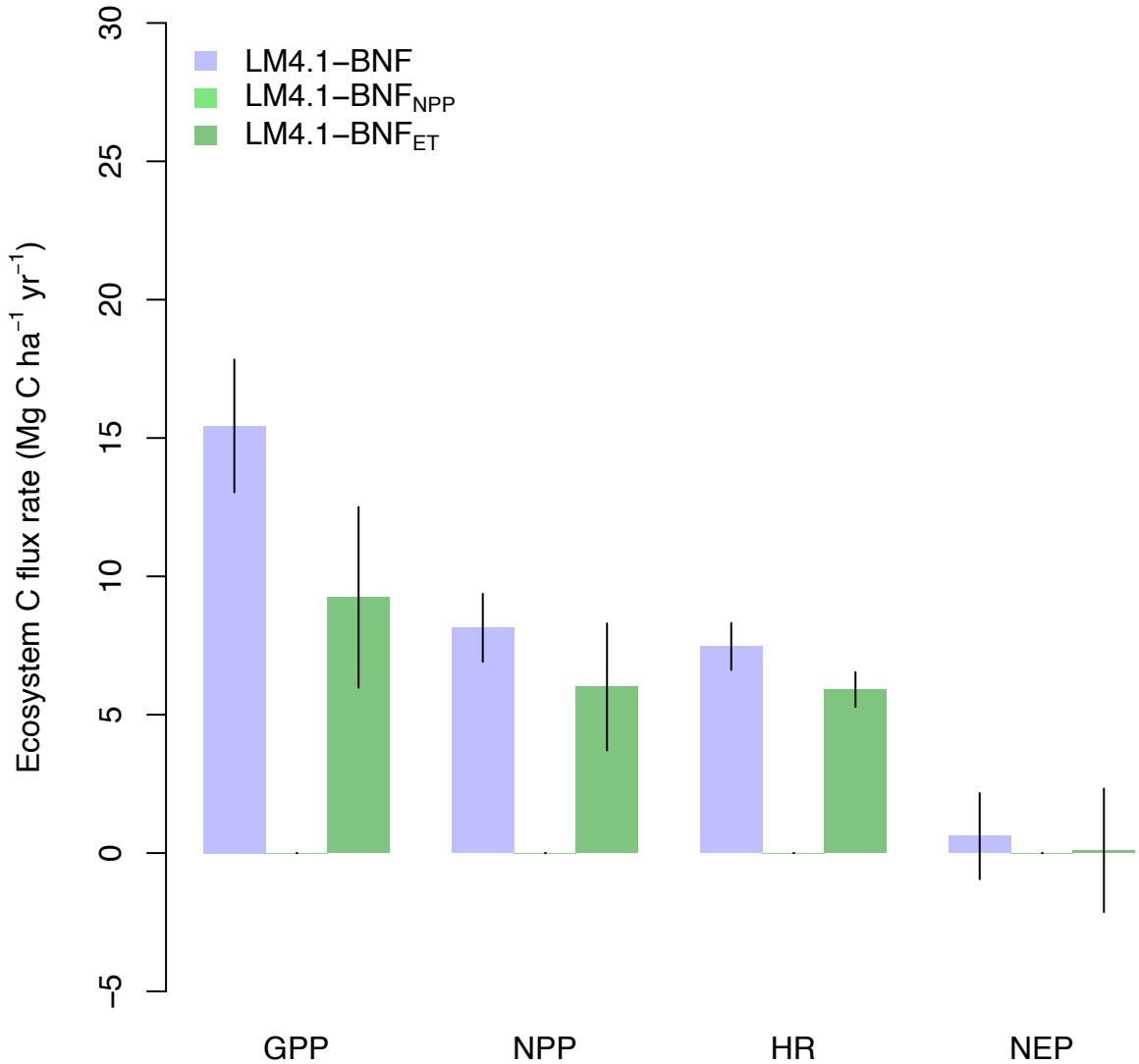

**Figure D6: Simulated gross primary production (GPP), net primary production (NPP), heterotrophic respiration (HR), and net ecosystem production (NEP) from LM4.1-BNF, LM4.1-BNF$_{NPP}$, and LM4.1-BNF$_{ET}$. Simulated data are averaged over the last 100 yr of the 300 yr simulation to reflect the data which is from mature forests. Error bars indicate two standard deviations.**

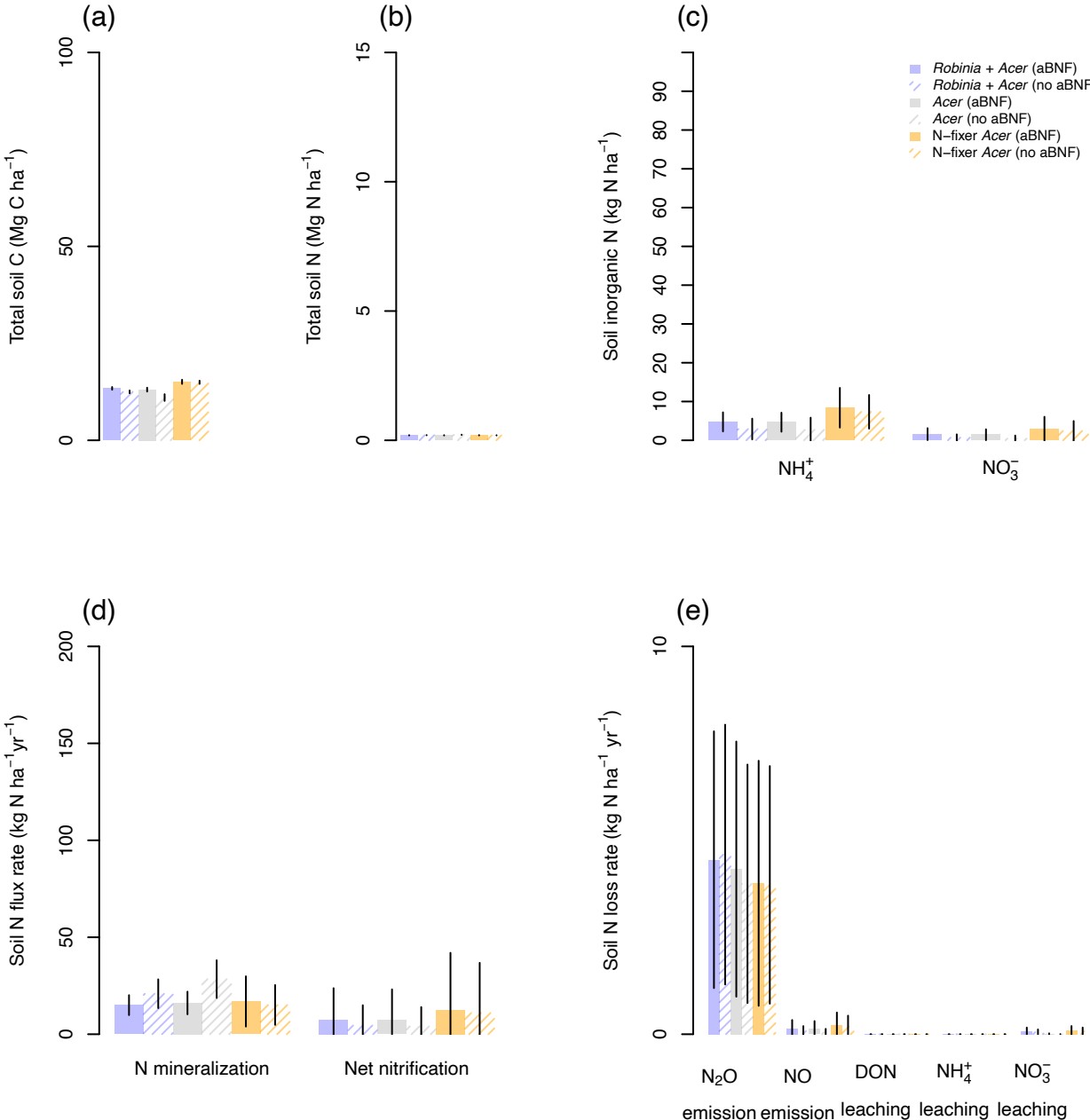

**Figure D7: Simulated soil C and N pools, soil N fluxes, and soil N loss rates from LM4.1-BNF initialised with both *Robinia* and *Acer*, only *Acer*, and only N-fixer *Acer*, with and without asymbiotic BNF. (a) Simulated total soil C (depth 0-10 cm). (b) Simulated total soil N (depth 0-10 cm). (c) Simulated soil $NH_4^+$ and $NO_3^-$ (depth 0-10 cm). (d) N mineralization rate and net nitrification rate (depth 0-10 cm). (e) Simulated $N_2O$ and NO emission rate and simulated dissolved organic N (DON), $NH_4^+$, and $NO_3^-$ leaching rate. Simulated data are averaged over the last 100 yr of the 300 yr simulation to reflect the site data which is from mature forests. Error bars indicate two standard deviations.**


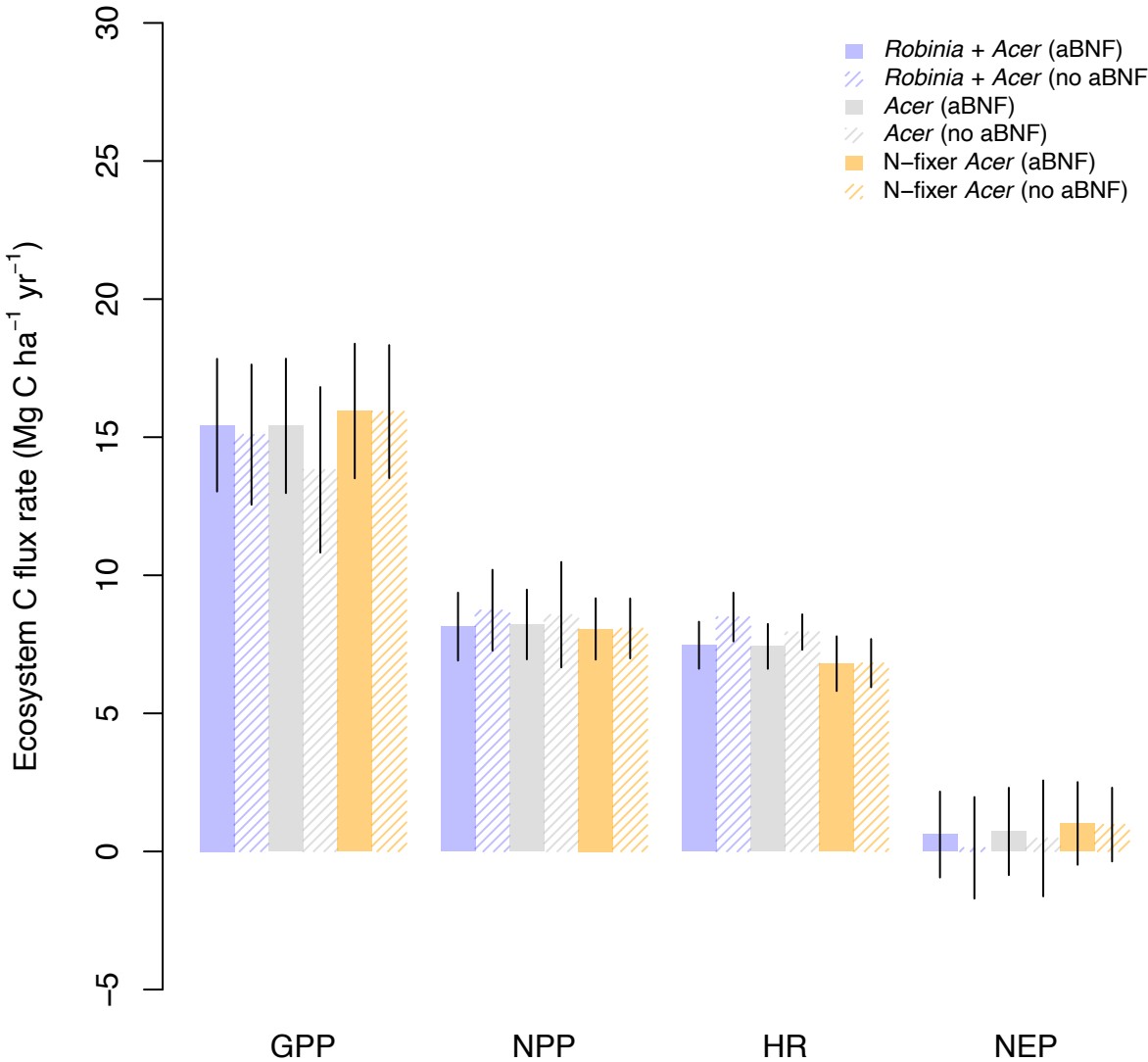

**Figure D8: Simulated gross primary production (GPP), net primary production (NPP), heterotrophic respiration (HR), and net ecosystem production (NEP) from LM4.1-BNF initialised with both _Robinia_ and _Acer_, only _Acer_, and only N-fixer _Acer_, with and without asymbiotic BNF. Simulated data are averaged over the last 100 yr of the 300 yr simulation to reflect the data which is from mature forests. Error bars indicate two standard deviations.**

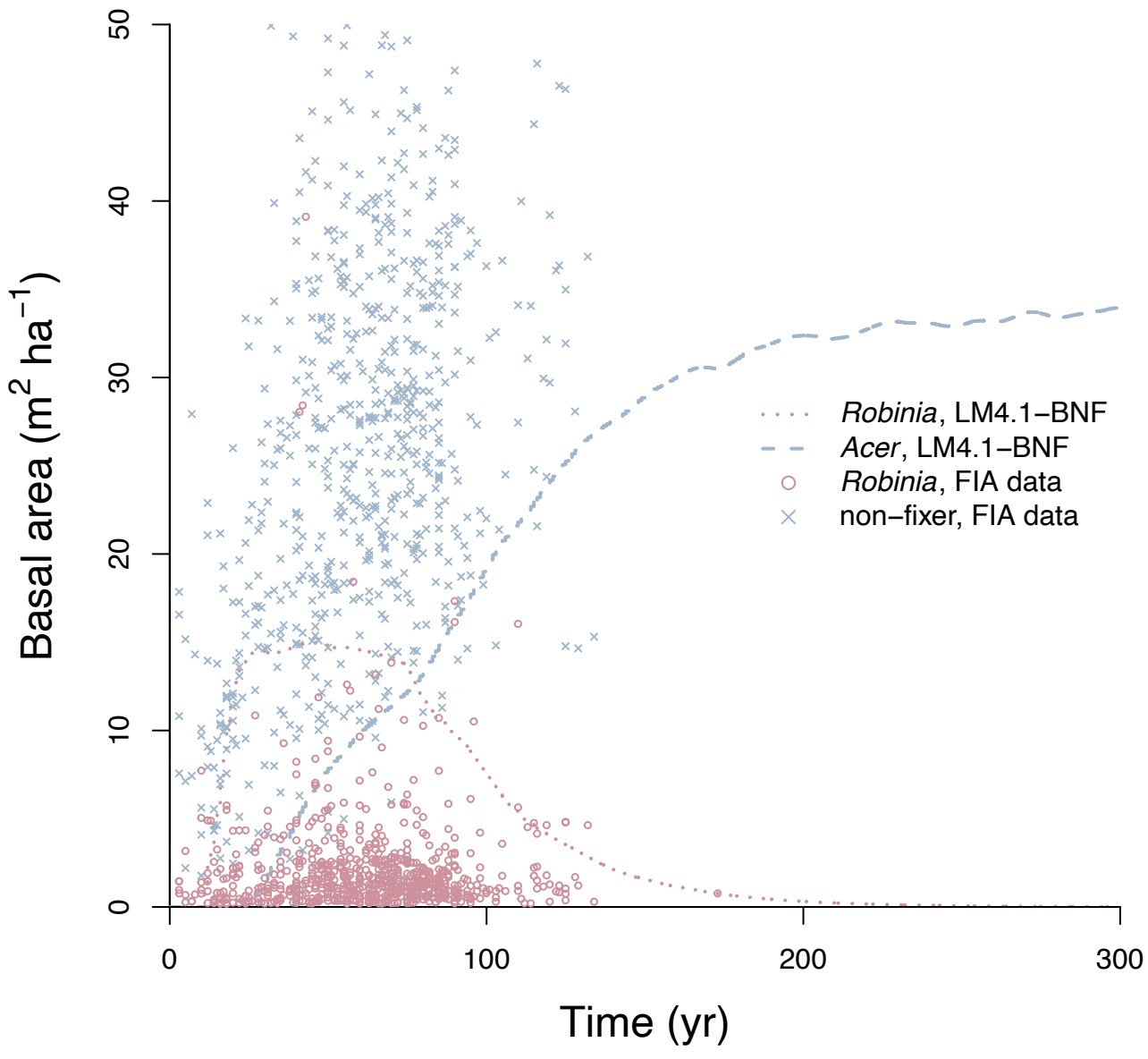

**Figure D9: Simulated absolute basal area of *Acer* and *Robinia* over time compared to FIA data (in North Carolina). Simulated data are trees with dbh > 12.7 cm to reflect the dbh range of FIA data. FIA data of all non-fixing trees are aggregated to represent *Acer*. Each point represents an FIA plot.**

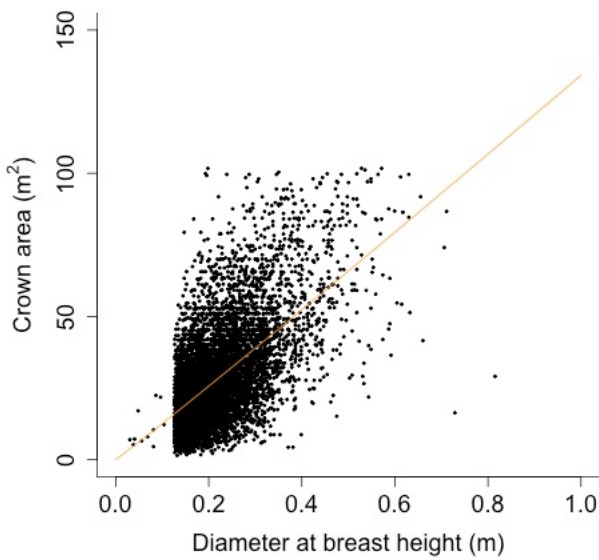

**Figure D10: Crown area model fit to FIA Forest Health Monitoring data for *Acer*.**

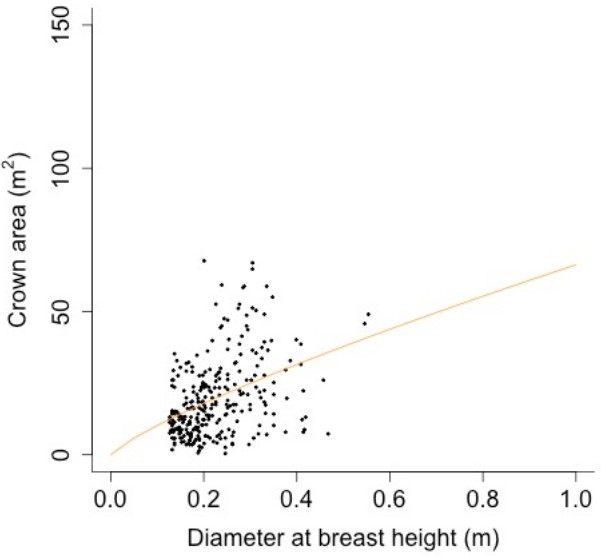

**Figure D11: Crown area model fit to FIA Forest Health Monitoring data for *Robinia*.**

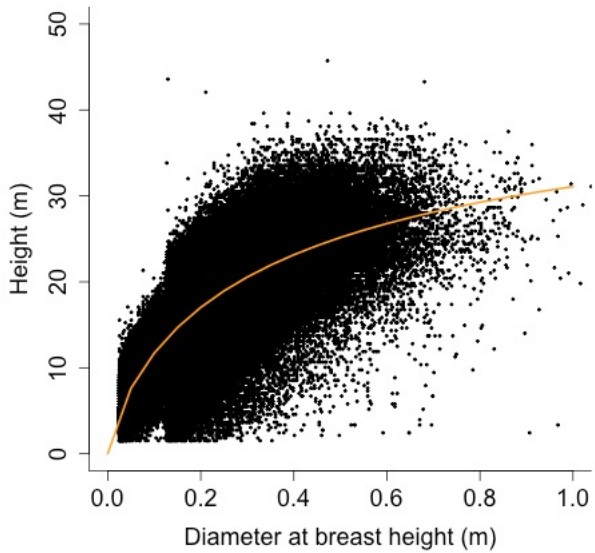

**Figure D12: Height model fit to FIA data for *Acer*.**

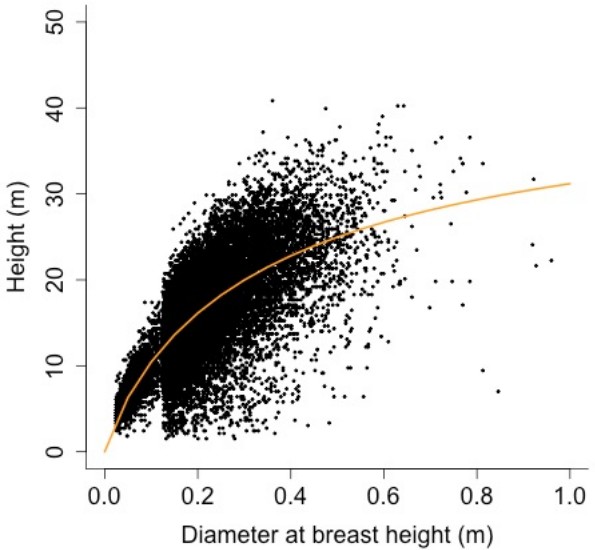

**Figure D13: Height model fit to FIA data for *Robinia*.**

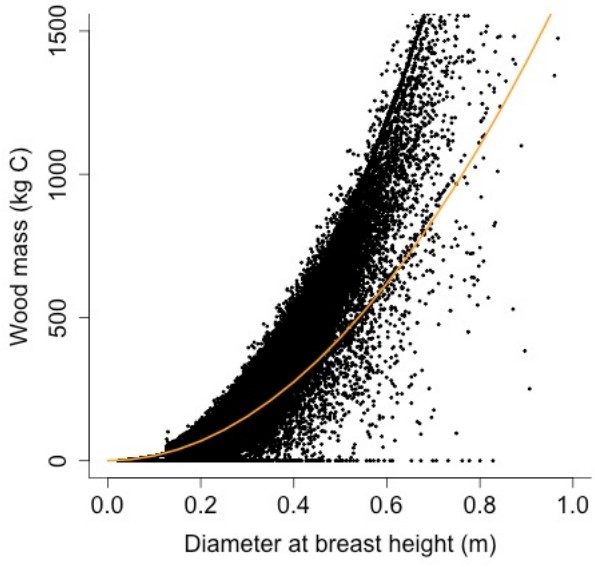


**Figure D14: Wood mass model fit to FIA data for *Acer* (displayed at mean height of *Acer*).**

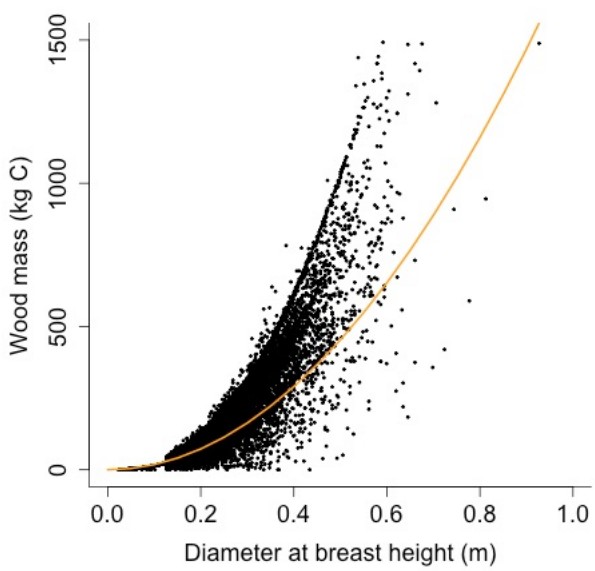

**Figure D15: Wood mass model fit to FIA data for *Robinia* (displayed at mean height of *Robinia*).**

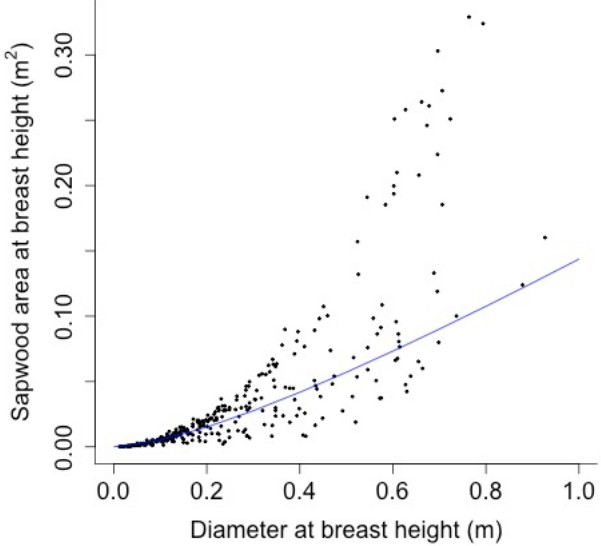

**Figure D16: Sapwood area at breast height model fit to BAAD for *Acer*.**

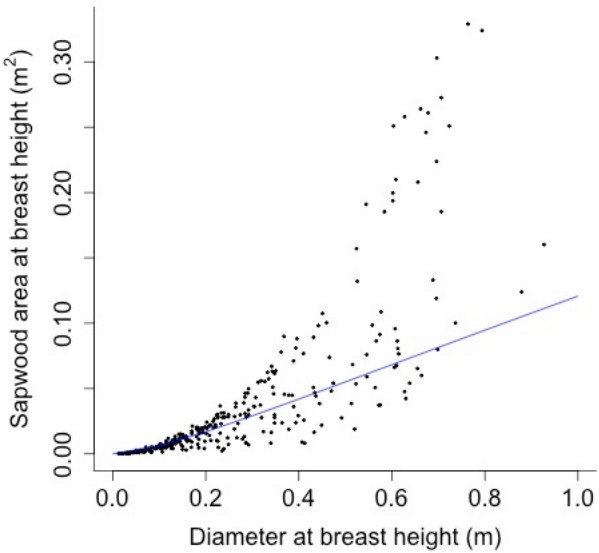

**Figure D17: Sapwood area at breast height model fit to BAAD for *Robinia*.**

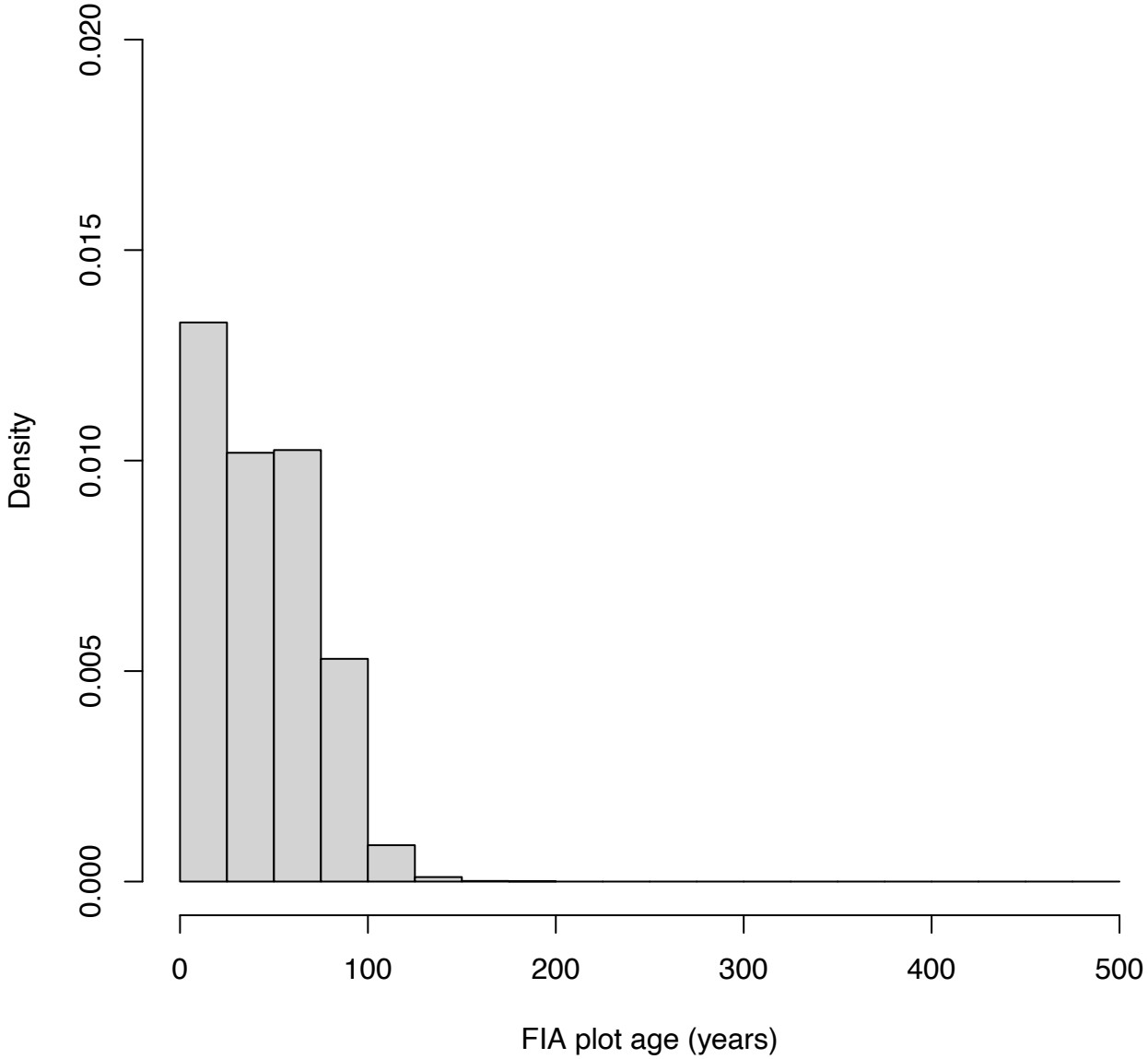

**Figure D18: Histogram of FIA stand age for FIA plots in North Carolina. These distributions were used to weigh the data displayed in Fig. 2 and 3.**

**Table D1: Vegetation type-specific parameters.**

| Parameter | Vegetation type | Value | Unit | Source |
|---|---|---|---|---|
| $C{:}N_l$ | *Acer* | 30 | kg C kg N$^{-1}$ | TRY database |

| | Robinia | 14 | | |
|---|---|---|---|---|
| $\rho_{wood}$ | Acer | 340 | kg C m$^{-3}$ | TRY database |
| | Robinia | 280 | | |
| LMA | Acer | 0.0482 | kg C m$^{-2}$ | TRY database |
| | Robinia | 0.0380 | | |
| $V_{cmax}$ at 15°C | Acer | 17 | μmol m$^{-2}$ s$^{-1}$ | TRY database, converted to 15°C using method from Medlyn et al., 2002 |
| | Robinia | 23 | | |
| $\alpha_{HT}$ | Acer | 46.175656 | m | see Appendix C |
| | Robinia | 43.87161 | | |
| $\theta_{HT}$ | Acer | 0.782971 | unitless | see Appendix C |
| | Robinia | 0.89594 | | |
| $\gamma_{HT}$ | Acer | 0.485517 | unitless | see Appendix C |
| | Robinia | 0.40675 | | |
| $\alpha_{BM}$ | Acer | 0.3922393 | unitless | see Appendix C |
| | Robinia | 0.3383768 | | |
| $\alpha_{CA}$ | Acer | 134.18322 | unitless | see Appendix C |
| | Robinia | 66.3140 | | |
| $\theta_{CA}$ | Acer | 1.02731 | unitless | see Appendix C |
| | Robinia | 0.8128 | | |
| $\varphi_{CSA}$ | Acer | 0.21346 | m$^{-1}$ | see Appendix C |
| | Robinia | 0.16983 | | |
| $\mu_C$ | Acer | 0.01810868 | yr$^{-1}$ | see Appendix C |
| | Robinia | 0.03105176 | | |
| $\mu_U$ | Acer | 0.04024044 | yr$^{-1}$ | see Appendix C |
| | Robinia | 0.08785878 | | |
| $r_{Nfix}$ | Robinia | 6.3 | kg N kg C$^{-1}$ yr$^{-1}$ | see Appendix C |

**Table D2: General parameters.**

| Parameter | Value | Unit | Source |
|---|---|---|---|
| N uptake by roots and symbionts | | | |
| $r_{NO_3}$ | 0.1 | kg N m$^{-3}$ yr$^{-1}$ | Sulman et al., 2019 |
| $k_{M,NO_3}$ | 0.005 | kg N m$^{-3}$ | Sulman et al., 2019 |
| $r_{NH_4}$ | 0.1 | kg N m$^{-3}$ yr$^{-1}$ | Sulman et al., 2019 |
| $k_{M,NH_4}$ | 0.005 | kg N m$^{-3}$ | Sulman et al., 2019 |
| $\Delta z(k)$ | 0.02, 0.04, 0.04, 0.05, 0.05, 0.10, 0.10, 0.20, 0.20, 0.20, 0.40, 0.40, 0.40, 0.40, 0.40, 1.0, 1.0, 1.0, 1.5, 2.5 | m | Sulman et al., 2019 |
| $r_{root}$ | 0.00029 | m | Jackson et al., 1997 |
| SRL | 24545 | m kg C$^{-1}$ | Jackson et al., 1997 |
| SRA | 45 | m$^2$ kg C$^{-1}$ | Jackson et al., 1997 |
| $r_{rhiz}$ | 0.001 | m | Sulman et al., 2019 |
| $r_{NO_3,AM}$ | 0.1 | kg N m$^{-3}$ yr$^{-1}$ | Sulman et al., 2019 |
| $k_{AM,NO_3}$ | 0.005 | kg N m$^{-3}$ | Sulman et al., 2019 |
| $r_{NH_4,AM}$ | 0.1 | kg N m$^{-3}$ yr$^{-1}$ | Sulman et al., 2019 |

| | | | |
|---|---|---|---|
| $k_{AM,NH_4}$ | 0.005 | kg N m$^{-3}$ | Sulman et al., 2019 |
| $k_{AM}$ | 0.3 | kg C m$^{-3}$ | Sulman et al., 2019 |
| $V_{EM,max,ref,i}$ | 2, 0.3, 2 | yr$^{-1}$ | Sulman et al., 2019 |
| $E_{a,i}$ | 6000, 40000, 6000 | kJ mol$^{-1}$ | Sulman et al., 2019 |
| $R$ | 8.314472 | kJ mol$^{-1}$ K$^{-1}$ | - |
| $T_{ref}$ | 293.15 | K | - |
| $k_{M.EM}$ | 0.015 | unitless | Sulman et al., 2019 |
| $\varepsilon_{C,i,EM}$ | 0.1, 0.05, 0.1 | unitless | Sulman et al., 2019 |
| $\varepsilon_{N,i,EM}$ | 0.7, 0.7, 0.7 | unitless | Sulman et al., 2019 |
| $r_{NO_3,EM}$ | 0.2 | kg N m$^{-3}$ yr$^{-1}$ | Sulman et al., 2019 |
| $k_{EM,NO_3}$ | 0.001 | kg N m$^{-3}$ | Sulman et al., 2019 |
| $r_{NH_4,EM}$ | 0.2 | kg N m$^{-3}$ yr$^{-1}$ | Sulman et al., 2019 |
| $k_{EM,NH_4}$ | 0.001 | kg N m$^{-3}$ | Sulman et al., 2019 |
| $k_{EM}$ | 0.3 | kg C m$^{-3}$ | Sulman et al., 2019 |
| Asymbiotic BNF | | | |
| $V_{max,ref,i}$ | 18, 0.2, 4.5 | yr$^{-1}$ | Sulman et al., 2019 |
| $k_M$ | 0.045 | unitless | Sulman et al., 2019 |
| $\theta_{sat}$ | 0.439 | m$^3$ m$^{-3}$ | Sulman et al., 2019 |
| $V_{denit,max,ref,i}$ | 0.018, 0.00025, 0.0045 | yr$^{-1}$ | Sulman et al., 2019 |
| $k_{M,denit}$ | 0.045 | unitless | Sulman et al., 2019 |
| $k_{denit}$ | 0.0027 | yr | Sulman et al., 2019 |
| $f_{denit}$ | 0.93 | kg N kg C$^{-1}$ | Sulman et al., 2019 |
| $\tau_M$ | 0.25 | yr | Sulman et al., 2019 |
| $\varepsilon_{t,M}$ | 0.6 | unitless | Sulman et al., 2019 |
| $C:N_M$ | 10.0 | kg C kg N$^{-1}$ | Chapin et al., 2011 |
| $r_{Nfix\ asymb}$ | 0.024 | kg N kg C$^{-1}$ yr$^{-1}$ | see Appendix C |
| $\varepsilon_{N,i}$ | 0.7, 0.4, 0.7 | unitless | Sulman et al., 2019 |
| $\varepsilon_{C,i}$ | 0.6, 0.05, 0.6 | unitless | Sulman et al., 2019 |
| $V_{max,ref,NH_4}$ | 365 | yr$^{-1}$ | Sulman et al., 2019 |
| $E_{a,NH_4}$ | 37000 | kJ mol$^{-1}$ | Sulman et al., 2019 |
| $V_{max,ref,NO_3}$ | 365 | yr$^{-1}$ | Sulman et al., 2019 |
| $E_{a,NO_3}$ | 37000 | kJ mol$^{-1}$ | Sulman et al., 2019 |
| Plant growth and N limitation | | | |
| $C:N_r$ | 42 | kg C kg N$^{-1}$ | Roumet et al., 2016 |
| $C:N_{sw}$ | 287 | kg C kg N$^{-1}$ | Meerts, 2002 |
| $C:N_{hw}$ | 427 | kg C kg N$^{-1}$ | Meerts, 2002 |
| $LAI_{max}$ | 5.1 | m$^2$ m$^{-2}$ | Asner et al., 2003 |
| $\varphi_{RL}$ | 0.79 | unitless | see Appendix C |
| $f_{br}$ | 0.255 | unitless | see Appendix C |
| $q$ | 4 | unitless | Weng et al., 2015 |
| $r_{leakage,C}$ | 0.131 | yr$^{-1}$ | see Appendix C |
| $r_{leakage,N}$ | 0.15 | yr$^{-1}$ | Sulman et al., 2019 |
| Plant C allocation to symbionts | | | |
| $f_{alloc,AM}$ | 0.01 | yr$^{-1}$ | estimated (see Appendix C) |
| $f_{alloc,EM}$ | 0.01 | yr$^{-1}$ | estimated (see Appendix C) |

| | | | |
|---|---|---|---|
| $f_{alloc,Nfix}$ | 0.1 | yr$^{-1}$ | estimated (see Appendix C) |
| $f_{alloc,Nfix,min}$ | 0.05 | kg C indiv$^{-1}$ yr$^{-1}$ | D. N. L. Menge, unpublished data |
| $C{:}N_{alloc}$ | 1000 | kg C kg N$^{-1}$ | Sulman et al., 2019 |
| Growth and turnover of symbionts | | | |
| $\varepsilon_{symb}$ | 0.666 | unitless | Sulman et al., 2019 (for plant growth) |
| $r_{growth}$ | 65 | yr$^{-1}$ | see Appendix C |
| $\xi_{AM}$ | 1.25 | yr$^{-1}$ | Sulman et al., 2019 (for fine roots) |
| $\xi_{EM}$ | 1.25 | yr$^{-1}$ | Sulman et al., 2019 (for fine roots) |
| $\xi_{Nfix}$ | 1.25 | yr$^{-1}$ | Sulman et al., 2019 (for fine roots) |
| $cost_{Nfix}$ | 4.8 | kg C kg N$^{-1}$ | Gutschick et al., 1981 |
| $\tau_{AM}$ | 2.0 | yr | Sulman et al., 2019 (for fine roots) |
| $\tau_{EM}$ | 2.0 | yr | Sulman et al., 2019 (for fine roots) |
| $\tau_{Nfix}$ | 2.0 | yr | Sulman et al., 2019 (for fine roots) |
| $C{:}N_{AM}$ | 10 | kg C kg N$^{-1}$ | Johnson, 2010 |
| $C{:}N_{EM}$ | 14 | kg C kg N$^{-1}$ | Zhang and Elser, 2017 |
| $C{:}N_{Nfix}$ | 6 | kg C kg N$^{-1}$ | Boring and Swank, 1984 |
| $r_{up,veg}$ | 250 | yr$^{-1}$ | Sulman et al., 2019 |
| Soil N$_2$O and NO emissions | | | |
| $V_{nit,max,ref}$ | 5 | yr$^{-1}$ | Sulman et al., 2019 |
| $E_{a,nit}$ | 37000 | kJ mol$^{-1}$ | Sulman et al., 2019 |
| LM4.1-BNF$_{NPP}$ | | | |
| $a_{NPP}$ | 0.0018 | kg N m$^{-2}$ yr$^{-1}$ | Meyerholt et al., 2016 |
| $b_{NPP}$ | -3 | m$^2$ yr kg C$^{-1}$ | Meyerholt et al., 2016 |
| LM4.1-BNF$_{ET}$ | | | |
| $a_{ET}$ | 2.34e-6 | kg N mm$^{-1}$ m$^{-2}$ | Meyerholt et al., 2016 |
| $b_{ET}$ | -1.72e-5 | kg N m$^{-2}$ yr$^{-1}$ | Meyerholt et al., 2016 |

**Table D3: Summary of the spin up and numerical experiments.**

| | Spin up | Numerical experiments |
|---|---|---|
| Atmospheric CO$_2$ concentration | Pre-industrial: 284.26 ppm (Meinshausen et al., 2017) | mean 1948 – 1978: 324.53 ppm (Dlugokencky and Tans, 2020) |
| Meteorological forcing | 1948-1978 (Sheffield et al., 2006) | 1948-1978 (Sheffield et al., 2006) |
| N deposition rate | 1993 (Dentener, 2006) | 1993 (Dentener, 2006) |

**Table D4: Initial densities and heights for simulations.** The initial densities of *Robinia* and *Acer* were derived from the US FIA database seedling data for plots with at least one *Robinia* individual in North Carolina. Diameter at breast height is determined from height by allometry (Eq. (A41)).

| BNF representation | Initialised species | Density | Height | Diameter at breast height |
|---|---|---|---|---|
| LM4.1-BNF | *Acer* | 0.54 indiv m$^{-2}$ | 0.5 m | 0.00160 m |
| | *Robinia* | 0.13 indiv m$^{-2}$ | 0.5 m | 0.00336 m |
| LM4.1-BNF | *Acer* | 0.5 indiv m$^{-2}$ | 0.5 m | 0.00160 m |
| LM4.1-BNF$_{NPP}$ | *Acer* | 0.5 indiv m$^{-2}$ | 0.5 m | 0.00160 m |
| LM4.1-BNF$_{ET}$ | *Acer* | 0.5 indiv m$^{-2}$ | 0.5 m | 0.00160 m |


**Table D5: Validated variables and data sources.**

| Variable | Data source for Coweeta Hydrologic Laboratory |
|---|---|
| dbh growth rate of each vegetation type | FIA database tree data, North Carolina plots with at least one *Robinia* individual (US Forest Service, 2020a) |
| dbh distribution | FIA database tree data, North Carolina plots with at least one *Robinia* individual (US Forest Service, 2020a) |
| Basal area fraction of each vegetation type | FIA database tree data, North Carolina plots with at least one *Robinia* individual (US Forest Service, 2020a) |
| Total plant biomass C | FIA database tree data, North Carolina plots with at least one *Robinia* individual (US Forest Service, 2020a) |
| Asymbiotic BNF rate | Todd et al., 1978 |
| Symbiotic BNF rate | Boring and Swank, 1984 |
| Total soil C and N | FIA database soil data, North Carolina plots (US Forest Service, 2020a) |
| | Knoepp, 2009a, 2018 |
| Soil NH$_4^+$ and NO$_3^-$ | Knoepp, 2009a, 2018 |
| N mineralization and net nitrification rates | Knoepp, 2009b, 2009a |
| N$_2$O and NO emission rates | Stehfest and Bouwman, 2006 |
| DON, NH$_4^+$, and NO$_3^-$ leaching rate | Swank and Waide, 1988 |
| Gross primary production, heterotrophic respiration, net primary production, and net ecosystem production | Anderson-Teixeira et al., 2018 |

**Code availability**

Please contact sk4220@columbia.edu for code. Full GFDL land model code is available upon request through the GFDL.

**Author contribution**

S.K.-G. designed and implemented LM4.1-BNF in collaboration with D.M, S.M., and E.S.. I.M.C., S.P., and T.B. advised on the new model. S.M. and E.S. developed the initial implementation of LM4.1 with LM3-SNAP-based N cycling. S.K.-G. conducted the analysis and wrote the initial draft. All authors contributed to the writing.

## Competing interests

The authors declare that they have no conflict of interest.

## Acknowledgements

We thank Ben Sulman for initial assistance with the model. We thank Anika Staccone for assistance with the US FIA database. We thank the Menge lab and the Soper lab for their comments on the manuscript. Data from several databases were used (Appendix B). This study was supported by the National Science Foundation, award NSF DEB-1457650. We acknowledge the support of the Natural Sciences and Engineering Research Council of Canada. Cette recherche a été financée par le Conseil de recherches en sciences naturelles et en génie du Canada.

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
