# Peer review of "A novel representation of biological nitrogen fixation and competitive dynamics between nitrogen-fixing and non-fixing plants in a land model (GFDL LM4.1-BNF)"

_Biogeosciences, 2020_

## Author Comment (AC1)

Kou-Giesbrecht et al., presented a new version of GFDL land model that considers symbiotic fixation, competitive interactions between fixer and non-fixer plants. The model development is convincing and well explained, while the following up model evaluation needs to be improved, more discussions are needed to clarify the simulated C/N dynamics. Below are my major comments and some specific suggestions.

1. Since the major development is N2 fixation, it will be good to emphasize the model evaluation more on N2 fixation. Figure 6 showed that the LM4.1-BNF capture asymbiotic fixation rate for the mature forests, however, the LM4.1-BNF performed worse than LM3-SNAP (or maybe equal) for simulated symbiotic BNF rate along with the forest successional development.

One strategy could be using a part of the data to first tune the model parameters that are related to N2 fixation processes (e.g., rNfix) and then use the rest for evaluation purposes.

A deep dive into LM4.1-BNF simulated N2 fixation rate is further needed. For example, what are the temporal dynamics of Nodule biomass, NSC, Nstress, soil microbe biomass, how they drive the changes of symbiotic and asymbiotic BNF?

*AUTHORS' RESPONSE:*
*This is an excellent suggestion. We have rewritten Section 4 to emphasise BNF. The new version of Fig. 6b (below) shows similar simulated symbiotic BNF rates by LM4.1-BNF and LM3-SNAP in early succession but a substantial difference between simulated symbiotic BNF rates by LM4.1-BNF and LM3-SNAP in late succession: LM3-SNAP simulates a mean symbiotic BNF rate over the final 100 years of 8 kg N ha$^{-1}$ yr$^{-1}$, whereas LM4.1-BNF simulates a mean symbiotic BNF rate over the final 100 years of 0 kg N ha$^{-1}$ yr$^{-1}$ which reflects observations at Coweeta Hydrologic Laboratory of 0 kg N ha$^{-1}$ yr$^{-1}$ (Boring and Swank, 1984). This is extremely important because overestimation of symbiotic BNF in late succession by LM3-SNAP causes several problems. First, LM3-SNAP simulates lower total plant biomass C than LM4.1-BNF (mean 44 vs. 173 Mg C ha$^{-1}$ over the final 100 years) because of the sustained high C cost of symbiotic BNF. Observations suggest that total plant biomass C is 160 Mg C ha$^{-1}$. Second, LM3-SNAP overestimates N losses in comparison to LM4.1-BNF, especially NO$_3^-$ leaching rate (mean 8.3 vs. 0.1 kg N ha$^{-1}$ yr$^{-1}$ over the final 100 years). Observations suggest that NO$_3^-$ leaching rate is 0.1 kg N ha$^{-1}$ yr$^{-1}$. Accurately estimating N leaching rates is important given downstream consequences such as eutrophication and acidification.*
*Following the reviewer's suggestion (here and in the reviewer's third major point), we have conducted a sensitivity analysis of $f_{alloc,Nfix}$, which is the parameter describing plant C allocation to N-fixing bacteria (Fig. R1). This is the only parameter which was tuned rather than derived from observations because there are no studies or available methodologies which explicitly measure plant C allocation to N-fixing bacteria. Essentially, we examined a range of parameter values for $f_{alloc,Nfix}$ (0.01, 0.02, 0.05, 0.10, 0.20, 0.50, and 1.00). Then, we compared nodule biomass C to literature observations which were not used for model parameterisation (Boring and Swank, 1984). Parameter values were chosen such that simulated nodule biomass C was most similar to observed nodule biomass C. We have added*

*this sensitivity analysis to Appendix C. As a result, early successional BNF simulated by LM4.1-BNF was improved.*

*Additionally, we have added an analysis of the factors controlling both symbiotic and asymbiotic BNF rate (Fig. D3 and D4 (below)). For symbiotic BNF we present nodule biomass C, non-structural C, and N stress. For asymbiotic BNF we present soil microbe biomass C, soil microbe biomass N, soil C, and soil N. The relationship between these factors and BNF is specified by the model equations and we present figures for their visualization.*

*Figure 6 (new version):*

[Figure]

**Figure R1: Sensitivity analysis of $f_{alloc,Nfix}$ in LM4.1-BNF. This figure follows Fig. 6 in the main text (and above) for different values of $f_{alloc,Nfix}$. a. $f_{alloc,Nfix} = 0.01$, b. $f_{alloc,Nfix} = 0.02$, c. $f_{alloc,Nfix} = 0.05$, d. $f_{alloc,Nfix} = 0.1$, e. $f_{alloc,Nfix} = 0.2$, f. $f_{alloc,Nfix} = 0.5$, and g. $f_{alloc,Nfix} = 1.0$.**

[Figure]

***Figure D3: Controls of asymbiotic BNF. a. Soil microbe biomass C, b. soil microbe biomass N, c. soil C, and d. soil N.***

[Figure]

***Figure D4: Controls of symbiotic BNF. a. Nodule biomass C, b. non-structural C, and c. N stress.***

[Figure]

2. The LM4.1-BNF simulated C cycle is convincing (e.g., biomass, GPP, NPP), however, the N cycle showed largely bias, compared with observations (e.g., Figure 7). LM4.1-BNF has too much inorganic N (both NH4 and NO3), while very low soil organic N. Does it imply that the soil nitrogen immobilization rate and soil organic matter formation rate are largely underestimated? Or the soil organic matter C:N stoichiometry was problematic?

LM4.1-BNF had reasonable NO3 leaching loss, but overestimated N2O emission, which means the system is too open/leaky. Is it part of the reason why less available inorganic nitrogen is incorporated into soil organic nitrogen pool?

***AUTHORS' RESPONSE:***

*While the alignment between simulated and observed N cycling metrics are not perfect, almost all simulated soil C and N cycling metrics (total soil C, total soil N, ammonium, nitrate, N mineralization rate, net nitrification, $N_2O$ emission rate, NO emission rate, and $NO_3^-$ leaching rate) are within two standard deviations of observations (with the exception of $NH_4^+$ leaching rate, which was underestimated by LM4.1-BNF), and are thus reasonable estimations. Generally, validation of land model N cycling is not as rigorous as our study, which examined a multitude of pools and fluxes of N at the site scale. Additionally, it is important to note that soil N cycling is notorious for its high variability (Kuzyakov and Blagodatskaya, 2015). Simulated $N_2O$ emission rate, which was not represented by LM3-SNAP, is slightly overestimated by LM4.1-BNF but is within two standard deviations of observations and is particularly notorious for its high variability (Barton et al., 2015). Unfortunately, N immobilization rate and soil organic matter formation rate data are not available for Coweeta Hydrologic Laboratory. It is possible that slightly underestimated total soil C and total soil N but slightly overestimated soil inorganic N is due to the protection of soil organic matter via physical occlusion in microaggregates or chemical adsorption to mineral surfaces (Krull et al., 2003), which is currently not represented but will be represented in the following version.*

3. Model structure change versus parameterization.  LM4.1-BNF improve the existing model structure and showed some improvement in model performance at the site level. But 1) how much of the performance improvement could be achieved just by tuning existing parameters; 2) how much of additional model uncertainty is introduced by the model structure changes and by adding new model parameters? I would suggest running some sensitivity tests (e.g., perturb critical model parameters) for both LM4.1-BNF and LM3-SNAP to fully answer those questions.

*AUTHORS' RESPONSE:*
*A major accomplishment of LM4.1-BNF is that it combines LM4.1 (Shevliakova et al., in prep), which represents community dynamics but not N cycling, and LM3-SNAP (Sulman et al., 2019), which represents N cycling but not community dynamics. LM4.1-BNF merges their advantages, forming a land model that represents both community dynamics and N cycling. Incorporating community dynamics and N cycling (independently and ultimately together) are both a key focus of land model development.*
*The main parameter in LM3-SNAP that controls symbiotic BNF (note that asymbiotic BNF is not represented in LM3-SNAP) is the efficiency of N-fixing bacteria (ε), which describes plant C allocation to N-fixing bacteria. We have conducted a sensitivity analysis of ε (Fig. R2). It showed that, for all values of ε, symbiotic BNF persists in late succession because there is no competitive exclusion of N-fixing plants by non-fixing plants in LM3-SNAP, which represents a single general plant C pool capable of BNF and cannot represent community dynamics. As such, it would not be possible to effectively represent symbiotic BNF in LM3-SNAP – including the representation of community dynamics from LM4.1 is critical.*
*Unfortunately, the uncertainty introduced by structure vs. parameterisation is not feasible to address because of the multitude of features that have changed from LM4.1 and LM3-SNAP in developing LM4.1-BNF by incorporating both community dynamics and N cycling. However, we believe that the importance of their incorporation and the clear improvements in the representations of asymbiotic and symbiotic BNF justify potential uncertainty.*

*Note that all parameters in LM4.1-BNF were derived from observations with the exception of $f_{alloc,Nfix}$, as described in our response to the reviewer's first major point. We have conducted a sensitivity analysis for $f_{alloc,Nfix}$ (Fig. R1). Because all other parameters were derived from observations, we did not conduct a sensitivity analysis for these parameters.*

*Figure R2: Sensitivity analysis of ε in LM3-SNAP. This figure follows Fig. 6 in the main text (and above) for different values of ε. a. ε = 0.1, b. ε = 0.2, c. ε = 0.4, and d. ε = 0.5.*

[Figure]

Specific comments:
P1L13 is important for nitrogen enabled land model
- ***AUTHORS' RESPONSE: We have changed the wording of this sentence.***
P2L47 another group of land models simulates N2 fixation with nitrogenase enzymatic activity (e.g., Vmax, KM kinetic parameters), root nodulation, and temperature constraint (CABLE, Wang 2007; ELM-ECA Zhu 2019).
- ***AUTHORS' RESPONSE: Thank you for pointing us to these land models. We had referenced CABLE in the following paragraph and we have added a reference to ELM in the following paragraph.***

P2L63 downregulation of N fixation under weak N limitation must be discussed together with upregulation of N fixation under strong P limitation. Otherwise, the model will simulate an extremely low N fixation rate over tropical ecosystems that has abundant N.

- *AUTHORS' RESPONSE: Because LM4.1 does not represent P cycling we unfortunately cannot explore the relationships between BNF and P cycling in this study. Additionally, the links between symbiotic BNF and P limitation are unclear. While it has been previously suggested that N-fixing plants can invest N into P acquisition (i.e., phosphatases), explaining their persistence in non-N-limited and P-limited ecosystems such as tropical forests (Houlton et al., 2008), recent empirical studies have found no relationship between symbiotic BNF and phosphatase activity (Batterman et al., 2018; Sullivan et al., 2014). Furthermore, in a non-N-limited tropical ecosystem, we would expect low symbiotic BNF (Batterman et al., 2013). Plant N demand is satisfied by N mineralization, asymbiotic BNF, and atmospheric N deposition (Hedin et al., 2009).*

P3 L70-77. It will be good to add some discussion about the importance of asymbiotic N2 fixation so that the existing land model should include this flux. For example, how much of global N2 fixation is from asymbiotic fixation, how efficient can plants get nitrogen from asymbiotic pathway.

- *AUTHORS' RESPONSE: This is an excellent suggestion. We have added a description of the importance of asymbiotic BNF at the global scale to the main text.*

P4L101. It will make the model evaluation more concrete if the model could run at least two sites (tropical site with abundant soil N versus the temperate site with relatively strong N limitation).

- *AUTHORS' RESPONSE: We agree that an analysis of a tropical N-fixing tree / site would be extremely interesting. However, this would be a substantial exercise on top of the intensive study that we have already performed and would deserve its independent journal article. Furthermore, it is common that land model development and initial analyses are performed at a single site, then expanded in further work.*

P4L108-117 belongs to section 2. model description

- *AUTHORS' RESPONSE: We believe that it is important to provide a brief description of the main analyses of our study in the "Introduction". We have described these in greater detail in the "Numerical experiments and evaluation description".*

P4L122 Does LM4.1-BNF have to run at fixed spatial resolution (roughly 1degree by 1 degree)? Or it could run at any resolution? How about the temporal resolution?

- *AUTHORS' RESPONSE: LM4.1-BNF can be simulated at a single grid cell (which is approximately 100 km by 100 km), as in our study. LM4.1-BNF can also be simulated at multiple grid cells or at the global scale, and can be coupled with the GFDL atmosphere model to serve as a base for the GFDL climate and Earth system models (Zhao et al., 2018a, 2018b). We have provided more detail on this in Section 2.1. Additionally, LM4.1-BNF can be simulated for any length of time. We simulated LM4.1-BNF for 300 years because this is a reasonable time scale for forest succession. In LM4.1-BNF, different processes occur at different time scales to reflect observations, as described in Appendix A.*

P5L138-140. What are fine root, sapwood, heartwood, seed C:N ratios, how about soil organic pools C:N ratios?

- *AUTHORS' RESPONSE: The C:N ratios for fine root, sapwood, heartwood and seed are given in Table D2. We did not present them in Table 1 because they do not differ between vegetation types. We added a reference to Table D2 to clarify. The soil C:N ratio is not fixed. We added a description to the main text to clarify.*

L6 Eq1,2. How to parameterize rNfix? What is the functional shape of f(T)?

- *AUTHORS' RESPONSE: Thank you for pointing out that this was unclear. We have added a description of the derivation of $r_{Nfix}$ to Appendix C. Essentially, $r_{Nfix}$ was calculated with data from Bytnerowicz et al., (in review) which measured nodule biomass % C and symbiotic BNF rate per unit nodule biomass using $^{15}N_2$ incorporation. The shape of the soil temperature dependence function for symbiotic BNF is a modified beta distribution (Yan and Hunt, 1999) and was previously displayed in Fig. D1. We have moved this figure to the main text and provided more details on its shape in the main text.*

P7L180 One microbe pool for all soil microbial activities? Please justify.

- *AUTHORS' RESPONSE: Although combining decomposers, nitrifiers, denitrifiers, and asymbiotic N-fixers in a single soil microbe C pool is a simplification, we believe that representing separate soil microbe C pools for each function would introduce too much complexity (equations and parameters) given that the results from a single soil microbe C pool are reasonable.*

P7L185 How to parameterize rNfixasymb? What is the functional shape of f(T)?

- *AUTHORS' RESPONSE: Thank you for pointing out that this was unclear. The derivation of $r_{Nfix\ asymb}$ was described in Appendix C. Essentially, we divided observed asymbiotic BNF rate (Reed et al., 2011) by observed soil microbe biomass C (Chapin et al., 2011; Scharlemann et al., 2014). The shape of the soil temperature dependence function for asymbiotic BNF is a modified normal distribution function and was previously displayed in Fig. D1. We have moved this figure to the main text and provided more details on its shape in the main text.*

P11L312 Did LM4.1-BNF-NPP and LM4.1-BNF-ET restart from steady-state of LM4.1-BNF spinup simulation? Will LM4.1-BNF spinup steady-state differ dramatically from LM4.1-BNF-NPP(/ET) steady-state?

- *AUTHORS' RESPONSE: LM4.1-BNF, LM4.1-BNF$_{NPP}$, and LM4.1-BNF$_{ET}$ numerical experiments were all initialised from the LM4.1-BNF spin up. Because the LM4.1-BNF spin up only included Acer (which is a non-fixer), it is the same for LM4.1-BNF, LM4.1-BNF$_{NPP}$, and LM4.1-BNF$_{ET}$ which only differ with respect to symbiotic BNF.*

P13L365 Here need more discussion about the model bias in dbh growth rate. For example, is that because N stress is too weak?

- *AUTHORS' RESPONSE: Even though the alignment between simulated and observed dbh growth rates is not perfect, LM4.1-BNF makes reasonable estimations. Additionally, the central improvement is that LM3-SNAP (and most other land models) cannot distinguish between plant cohorts with different dbhs or distinguish between Robinia and Acer (because they cannot represent community dynamics). As the reviewer suggested in their first major point, this is not the focus of our study, so we have rewritten Section 4 to emphasise BNF.*

P16Figure3 The y-axis scale is misleading (suggest not using log-scale). The model vs data difference is actually big. Again, discuss why and how it occur in LM4.1 model. How is the LM4.1-SNAP simulated density, compared with FIA data?

- *AUTHORS' RESPONSE: Thank you for this suggestion. We have changed the logarithmic scale to a linear scale. Similarly to our response to P15, even though the alignment between simulated and observed dbh distributions is not perfect, LM4.1-BNF makes reasonable estimations. Additionally, the central improvement is that LM3-SNAP (and most other land models) cannot distinguish between plant cohorts with different dbhs or distinguish between Robinia and Acer (because they cannot represent community dynamics). As the reviewer suggested in their first major point, this is not the focus of our study, so we have rewritten Section 4 to emphasise BNF.*

P25L480 and Figure 10. What's the implication of zero BNF rate during the late succession when the model initialized with mixed Acer and Robinnia? The whole ecosystem will rely on soil mineralization generated inorganic N? Will all plants gradually die?

- *AUTHORS' RESPONSE: Low or zero BNF rate is expected in late succession (Boring and Swank, 1984). Plant N demand is satisfied by N mineralization, asymbiotic BNF, and atmospheric N deposition. Please see our response to the reviewer's first major point for more details.*

Reference

Wang, Y.P., Houlton, B.Z. and Field, C.B., 2007. A model of biogeochemical cycles of carbon, nitrogen, and phosphorus including symbiotic nitrogen fixation and phosphatase production. Global Biogeochemical Cycles, 21(1).

Zhu, Q., Riley, W.J., Tang, J., Collier, N., Hoffman, F.M., Yang, X. and Bisht, G., 2019. Representing nitrogen, phosphorus, and carbon interactions in the E3SM land model: Development and global benchmarking. Journal of Advances in Modeling Earth Systems, 11(7), pp.2238-2258.

*Literature cited*

Barton, L., Wolf, B., Rowlings, D., Scheer, C., Kiese, R., Grace, P., Stefanova, K. and Butterbach-Bahl, K.: Sampling frequency affects estimates of annual nitrous oxide fluxes, Sci. Rep., 5, 1–9, doi:10.1038/srep15912, 2015.

Batterman, S. A., Hedin, L. O., van Breugel, M., Ransijn, J., Craven, D. J. and Hall, J. S.: Key role of symbiotic dinitrogen fixation in tropical forest secondary succession, Nature, 502(7470), 224–227, doi:10.1038/nature12525, 2013.

Batterman, S. A., Hall, J. S., Turner, B. L., Hedin, L. O., LaHaela Walter, J. K., Sheldon, P. and van Breugel, M.: Phosphatase activity and nitrogen fixation reflect species differences, not nutrient trading or nutrient balance, across tropical rainforest trees, Ecol. Lett., 21(10), 1486–1495, doi:10.1111/ele.13129, 2018.

Boring, L. R. and Swank, W. T.: The Role of Black Locust (Robinia Pseudo-Acacia) in Forest Succession, J. Ecol., 72(3), 749–766, 1984.

Bytnerowicz, T. A., Akana, P. R., Griffin, K. L. and Menge, D. N. L.: The temperature sensitivity of woody dinitrogen fixation across species and growing temperatures, n.d.

Chapin, F. S., Matson, P. A. and Vitousek, P. M.: Principles of Terrestrial Ecosystem Ecology, Springer Science & Business Media., 2011.

Hedin, L. O., Brookshire, E. N. J., Menge, D. N. L. and Barron, A. R.: The Nitrogen Paradox in Tropical Forest Ecosystems, Annu. Rev. Ecol. Evol. Syst., 40(1), 613–635, doi:10.1146/annurev.ecolsys.37.091305.110246, 2009.

Houlton, B. Z., Wang, Y. P., Vitousek, P. M. and Field, C. B.: A unifying framework for dinitrogen fixation in the terrestrial biosphere, Nature, 454(7202), 327–330, doi:10.1038/nature07028, 2008.

Krull, E. S., Baldock, J. A. and Skjemstad, J. O.: Importance of mechanisms and processes of the stabilisation of soil organic matter for modelling carbon turnover, Funct. Plant Biol., 30(2), 207–222, doi:10.1071/FP02085, 2003.

Kuzyakov, Y. and Blagodatskaya, E.: Microbial hotspots and hot moments in soil: Concept & review, Soil Biol. Biochem., 83, 184–199, doi:10.1016/j.soilbio.2015.01.025, 2015.

Reed, S. C., Cleveland, C. C. and Townsend, A. R.: Functional Ecology of Free-Living Nitrogen Fixation: A Contemporary Perspective, Annu. Rev. Ecol. Evol. Syst., 42(1), 489–512, doi:10.1146/annurev-ecolsys-102710-145034, 2011.

Scharlemann, J. P. W., Tanner, E. V. J., Hiederer, R. and Kapos, V.: Global soil carbon: Understanding and managing the largest terrestrial carbon pool, Carbon Manag., 5(1), 81–91, doi:10.4155/cmt.13.77, 2014.

Shevliakova, E., Malyshev, S., Martinez-Cano, I., Milly, P. C. D., Pacala, S. W., Ginoux, P., Dunne, K. A., Dunne, J. P., Dupuis, C., Findell, K., Ghannam, K., Horowitz, L. W., John, J. G., Knutson, T. R., Krasting, P. J., Naik, V., Zadeh, N., Zeng, F. and Zeng, Y.: The land component LM4.1 of the GFDL Earth System Model ESM4.1: biophysical and biogeochemical processes and interactions with climate, n.d.

Sullivan, B. W., Smith, W. K., Alan, R., Nasto, M. K., Reed, S. C. and Chazdon, R. L.: Spatially robust estimates of biological nitrogen (N) fixation imply substantial human alteration of the tropical N cycle, Proc. Natl. Acad. Sci., 111(22), 8101–8106, doi:10.1073/pnas.1511978112, 2014.

Sulman, B. N., Shevliakova, E., Brzostek, E. R., Kivlin, S. N., Malyshev, S., Menge, D. N. L. and Zhang, X.: Diverse mycorrhizal associations enhance terrestrial C storage in a global model, Global Biogeochem. Cycles, 33(4), 501–523, doi:10.1029/2018GB005973, 2019.

Yan, W. and Hunt, L. A.: An equation for modelling the temperature response of plants using only the cardinal temperatures, Ann. Bot., 84(5), 607–614, doi:10.1006/anbo.1999.0955, 1999.

Zhao, M., Golaz, J.-C., Held, I. M., Guo, H., Balaji, V., Benson, R., Chen, J.-H., Chen, X., Donner, L. J., Dunne, J. P., Dunne, K., Durachta, J., Fan, S.-M., Freidenreich, S. M., Garner, S. T., Ginoux, P., Harris, L. M., Horowitz, L. W., Krasting, J. P., Langenhorst, A. R., Liang, Z., Lin, P., Lin, S.-J., Malyshev, S. L., Mason, E., Milly, P. C. D., Ming, Y., Naik, V., Paulot, F., Paynter, D., Phillipps, P., Radhakrishnan, A., Ramaswamy, V., Robinson, T., Schwarzkopf, D., Seman, C. J., Shevliakova, E., Shen, Z., Shin, H., Silvers, L. G., Wilson, J. R., Winton, M., Wittenberg, A. T., Wyman, B. and Xiang, B.: The GFDL Global Atmosphere and Land Model AM4.0/LM4.0: 1. Simulation Characteristics With Prescribed SSTs, J. Adv. Model. Earth Syst., 10(3), 691–734, doi:10.1002/2017MS001208, 2018a.

Zhao, M., Golaz, J.-C., Held, I. M., Guo, H., Balaji, V., Benson, R., Chen, J.-H., Chen, X., Donner, L. J., Dunne, J. P., Dunne, K., Durachta, J., Fan, S.-M., Freidenreich, S. M., Garner, S. T., Ginoux, P., Harris, L. M., Horowitz, L. W., Krasting, J. P., Langenhorst, A. R., Liang, Z., Lin, P., Lin, S.-J., Malyshev, S. L., Mason, E., Milly, P. C. D., Ming, Y., Naik, V., Paulot, F., Paynter, D., Phillipps, P., Radhakrishnan, A., Ramaswamy, V., Robinson, T., Schwarzkopf, D., Seman, C. J., Shevliakova, E., Shen, Z., Shin, H., Silvers, L. G., Wilson, J. R., Winton, M., Wittenberg, A. T., Wyman, B. and Xiang, B.: The GFDL Global Atmosphere and Land Model AM4.0/LM4.0: 2. Model Description, Sensitivity Studies, and Tuning Strategies, J. Adv. Model. Earth Syst., 10(3), 735–769, doi:10.1002/2017MS001209, 2018b.

---

## Author Comment (AC2)

In this study, Kou-Giesbrecht reported the GFDL LM4.1-BNF model with a new representation of biological nitrogen fixation and evaluated the impact of competition between nitrogen-fixing and non-fixing plants on simulated carbon, nitrogen and demographic dynamics in a temperate forest site. They showed the LM4.1-BNF did a fair job in simulating the many lumped variables reported in the US forest inventory and analysis database for a temperate forest site at Coweeta Hydrologic Laboratory in North Carolina. Particularly, they showed that the competition between N-fixing and non-N fixing plants is an important factor in interpreting the dynamics of carbon accumulation. Overall, the paper is clearly written, but there are some issues need to be resolved before the model is able to be considered as doing sufficiently well.

More details are listed below

The abstract is generally OK; however, it lacks details on the model performance. The authors may consider to add more content from their model evaluation against the observational data.

*AUTHORS' RESPONSE: Thank you for this excellent suggestion. We have added a comparison between simulated and observed symbiotic BNF and asymbiotic BNF to the abstract to emphasise the ability of LM4.1-BNF to effectively reproduce them: "LM4.1-BNF effectively reproduces asymbiotic BNF rate (13 kg N ha$^{-1}$ yr$^{-1}$) in comparison to observations (11 kg N ha$^{-1}$ yr$^{-1}$). Importantly, LM4.1-BNF effectively reproduces the temporal dynamics of symbiotic BNF rate: LM4.1-BNF simulates a symbiotic BNF rate pulse in early succession that reaches 73 kg N ha$^{-1}$ yr$^{-1}$ at 15 years then declines to ~ 0 kg N ha$^{-1}$ yr$^{-1}$ at 300 years, similarly to observed symbiotic BNF which reaches 75 kg N ha$^{-1}$ yr$^{-1}$ at 17 years then declines to ~ 0 kg N ha$^{-1}$ yr$^{-1}$ in late successional forests."*

Technical description is long but written well.

*AUTHORS' RESPONSE: We have attempted to reduce Section 2 (Model description) by summarising the relevant aspects of LM4.1-BNF (asymbiotic BNF, symbiotic BNF, N limitation, etc.) and describing LM4.1-BNF in detail in Appendix A.*

For results, my major complain is the authors have yet to demonstrate the model LM4.1-BNF compares well with high frequency temporal data, such as eddy flux measurement of carbon and water fluxes. The comparison with lumped data is fair, but not great. This is especially important to evaluate the effect of new temperature response function. Perhaps the authors should consider applying the model for a site with eddy flux measurements as well?

*AUTHORS' RESPONSE: This is an excellent point. We have added an analysis of eddy covariance data from Coweeta Hydrologic Laboratory to the manuscript. We compared observed and simulated net ecosystem exchange / production at the hourly timescale for the growing season and for the non-growing season (Fig. 7, below). LM4.1-BNF effectively reproduces net ecosystem exchange / production at the hourly timescale, and is improved relative to LM3-SNAP.*

*Figure 7: Simulated net ecosystem production (NEP) by LM4.1-BNF and LM3-SNAP compared to net ecosystem exchange (NEE) CHL site data at the hourly timescale. Simulated data are averaged over the last 100 yr of the 300 yr simulation to reflect the data which is from mature forests.*

[Figure]

Further the discussion is a little bit detached from results. Authors may consider move some of the analysis into discussion to better explain the significance of updated processes.

*AUTHORS' RESPONSE: Thank you for this suggestion. We have split Section 4 (Evaluation) into two sections: 4.1 Evaluation results and 4.2 Evaluation discussion. We hope that this will clarify the results and their significance.*

Code availability: some of my colleagues say "upon request" is a bad exercise. Authors should at least provide whom to send such a request, or provide a web link to send such a request.

*AUTHORS' RESPONSE: Thank you for pointing this out. We have added a contact email to the Code availability section. Additionally, LM4.1 code will be made available shortly (Shevliakova et al., in prep).*

Other comments

Table 2. Missing group separation between 2nd and 3rd sets of analyses?

*AUTHORS' RESPONSE: We did not distinguish between the second and third analyses because we used the same six numerical experiments for both analyses. We have clarified this in the table caption.*

Figure 2. Why does the model under predict the low dbh growth rates?

*AUTHORS' RESPONSE: Even though the alignment between simulated and observed dbh growth rates is not perfect, LM4.1-BNF makes reasonable estimations. Additionally, the central improvement is that LM3-SNAP (and most other land models) cannot distinguish between plant cohorts with different dbhs or distinguish between Robinia and Acer (because they cannot represent community dynamics).*

Figure 4. Use stronger color contrast between two FIA data? Or maybe even different symbols? Currently, it is not easy to differentiate them.

*AUTHORS' RESPONSE: Thank you for pointing out that this was unclear. We have changed the colours and symbols (below).*

*Figure 4: Simulated relative basal area of Acer and Robinia over time compared to FIA data (in North Carolina). Simulated data are trees with dbh > 12.7 cm to reflect the dbh range of FIA data. FIA data of all non-fixing trees are aggregated to represent Acer. Each point represents an FIA plot. See Fig. D9 for absolute basal area.*

[Figure]

Figure 6. What is the uncertainty of the data points? Also, why does LM3-SNAP predict more evident oscillations?

*AUTHORS' RESPONSE: Unfortunately, Boring and Swank 1984 did not provide uncertainty for their symbiotic BNF rate measurements. LM3-SNAP exhibits more evident oscillations in late-successional symbiotic BNF because it is non-zero, whereas, in LM4.1-BNF, late-successional symbiotic BNF is zero. Both LM3-SNAP and LM4.1-BNF are forced with the same meteorological forcing* (Sheffield, Goteti, & Wood, 2006) *which spans 1948 to 1978 and is looped, causing cycles of approximately 30 years. If late-successional symbiotic BNF were non-zero in LM4.1-BNF, it would also exhibit evident oscillations.*

Figure 9a: what happened to LM4.1-BNF$_{NPP}$? Why its time series is much shorter?

*AUTHORS' RESPONSE: Thank you for pointing out that this was unclear. In LM4.1-BNF$_{NPP}$, BNF (and N deposition and mineralization) is insufficient to sustain total plant biomass C accumulation, i.e., the plant dies. We have extended the line representing LM4.1-BNF$_{ET}$ to 300 years to clarify that it maintains zero total plant biomass C.*

Equation (A1), why is there no adsorption effect considered? The behavior of NH4 and NO3 are quite different in soil.

*AUTHORS' RESPONSE: This is an excellent point. The different behaviour of NH$_4^+$ and NO$_3^-$ in soil is reflected by solubility parameters, which differ between NH$_4^+$ and NO$_3^-$. The solubility of NO$_3^-$ is greater than that of NH$_4^+$ which reflects a higher cation exchange capacity than anion exchange capacity. These solubility parameters determine $NO_3(k)$ and $NH_4(k)$ in Equation (A1). We have added a description of this to Appendix A.*
* * *
*Literature Cited*

Sheffield, J., Goteti, G., & Wood, E. F. (2006). Development of a 50-Year High-Resolution Global Dataset of Meteorological Forcings for Land Surface Modeling. *Journal of Climate*, *19*(13), 3088–3111. https://doi.org/10.1175/JCLI3790.1